# DUSP26 protects against acute kidney injury by dephosphorylating p53 at serine 312

Ying Fu[1,2,3], Yu Xiang[1], Yun Han[1], Juan Cai[1], Shaobin Duan[1], Anqun Chen[1,2,4] ✉ & Zheng Dong [1,2,5,6] ✉

Acute kidney injury (AKI) remains a leading cause of morbidity and mortality, yet the molecular pathways driving kidney tubule damage in AKI are not fully understood. Here, we report dual-specificity phosphatase 26 (DUSP26) as a critical regulator of kidney tubule injury in AKI. In our study, DUSP26 expression was markedly reduced in kidney biopsies from AKI patients of both sexes and in male murine models of cisplatin nephrotoxic and ischemic AKI. This down-regulation was driven by hypermethylation of the gene promoter of DUSP26 in kidney tubular cells. Loss of DUSP26 exacerbated tubular damage, whereas knock-in of DUSP26 specifically in kidney proximal tubule cells conferred protection. Mechanistically, DUSP26 directly bound to p53 to dephosphorylate it at serine 312, dampening the transcriptional activity of p53 towards cell death genes. Pharmacologic inhibition of DUSP26 sensitized kidneys to AKI, whereas DUSP26 overexpression was protective. Pharmacologic inhibition of DUSP26 also exacerbated ischemia-reperfusion injury in the liver. These findings uncover DUSP26 as a key phosphatase guarding against tissue injury by dephosphorylating p53 at serine 312, and highlight the DUSP26-p53 axis as a promising therapeutic target.

Acute kidney injury (AKI) is a major kidney disease characterized by a rapid decline in renal function that is associated with high morbidity, mortality, and healthcare costs[1,2]. Intrarenal causes account for the majority of clinical cases of AKI, with the primary causes being ischemic injury or nephrotoxic injury[3]. Ischemic AKI frequently occurs in clinical settings such as major surgery, hypotension, myocardial infarction, dehydration, and kidney transplantation[4]. At the pathological level, AKI is marked by injury and death of renal tubular epithelial cells[5], which are particularly vulnerable to hypoxia, toxins, proteinuria, and metabolic stress[6,7]. Importantly, AKI is now recognized as a major contributor to the development of chronic kidney disease (CKD), yet no effective therapies currently exist to substantially mitigate AKI or accelerate kidney recovery after AKI[8]. Emerging single-nucleus transcriptomic analyses of human AKI have revealed heterogeneous injury states of kidney tubular cells, underscoring the need to delineate the molecular regulators of epithelial cell fate in AKI[9].

Among the regulatory mechanisms of AKI, the dynamic balance of protein phosphorylation and dephosphorylation—play a central role in modulating protein activity and signaling pathways[10]. While kinase-mediated phosphorylation events have been extensively studied in AKI, implicating pathways such as JAK/STAT and MAPK[11–15], much less is known about the role of phosphatases. In this regard, there are reports about classical dual-specificity phosphatases (DUSPs) in oxidative stress, inflammation, and pyroptosis in AKI[16–18]. However, the contribution of non-classical DUSPs and the full landscape of phosphatase-mediated regulation in AKI remains poorly defined. DUSP26 is an

[1]Department of Nephrology, Hunan Key Laboratory of Kidney Disease and Blood Purification, Institute of Nephrology, The Second Xiangya Hospital at Central South University, Changsha, China. [2]National Clinical Research Center for Metabolic Diseases, The Second Xiangya Hospital of Central South University, Changsha, China. [3]Postdoctoral Mobile Station of Basic Medical Sciences, The Second Xiangya Hospital of Central South University, Changsha, Hunan, China. [4]Department of Nephrology, The First Affiliated Hospital of Chongqing Medical University, Chongqing, China. [5]Department of Cellular Biology and Anatomy, Medical College of Georgia at Augusta University, Augusta, GA, USA. [6]Charlie Norwood VA Medical Center, Augusta, GA, USA. ✉e-mail: anqunchen@163.com; zdong@augusta.edu

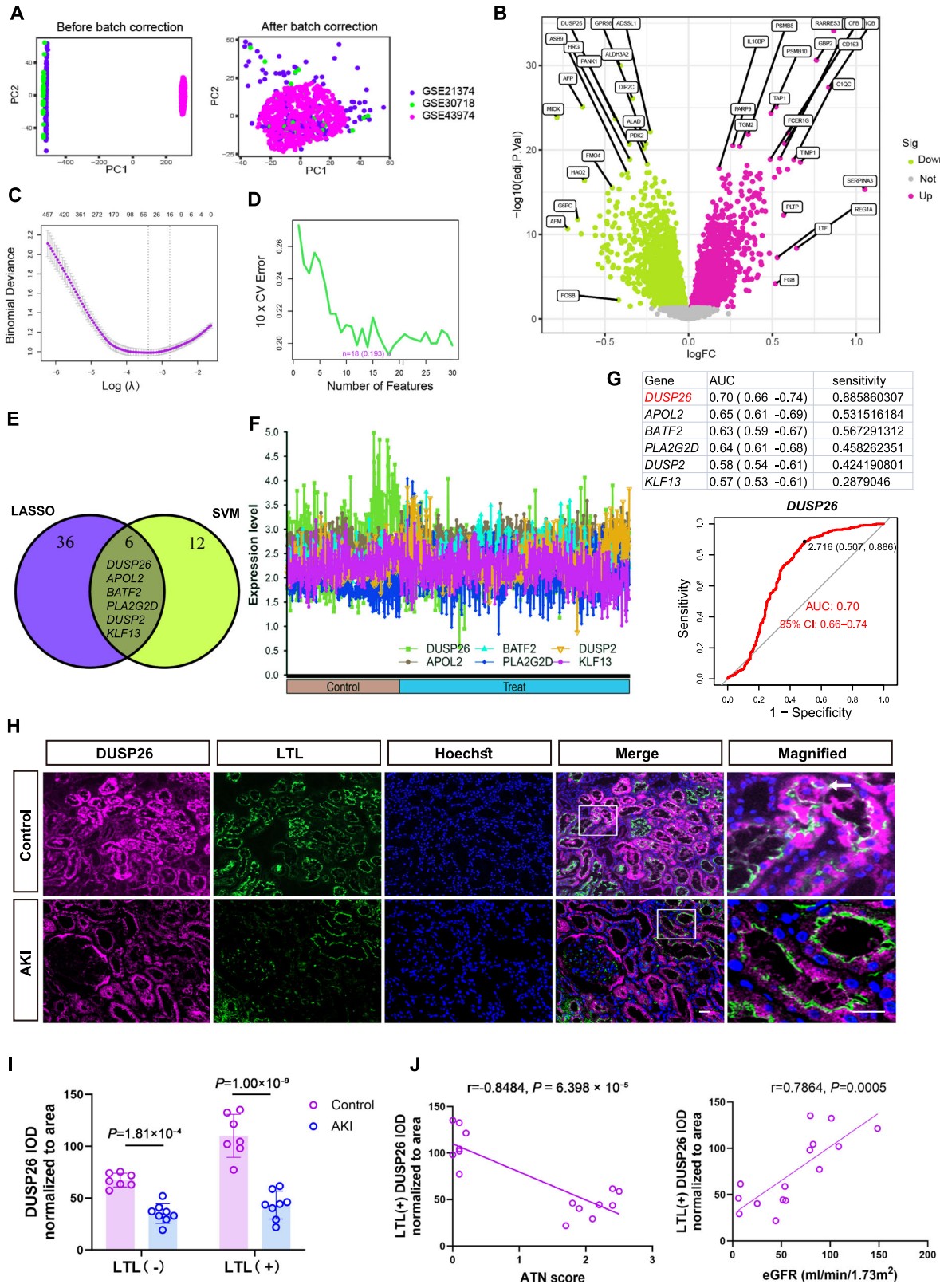

**Nature Communications** | (2026)17:3208

atypical (lacks the conserved ERK-binding motif) dual-specificity phosphatase with a conserved catalytic domain that enables it to target a range of substrates beyond canonical MAPKs[19]. Previous studies have shown that DUSP26 may dephosphorylate p38 while paradoxically enhancing JNK and ERK phosphorylation and signaling, indicating a complex regulatory profile[20,21]. Reflecting its broad biological significance, DUSP26 has been implicated in diverse pathologies

ranging from malignancies (such as neuroblastoma[22,23]) to metabolic diseases (including diabetes[24] and diabetic cardiomyopathy[25]), underscoring its context-dependent roles in cell signaling and disease progression. In search of potential therapeutic targets in AKI, we performed integrative bioinformatic analyses of publicly available AKI datasets from the Gene Expression Omnibus (GEO) and applied machine learning algorithms including LASSO regression and support

**Fig. 1 | Down-regulation of DUSP26 in human AKI. A** Principal component analysis (PCA) of three GEO datasets (GSE21374, GSE30718, GSE43974) before and after batch correction. **B** Volcano plot of DEGs (AKI vs control) analyzed by limma (empirical Bayes moderated t test, two-sided; Benjamini–Hochberg FDR). log2 fold change (logFC) is plotted against −log10(FDR); symbols/categories are defined in the in-panel key. **C** Least Absolute Shrinkage and Selection Operator (LASSO) logistic regression with 10-fold cross-validation in the combined GEO cohort (Control $n = 288$; AKI $n = 587$ biologically independent samples). Points show mean deviance and error bars indicate ±1 s.e.; dashed lines indicate λ_min and λ_1se. **D** Support Vector Machine-Recursive Feature Elimination (SVM-RFE) with 10-fold cross-validation (CV). **E** Overlap of genes identified by LASSO and SVM-RFE. **F** Z-scored expression of the overlapping genes in control and AKI samples. **G** Receiver Operating Characteristic (ROC) analysis. AUC values are shown. **H** Representative immunofluorescence (IF) staining of human kidney biopsies from healthy controls ($n = 7$) and AKI patients ($n = 8$) stained for DUSP26, LTL and nuclei (Hoechst). Scale bar = 50 μm. Similar staining patterns were observed across biologically independent patient biopsies. **I** Quantification of DUSP26 signal (integrated optical density (IOD)/area) in LTL− and LTL+ tubules from (**H**). Individual data points are overlaid; each dot represents one biologically independent patient ($n = 7$ controls; $n = 8$ AKI). For each patient, multiple tubules were quantified and averaged. Data are mean ± SEM; two-way ANOVA with Sidak's multiple-comparisons test (two-sided); exact adjusted $P$ values are shown in the plot. **J** Pearson correlation (two-sided) between LTL + DUSP26 IOD/area and acute tubular necrosis (ATN) score (left) or estimated glomerular filtration rate (eGFR) (right): ATN, $r = -0.8484$, 95% CI [−0.9485, −0.5946], $n = 15$, $t(13) = -5.779$, $P = 6.398 \times 10^{-5}$; eGFR, $r = 0.7864$, 95% CI [0.4591, 0.9257], $n = 15$, $t(13) = 4.590$, $P = 0.0005067$. Lines indicate linear regression fits. Source data are provided as a Source Data file.

vector machines (SVM). These analyses highlighted DUSP26 as a promising candidate gene associated with AKI.

In this study, we systematically investigated the functional role and regulatory mechanisms of DUSP26 in AKI, focusing on its downstream targets and signaling effects in kidney proximal tubular epithelial cells. Our findings reveal a previously unrecognized role of DUSP26 in attenuating tubular injury in AKI through selective dephosphorylation of p53 at serine 312, offering mechanistic insights into AKI pathogenesis and pointing to a potential therapeutic axis for intervention.

## Results

### Down-regulation of DUSP26 in human AKI

To identify key regulators of AKI, we performed integrative transcriptomic analyses across three publicly available AKI-related datasets from the Gene Expression Omnibus (GEO). Following batch effect correction (Fig. 1A), we identified differentially expressed genes (DEGs) between AKI and control samples (Fig. 1B). To prioritize functionally relevant candidates, we applied two machine learning algorithms-least absolute shrinkage and selection operator (LASSO) regression and support vector machine-recursive feature elimination (SVM-RFE). LASSO regression identified optimal variables based on minimal binomial deviance (Fig. 1C), while SVM-RFE ranked genes according to 10-fold cross-validation error (Fig. 1D). Cross-comparison of both approaches revealed a set of overlapping candidate genes (Fig. 1E), among which *DUSP26* emerged as a robustly downregulated gene in AKI samples (Fig. 1F). Receiver operating characteristic (ROC) curve analysis further confirmed the predictive value of DUSP26 for distinguishing AKI from controls (Fig. 1G). To validate its relevance at the protein level, we performed immunofluorescence staining on human kidney biopsy specimens. The clinical data of the AKI patients are presented in Supplementary Table 1. Immunofluorescence analysis revealed decreased DUSP26 expression in kidney biopsy specimens from AKI patients compared to controls, as visualized by co-staining with the proximal tubule marker LTL (Fig. 1H). Quantitative analysis demonstrated significantly lower DUSP26 expression (IOD/area) in LTL-positive tubules from AKI patients (Fig. 1I). Notably, DUSP26 expression levels in LTL-positive tubules showed a strong inverse correlation with acute tubular necrosis (ATN) scores and a strong positive correlation with estimated glomerular filtration rate (eGFR) (Fig. 1J). These findings identify *DUSP26* as a downregulated gene in human AKI and suggest its potential role in maintaining kidney tubular integrity.

### DUSP26 is downregulated in cisplatin- and ischemia-induced acute kidney injury in mice

To explore whether DUSP26 downregulation observed in human AKI also occurs in experimental models, we examined its expression in mouse kidneys following cisplatin-induced nephrotoxic injury. Immunoblotting revealed a marked reduction in DUSP26 protein at 48 h post-injection, which remained significantly suppressed at 72 h (Fig. 2A, B). Given the reports about transcriptional repression of DUSP26 in cancer[20], we next assessed *Dusp26* mRNA that showed a significant decrease in cisplatin-treated mice kidneys (Fig. 2C). Immunofluorescence analysis further confirmed DUSP26 decrease in cisplatin AKI across both renal cortex and outer medulla (Fig. 2D), consistent with tubular injury localization. Supporting these in vivo findings, cisplatin also induced a marked decrease in DUSP26 expression in BUMPT cells (a mouse kidney proximal tubule cell line), with significant suppression at 12 h and further reduction by 36 h (Supplementary Fig. 2A, B).

To determine whether DUSP26 downregulation is a generalizable feature of AKI, we assessed DUSP26 expression in a bilateral renal ischemia-reperfusion injury (IRI) model. Mice subjected to 30 min of bilateral ischemia followed by 48 or 72 h of reperfusion exhibited significant reductions in renal DUSP26 protein (Fig. 2E, F), accompanied by decreased *Dusp26* mRNA expression (Fig. 2G). Immunofluorescence further demonstrated diminished DUSP26 staining within tubular epithelial cells of the injured kidneys (Fig. 2H). We next extended these findings to an in vitro model of hypoxia/reoxygenation (H/R) injury. BUMPT cells cultured under hypoxic conditions for 8 h followed by various durations of reoxygenation (2–10 h) exhibited time-dependent increases in apoptosis, as indicated by cleavage of caspase-3 (c-Cas3; Supplementary Fig. 2C). Notably, DUSP26 protein levels declined progressively from 6 to 10 h of reoxygenation (Supplementary Fig. 2D), and its expression showed a strong inverse correlation with c-Cas3 activation (Supplementary Fig. 2E). Together, these results demonstrate that DUSP26 is consistently downregulated across multiple models of AKI, both in vivo and in vitro, and its loss is closely associated with renal tubular cell injury and apoptosis.

### Promoter methylation mediates DUSP26 downregulation in proximal tubular cells following injury

Although limited information is currently available regarding the transcriptional regulation of DUSP26, existing studies suggest that its expression is primarily modulated at the mRNA level[19]. Recent studies have highlighted DNA methylation as a critical epigenetic mechanism underlying tubular cell dysfunction and gene repression during AKI[26]. Notably, DUSP26 downregulation has been linked to promoter DNA methylation in cancer cells, where treatment with 5-Aza-2′-deoxycytidine (5-Aza, a DNA methyltransferase inhibitor) can restore DUSP26 expression[20]. To explore whether similar epigenetic regulation occurs in kidney tubular injury, we analyzed the promoter region of the mouse *Dusp26* gene using the MethPrimer database. This analysis revealed a prominent CpG island, suggesting potential susceptibility to DNA methylation-mediated transcriptional repression (Fig. 3A). To directly evaluate promoter methylation, we performed methylation-specific PCR (MSP) following bisulfite conversion of genomic DNA. Specific primer pairs were designed to distinguish methylated (M) and unmethylated (U) alleles within the CpG island of

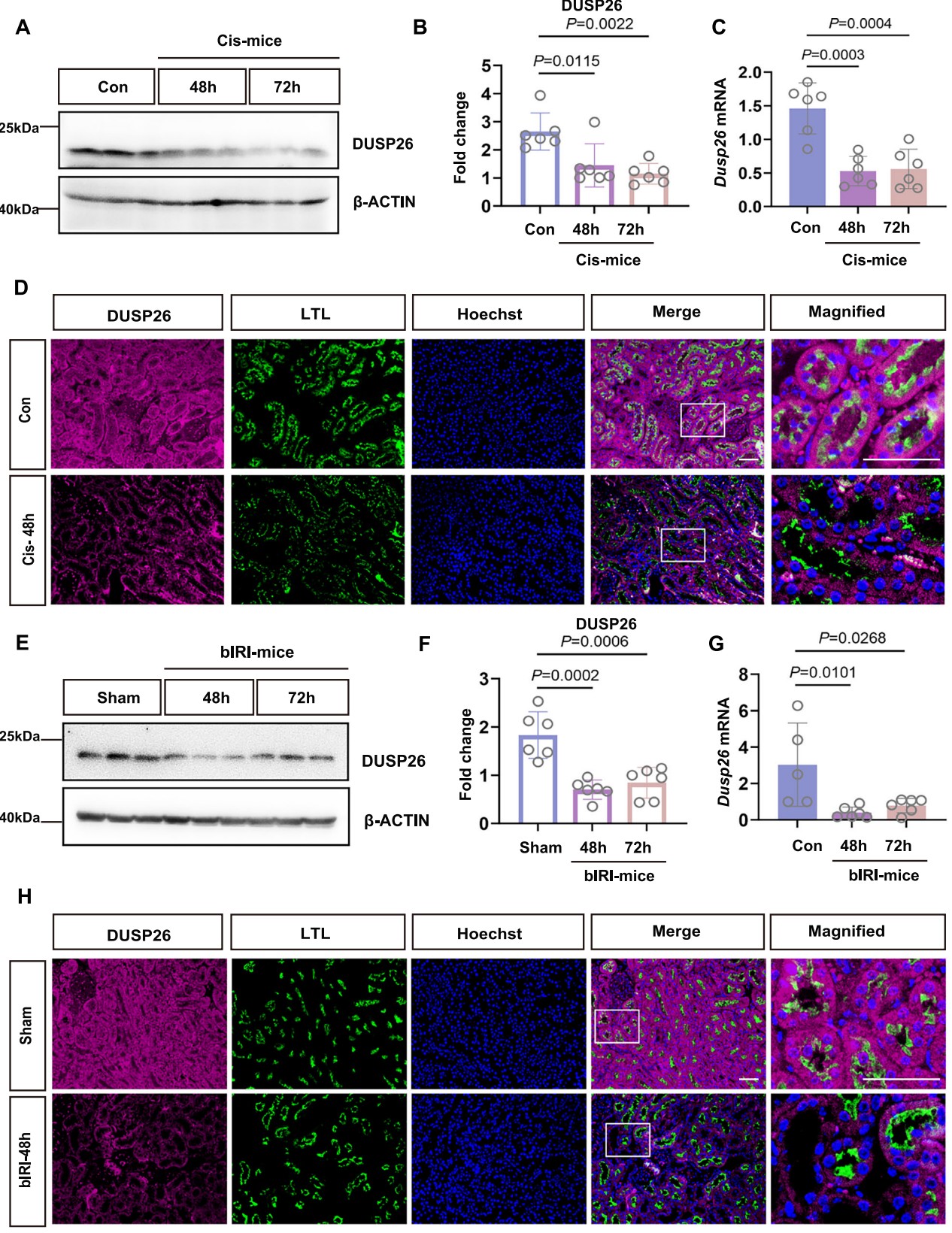

the *Dusp26* promoter (Fig. 3B). MSP revealed increased promoter methylation in both cisplatin-treated and hypoxia/reoxygenation (H/R)-treated BUMPT cells compared with controls, while 5-Aza treatment markedly reduced the methylation signal in both injury models (Fig. 3C). We next examined the kinetics of *Dusp26* re-expression following 5-Aza treatment. qPCR analysis showed that in cisplatin-injured

cells, *Dusp26* mRNA levels remained low at early time points and significantly increased only after 12–24 h of 5-Aza exposure (Fig. 3D). Western blotting further validated that DUSP26 protein levels were restored after 5-Aza treatment (Fig. 3E).

We next assessed the downstream functional consequences of this restoration. While 5-Aza treatment successfully restored DUSP26

**Fig. 2 | DUSP26 is downregulated in cisplatin- and ischemia-induced AKI in mice. A** Representative immunoblot of DUSP26 in kidney cortex and outer medulla at 48 and 72 h after cisplatin (Cis) injection; Con, saline-injected control. **B** Densitometric quantification of DUSP26 in (**A**), normalized to β-ACTIN. Individual data points are overlaid; each dot represents one biologically independent mouse (n = 6 per group). Data are mean ± SEM. **C** Quantitative reverse transcription PCR (qRT–PCR) analysis of *Dusp26* mRNA in kidney cortex and outer medulla at 48 and 72 h after Cis. Individual data points are overlaid; each dot represents one biologically independent mouse (n = 6 per group). Data are mean ± SEM. **D** Representative immunofluorescence (IF) of kidney sections 48 h after Cis showing DUSP26, Lotus tetragonolobus lectin (LTL) and nuclei (Hoechst). Scale bar, 50 μm. Images are representative of biologically independent mice (n = 6 per group) with similar results. **E** Representative immunoblot of DUSP26 in kidney cortex and outer medulla from male C57BL/6 mice subjected to bilateral renal ischemia (bIRI; 30 min) and reperfusion for 48 or 72 h; Sham, sham-operated control. **F** Densitometric quantification of DUSP26 in (**E**), normalized to β-ACTIN. Individual data points are overlaid; each dot represents one biologically independent mouse (n = 6 per group). Data are mean ± SEM. **G** qRT–PCR analysis of *Dusp26* mRNA in kidney cortex and outer medulla at 48 and 72 h after bIRI. Individual data points are overlaid; each dot represents one biologically independent mouse (n = 6 per group). Data are mean ± SEM. **H** Representative IF images showing DUSP26 and LTL in kidneys from Sham and bIRI mice at 48 h post-reperfusion; nuclei (Hoechst). Scale bar, 50 μm. Images are representative of biologically independent mice (n = 6 per group) with similar results. For (**B**, **C**, **F**, **G**), data were normalized to the respective control (Con or Sham) set to 1. Statistical significance was assessed by one-way ANOVA with Tukey's multiple-comparisons test (two-sided; adjusted *P* values). Exact *P* values are shown in the plots. Source data are provided as a Source Data file.

expression (Fig. 3E) and subsequently reduced the phosphorylation of p53 at Ser312 (p-p53(S312)), it paradoxically failed to suppress, and in fact significantly increased, levels of cleaved caspase-3 (c-Cas3) (Supplementary Fig. 8). This suggests that the global, non-specific demethylating activity of 5-Aza likely activates other pro-apoptotic pathways, masking the specific benefit of DUSP26 restoration. To further investigate the mechanism driving this hypermethylation, we assessed promoter occupancy of both the maintenance methyltransferase DNMT1 and the de novo methyltransferases DNMT3A and DNMT3B. ChIP–qPCR analysis revealed significantly increased enrichment of DNMT1 and DNMT3A at the *Dusp26* promoter in cisplatin-treated cells compared with controls, whereas DNMT3B enrichment was not significantly altered (Fig. 3F). Consistent with this finding, total protein levels of the DNMTs themselves were upregulated. Western blot analysis confirmed a time-dependent increase in the expression of DNMT1, DNMT3A, and DNMT3B in cisplatin-treated BUMPT cells (Supplementary Fig. 6A, B). To confirm whether this epigenetic silencing also occurs in vivo, we analyzed kidney tissue from a mouse model of IRI-induced AKI. Quantitative, site-specific analysis using targeted bisulfite sequencing revealed a widespread increase in methylation levels across the *Dusp26* promoter in IRI-AKI mice compared to controls (Fig. 3G, H). Notably, quantification of individual CpG sites confirmed significantly higher methylation at multiple specific loci in the AKI group (Fig. 3I). Collectively, these data demonstrate that DUSP26 is silenced in vitro through promoter hypermethylation accompanied by recruitment of the de novo methyltransferase DNMT3A together with DNMT1, and this epigenetic repression is recapitulated in an in vivo mouse model of AKI.

### DUSP26 functions as a protective factor against acute injury in proximal tubular epithelial cells

To assess the functional role of DUSP26 in acute tubular injury, we initially modulated its expression in BUMPT cells and examined apoptotic markers following cisplatin and H/R challenges. Knockdown of *Dusp26* with specific siRNAs significantly increased cleaved caspase-3 (c-Cas3) levels after cisplatin treatment, as demonstrated by immunoblotting and corresponding quantification (Fig. 4A, B). Consistently, *Dusp26*-knockdown cells exhibited markedly higher c-Cas3 expression following 8 h of H/R than control cells (Fig. 4C, D). Moreover, NSC87877 (a pharmacologic inhibitor of DUSP26) potentiated cisplatin-induced apoptosis as shown by TUNEL staining (Fig. 4E, F).

In contrast, stable overexpression of DUSP26 in BUMPT cells via lentiviral transduction conferred significant protection against cisplatin-induced injury. Immunoblot analysis revealed lower c-Cas3 levels in DUSP26-overexpressing cells after 24 h of cisplatin treatment than in cells transduced with control lentivirus (Fig. 4G, H), which was corroborated by fewer TUNEL-positive cells (Fig. 4I, J). DUSP26 overexpression similarly attenuated caspase 3 cleavage/activation following H/R injury (Fig. 4K, L).

Domain mapping of full-length DUSP26 (211 amino acids) confirmed the presence of a 131-amino acid dual-specificity phosphatase (DSP) catalytic domain, which contains the conserved DSP motif "VHCAVGVSRS" and the MKP motif "AYLM" (Fig. 4M). We generated a phosphatase-inactive mutant by substituting the catalytic cysteine at position 152 with serine (C152S) (Supplementary Fig. 3A). Importantly, this point mutation attenuated the protective effect of DUSP26, indicating that the phosphatase activity of DUSP26 is critical for its cytoprotective function (Fig. 4N, O). Collectively, these results highlight a protective role of DUSP26 in kidney proximal tubular cells via its phosphatase activity.

### Pharmacologic inhibition of DUSP26 sensitizes the kidney to AKI

NSC87877 has been identified as a small-molecule inhibitor of DUSP26[27]. Given the cytoprotective role of DUSP26 in BUMPT cells in vitro, we next investigated whether pharmacologic inhibition of DUSP26 would exacerbate kidney injury in vivo. Mice were pretreated with the DUSP26 inhibitor NSC87877 (NSC) prior to cisplatin administration. At 48 h of cisplatin treatment, NSC-pretreated mice exhibited significantly higher levels of serum creatinine and blood urea nitrogen (BUN) compared to vehicle-treated controls, indicating worsened AKI (Fig. 5A). Renal histology also revealed more severe tubular damage in NSC-treated mice, characterized by tubular dilation, brush border loss, epithelial cell sloughing, and cast formation (Fig. 5B), which was confirmed by a significantly higher tubular injury score (Fig. 5C). Consistent with these findings, immunofluorescence analysis demonstrated increased expression of kidney injury molecule-1 (KIM-1), a proximal tubule injury marker, in the kidneys of NSC-treated mice (Fig. 5D). TUNEL staining further revealed an increased number of apoptotic tubular cells in these tissues (Fig. 5E). Quantitative analysis confirmed a significantly higher number of TUNEL-positive cells in the NSC-pretreated group (Supplementary Fig. 4), suggesting that pharmacologic suppression of DUSP26 amplifies tubular cell death during toxic injury.

To determine whether these effects extend to ischemic injury, we subjected mice to bilateral renal ischemia-reperfusion injury (IRI), with or without NSC pretreatment. NSC administration significantly aggravated renal dysfunction, as evidenced by higher serum creatinine and BUN levels compared to vehicle-treated IRI mice (Fig. 5F). Histopathological analysis confirmed exacerbated tubular damage in the NSC-treated group (Fig. 5G), with increased injury scores (Fig. 5H). Furthermore, Western blot analysis of whole kidney lysates confirmed that NSC-pretreated mice had significantly higher protein levels of both KIM-1 and the apoptotic marker cleaved-Caspase 3 (c-Cas3) (Fig. 5I). Consistent with this finding, these mice also showed an increased number of TUNEL-positive apoptotic cells (Fig. 5J), which was confirmed by quantitative analysis (Supplementary Fig. 4). Together, these results indicate that DUSP26 inhibition sensitizes the

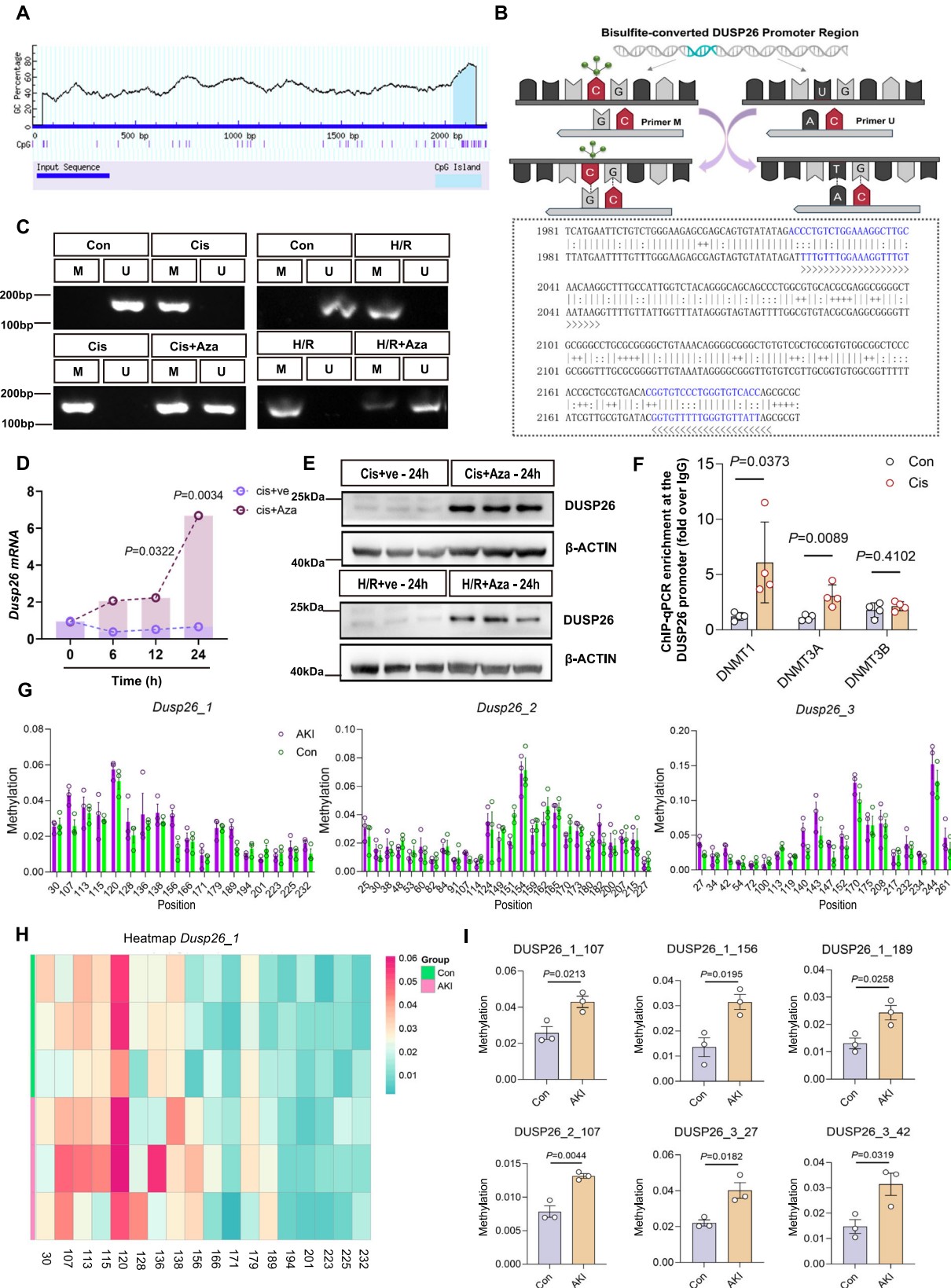

kidney to both toxic and ischemic injury, supporting a protective role of DUSP26 in maintaining tubular epithelial integrity in AKI.

## Proximal tubule-specific knock-in of DUSP26 attenuates AKI

To validate the protective role of DUSP26 in vivo using a genetic approach, we generated proximal tubule-specific DUSP26 knock-in

mice by crossing conditional Rosa26[loxP-STOP-loxP-Dusp26] mice with Pepck-Cre mice, yielding Rosa26[loxP-STOP-loxP-Dusp26]; Pepck-Cre animals (refer-red to as PT-Dusp26-KI), while Cre-negative littermates served as controls (PT-Dusp26-WT) (Fig. 6A, B). We validated the knock-in efficacy. Immunofluorescence staining confirmed strong DUSP26 expression co-localizing with the proximal tubule marker LTL in PT-

**Fig. 3 | Promoter methylation mediates DUSP26 downregulation in proximal tubular epithelial cells following injury. A** Visualization of CpG island distribution within the *Dusp26* promoter region using the MethPrimer database. **B** Genomic DNA was subjected to bisulfite conversion, and methylation-specific (M) and unmethylated (U) primers targeting the CpG-rich region of the *Dusp26* promoter were designed for PCR amplification. **C** Methylation-specific PCR (MSP) of the *Dusp26* promoter in BUMPT cells under cisplatin (Cis) ± 5-aza-2′-deoxycytidine (5-Aza) and hypoxia/reoxygenation (H/R) ± 5-Aza. M, methylated; U, unmethylated. Representative gel from three biologically independent experiments with similar results. **D** Quantitative reverse transcription PCR (qRT–PCR) time course of *Dusp26* mRNA in BUMPT cells treated with Cis (20 μM) with vehicle (Ve) or 5-Aza (1 μM). *n* = 4 biologically independent experiments (independent cell cultures/treatments); technical qPCR replicates were averaged within each experiment. **E** Representative immunoblot of DUSP26 in BUMPT cells treated with 5-Aza after Cis (24 h) or H/R (8 h). Representative blot from four biologically independent

experiments with similar results. **F** ChIP–qPCR showing enrichment of DNMT1, DNMT3A and DNMT3B at the *Dusp26* promoter (fold over IgG) under control vs Cis (24 h). Individual data points are overlaid; each dot represents one biologically independent ChIP sample (*n* = 4). **G** Targeted bisulfite sequencing (BS-seq) of three *Dusp26* promoter amplicons (*Dusp26*_1–3) in control (Con) and IRI-AKI mouse kidneys. Individual data points are overlaid; each dot represents one biologically independent mouse (*n* = 3 per group). Bars show mean methylation percentage at each CpG site; error bars indicate ± SEM. **H** Heatmap of CpG methylation frequency within *Dusp26*_1 from (**G**). **I** Quantification of representative CpG sites from (**G**, **H**). Individual data points are overlaid; each dot represents one biologically independent mouse (*n* = 3 per group). Data are mean ± SEM. Statistical significance was assessed by two-way ANOVA with Sidak's multiple-comparisons test (two-sided; adjusted *P* values) for (**D**, **F**) and unpaired Student's *t* test (two-sided) for (**I**); exact *P* values are shown in the plots. Source data are provided as a Source Data file.

Dusp26-KI mice, which was absent in PT-Dusp26-WT controls (Fig. 6C). This was further confirmed by Western blot, which showed markedly elevated DUSP26 protein levels in the kidneys of PT-Dusp26-KI mice (Fig. 6D). Baseline renal function did not differ significantly between these two groups. However, following cisplatin treatment, PT-Dusp26-KI mice displayed markedly lower serum creatinine and BUN levels at 24 h, indicating better renal function (Fig. 6E). Histological analysis revealed significantly less tubular damage in PT-Dusp26-KI mice compared with WT, as evidenced by H&E staining and quantification of injury scores (Fig. 6F, G). Immunoblotting further demonstrated reduced expression of cleaved caspase-3 and neutrophil gelatinase-associated lipocalin (NGAL), two markers of apoptosis and kidney injury, respectively, in PT-Dusp26-KI mice (Fig. 6H). Consistently, the knock-in group had less TUNEL staining (Fig. 6I). To determine whether this protective effect extends to ischemic injury, PT-Dusp26-KI and PT-Dusp26-WT mice were subjected to bilateral renal ischemia-reperfusion. Similar to the cisplatin model, PT-Dusp26-KI mice exhibited significantly better renal function and histology, as reflected by lower serum creatinine/BUN levels and less tubular injury (Fig. 6J-L). These findings confirm that proximal tubule-specific overexpression of DUSP26 attenuates kidney injury in both toxic and ischemic models of AKI, reinforcing its role as a tubular protective factor.

### DUSP26 suppresses pro-apoptotic gene expression via interaction with p53

To investigate the mechanisms by which DUSP26 protects kidney tubular epithelial cells, we analyzed transcriptomic changes associated with DUSP26 expression in human ischemic AKI from the GEO database (GSE21374, GSE30718, and GSE43974). AKI patients were stratified into high- and low-DUSP26 expression groups, revealing differential expression of p53 and related pro-apoptotic genes, with higher *BAX*, *BAK1*, *CASP3*, and *APAF1* expression in the DUSP26-low group (Fig. 7A). Co-expression analysis confirmed a strong negative correlation between DUSP26 and these p53-regulated apoptotic genes across datasets (Fig. 7B). To experimentally validate these associations in vivo, we measured pro-apoptotic gene expression in mice treated with the DUSP26 inhibitor NSC87877. In both cisplatin-induced AKI and ischemia-reperfusion injury (IRI) models, NSC significantly increased the expression of *Bax*, *Puma*, and *Noxa* mRNA in kidney tissues (Fig. 7C, D). Given that p53 is a key transcriptional regulator of these pro-apoptotic genes, we examined whether DUSP26 modulates p53 directly. DUSP26 knockdown increased total p53 protein levels in cisplatin-treated BUMPT cells, whereas DUSP26 overexpression suppressed p53 accumulation (Fig. 7E). Co-immunoprecipitation (Co-IP) assays further detected the interaction of DUSP26 with p53 following cisplatin exposure (Fig. 7F), suggesting a direct regulatory relationship. To explore whether this interaction involves post-translational modifications, we performed phosphoproteomic profiling in

BUMPT cells with DUSP26 overexpression or knockdown. Among differentially regulated phospho-sites, p53 phosphorylation at serine 312 emerged as the top-ranked site affected by DUSP26 expression, based on Andromeda scoring (Fig. 7G). This residue, conserved across species (corresponding to Ser315 in human p53), may be involved in regulating p53 transcriptional activity and protein stability[28–30]. Together, these findings demonstrate that DUSP26 negatively regulates pro-apoptotic gene expression by interacting with and modulating the phosphorylation status of p53, particularly at Ser312. These findings support a DUSP26–p53 axis that contributes to tubular cell survival during AKI.

### DUSP26 regulates tubular cell apoptosis in AKI by dephosphorylating p53 at serine 312

We then determined whether DUSP26 directly regulates p53 phosphorylation at serine 312. As shown in Fig. 8A, DUSP26 knockdown significantly enhanced, while DUSP26 overexpression suppressed, p53 phosphorylation at Ser312 (p-p53$^{S312}$) following cisplatin treatment. Similar findings were observed in vivo: PT-Dusp26-KI mice had less p-p53$^{S312}$ in kidney tissues than PT-Dusp26-WT littermates after cisplatin treatment (Fig. 8B). To confirm the specificity of this in vivo regulation, we comprehensively examined other key N-terminal phosphorylation sites. Crucially, in the same kidney samples, DUSP26 overexpression had no significant effect on the phosphorylation of p53 at Ser20 or Ser46. While a modest decrease was seen at Ser15, the profound inhibition of Ser312 contrasted with the lack of effect at other major N-terminal sites, demonstrating that the regulatory effect of DUSP26 is highly specific to the Ser312 site (Supplementary Fig. 7). To functionally validate the role of Ser312 phosphorylation, we reconstituted p53-knockout (p53-KO) BUMPT cells with either wild-type (WT) p53 or a Ser312-to-alanine (S312A) mutant. The constructs were confirmed by Sanger sequencing (Supplementary Fig. 3B). Upon cisplatin treatment, S312A-transfected cells exhibited markedly less cell death and cleaved caspase-3 than WT-p53 cells (Fig. 8C-E). Moreover, qPCR analysis showed that transcription of *Bax* and *Puma*, canonical p53 target genes, was significantly attenuated in S312A-expressing cells (Fig. 8F). We further evaluated this mechanism under H/R conditions. DUSP26 knockdown in BUMPT cells enhanced p-p53$^{S312}$ after 8 h of H/R, whereas DUSP26 overexpression suppressed this phosphorylation (Fig. 8G, H). Consistently, *Bax* and *Puma* expression during H/R was lower in S312A p53 cells than in WT-p53 cells (Fig. 8I). We also performed chromatin immunoprecipitation (ChIP) to assess p53's binding to the promoters of its target genes (Fig. 8J). ChIP-qPCR analysis revealed that while wild-type p53 robustly bound to the p53 response elements in both the *Bax* and *Puma* promoters, the binding of the p53-S312A mutant was significantly impaired (Fig. 8K). This indicates that Ser312 phosphorylation is critical for p53's ability to occupy the promoters of its pro-apoptotic targets.

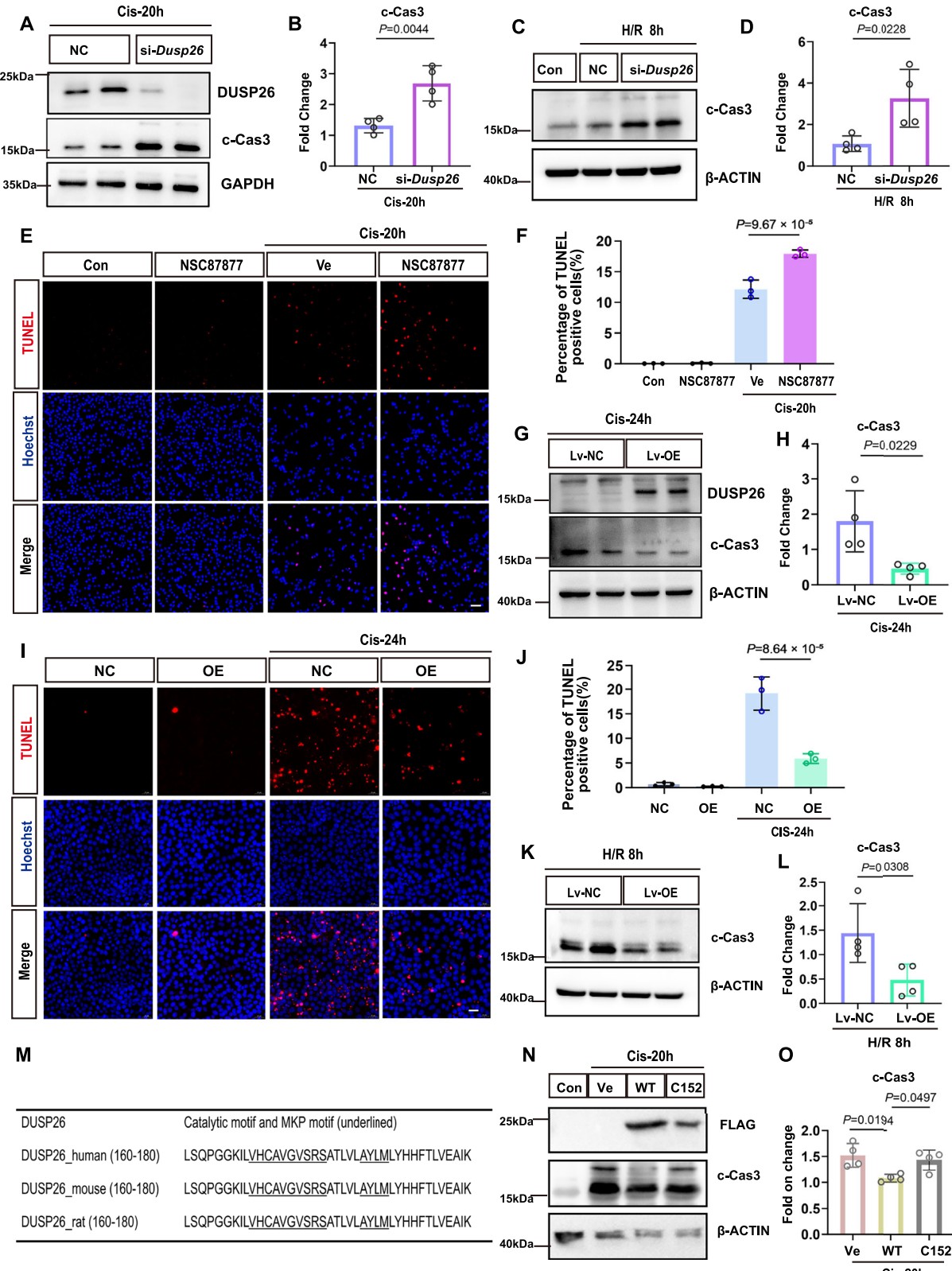

## Phosphorylation of p53 at Ser312 is required for its pro-apoptotic function and AKI in vivo

We further verified the role of p53 Ser312 phosphorylation in vivo using a genetic rescue model. To this end, we re-expressed either wild-type p53 (p53-WT), its non-phosphorylatable p53-S312A mutant, or an empty vector (EV) in p53-knockout (KO) mice via a kidney proximal tubule-specific Ksp-cadherin promoter-driven adeno-associated virus

(AAV) system (Fig. 9A, B). Immunofluorescence analysis confirmed efficient and specific AAV transduction in renal tubular epithelial cells (Fig. 9C). We validated the model's response to cisplatin injury. As expected, a robust p-p53(S312) signal was detected by Western blot in kidneys re-expressing WT p53. In stark contrast, this signal was absent in mice with S312A p53 mutant (Fig. 9D), confirming the non-phosphorylatable nature of the mutant. Importantly, re-expression

**Fig. 4 | DUSP26 functions as a protective factor against acute injury in proximal tubular epithelial cells. A** Immunoblot of DUSP26 and cleaved caspase-3 (c-Cas3) in BUMPT cells transfected with si-NC or si-*Dusp26* and treated with cisplatin. **B** Quantification of c-Cas3 in (**A**) normalized to GAPDH. *n* = 4 biologically independent experiments. **C** Immunoblot of c-Cas3 in BUMPT cells transfected with si-NC or si-*Dusp26* after hypoxia/reoxygenation (H/R). **D** Quantification of c-Cas3 in (**C**) normalized to β-ACTIN. *n* = 4 biologically independent experiments. (**E**) TUNEL staining of BUMPT cells pre-treated with NSC87877 or vehicle (Ve) and then treated with cisplatin; Scale bar, 50 μm. Representative of *n* = 3 biologically independent experiments with similar results. **F** Percentage of TUNEL-positive cells from (**E**). Each dot represents one biologically independent experiment (one independent culture well; *n* = 3); ≥5 random fields were quantified and averaged per experiment. **G** Immunoblot of DUSP26 and c-Cas3 in BUMPT cells stably infected with Lv-NC or Lv-OE and treated with cisplatin. **H** Quantification of c-Cas3 in (**G**) normalized to β-ACTIN. *n* = 4 biologically independent experiments. **I** TUNEL staining of Lv-NC and Lv-OE BUMPT cells treated with cisplatin. Scale bar, 50 μm. Images are representative of biologically independent experiments (*n* = 3) with similar results. **J** Percentage of TUNEL-positive cells from (**I**). *n* = 3 biologically independent experiments; ≥5 random fields were quantified and averaged per experiment. **K** Immunoblot of c-Cas3 in Lv-NC and Lv-OE BUMPT cells after H/R. **L** Quantification of c-Cas3 in (**K**) normalized to β-ACTIN. n = 4 biologically independent experiments. **M** Domain schematic of DUSP26 and phosphatase-inactive mutant (C152S). **N** Immunoblot of FLAG and c-Cas3 in BUMPT cells transfected with Ve, DUSP26-WT or DUSP26-C152S and treated with cisplatin. **O** Quantification of c-Cas3 in (**N**) normalized to β-ACTIN. *n* = 4 biologically independent experiments. Data are mean ± SEM. Statistical significance was assessed by unpaired Student's *t* test (two-sided) for (**B, D, H, L**) and one-way ANOVA with Tukey's multiple-comparisons test (two-sided; adjusted *P* values) for (**F, J, O**); exact *P* values are shown in the plots. Source data are provided as a Source Data file.

of WT p53 in p53-KO mice restored their susceptibility to cisplatin, leading to severe renal dysfunction (significantly elevated BUN and sCr), whereas the mice expressing the S312A mutant were strikingly protected (Fig. 9E). Histological analysis with H&E staining (Fig. 9F, G) and TUNEL staining (Fig. 9H, I) mirrored these functional data: the WT p53 rescue group exhibited extensive tubular injury and widespread apoptosis, whereas the S312A p53 group showed markedly reduced tubular damage and significantly fewer apoptotic cells. At the molecular level, the expression of kidney injury (NGAL) and apoptotic (cleaved caspase-3) markers were significantly increased in the WT p53 kidney, but this induction was largely blunted in the S312A group (Fig. 9J, K). Furthermore, the p53 target genes *Bax* and *Puma* were dramatically upregulated in the WT p53 kidneys, while their transcription was significantly attenuated in mice kidneys expressing the S312A p53 mutant (Fig. 9L). Taken together, these data provide direct evidence that the phosphorylation of p53 at Ser312 is required for its full pro-apoptotic transcriptional activity and the subsequent development of acute kidney injury (Fig. 10).

## Discussion

AKI is increasingly recognized as a complex clinical syndrome arising from a convergence of metabolic stress, inflammation, and regulated cell death programs[6,31–33]. Consistent with this multifactorial view, a recent genome-wide association study identified several genetic loci associated with AKI susceptibility[34], reinforcing the notion that diverse endogenous factors underlie AKI risk. Among these, cell death in kidney proximal tubules represents a pivotal pathological hallmark and determinant of AKI severity[35,36]. Landmark studies have demonstrated that pharmacological inhibition of p53 mitigates ischemic AKI by suppressing GTP depletion-induced apoptosis[36], while genetic ablation of p53 specifically in proximal tubules confers robust protection against both ischemic and nephrotoxic injury[37]. Notably, p53-driven pathways contribute to chronic kidney pathology as well; for example, a p53/miR-214/ULK1 cascade in diabetic nephropathy impairs protective autophagy in renal tubules, highlighting the broader impact of sustained p53 activation[38]. Despite this progress, the phosphatase systems that counterbalance stress-induced p53 activation remain poorly defined.

In this study, we identify DUSP26 as a previously unrecognized modulator of tubular stress responses. Integrating human AKI transcriptomic datasets, murine injury models, and CRISPR-based cellular systems, we demonstrate that DUSP26 exerts a cytoprotective role by selectively dephosphorylating p53 at serine 312. This residue corresponds to human Ser315, which has been previously implicated in modulating p53 protein stability and transcriptional activity in cancer biology[28,39]. Although the functional significance of Ser312 phosphorylation in murine models remains debated, with some studies suggesting it is dispensable for canonical tumor-suppressive roles[40,41], other evidence indicates that phosphorylation of murine p53 at Ser312 is necessary for full suppression of carcinogenesis in chemical injury models[41,42], highlighting that the impact of this site may be context-dependent. Our results establish the DUSP26-p53$^{S312}$ axis as a critical checkpoint that limits p53-driven apoptosis and preserves epithelial integrity.

Mechanistically, our findings establish the DUSP26–p53$^{S312}$ axis as a key modulatory node that restrains stress-induced apoptosis in the renal epithelium. The transcriptional activity of p53 is tightly regulated by site-specific phosphorylation events, with previous studies primarily focusing on canonical sites such as Ser15 and Ser20[43–45]. In contrast, we uncover a distinct mechanism in which DUSP26 selectively dephosphorylates p53 at Ser312—an evolutionarily conserved residue within the proline-rich domain—functioning as a context-dependent rheostat that fine-tunes p53's pro-apoptotic output under cell stress[30]. Using phospho-deficient (S312A) reconstitution in p53-null cells, we show that blockade of Ser312 phosphorylation leads to selective attenuation of *Bax* and *Puma* expression, decreased caspase-3 cleavage or apoptosis, and enhanced epithelial cell survival under injury conditions. Importantly, the pro-survival function of DUSP26 may extend beyond the kidney. As shown in our Supplementary Fig. 5, in a hepatic ischemia-reperfusion model, pharmacological inhibition of DUSP26 similarly potentiated epithelial damage, increased p53(S312) phosphorylation, and exacerbated apoptotic signaling (Supplementary Fig. 5), implicating this phosphatase as a conserved regulator of epithelial resilience across organs. This cross-tissue conservation supports the notion that stress-responsive phosphatases function as evolutionarily conserved checkpoints, enabling rapid and localized regulation of injury signals independent of global transcriptional responses[43,46].

Although the DUSP family has been extensively studied in inflammatory and oncogenic contexts, most renal investigations have centered on classical members such as DUSP1, DUSP2, and DUSP5, known for modulating MAPK signaling[16–18]. Notably, early work demonstrated that targeting DUSPs can broadly modulate MAPK cascades and immune responses[47]. Indeed, this classical MAPK-centric mechanism was recently suggested for DUSP26 as well[48]. DUSP26, in contrast, has largely been characterized in neuroblastoma and thyroid cancer[22,49], where it exhibits paradoxical effects by dephosphorylating p38 while potentiating ERK and JNK — underscoring the context-dependent nature of its substrate specificity[20,50]. Our study extends the functional repertoire of DUSP26 to renal epithelium and places it in direct opposition to pro-apoptotic p53 activation under stress. This finding, identified via our unbiased phosphoproteomic screen and validated with the p53-S312A mutant, highlights a non-canonical, MAPK-independent axis for DUSP26 in AKI. At the epigenetic level, we further identify hypermethylation of the DUSP26 promoter as a mechanism for its downregulation in AKI, aligning with growing evidence that injury-induced DNA methylation silences endogenous protective genes[51].

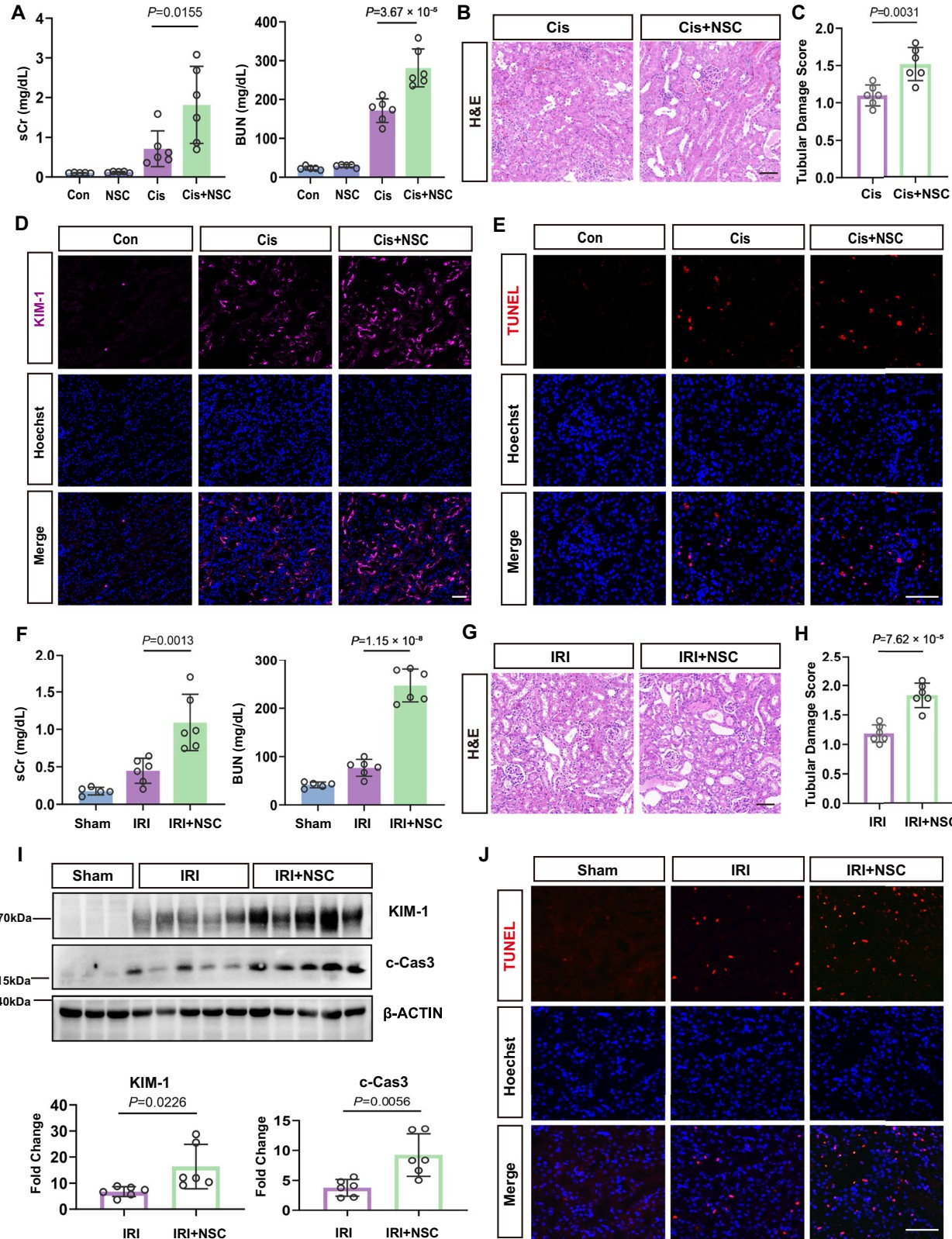

This insight offers a potential avenue for therapeutic intervention via demethylating agents or chromatin remodeling strategies.

It is critical, however, to contextualize the potential of such broad demethylating agents. Our data (Supplementary Fig. 8) confirmed that while the global DNA methyltransferase inhibitor 5-Aza successfully reversed *Dusp26* promoter methylation and partially reduced p-p53(S312) phosphorylation, it paradoxically failed to

suppress (and even increased) downstream cleaved caspase 3. This finding, while seemingly counterintuitive, is consistent with our previous work[52], where we demonstrated that 5-Aza exacerbates cisplatin-induced AKI. This phenomenon suggests 5-Aza acts as a "double-edged sword": while it restores DUSP26, it simultaneously activates other pro-apoptotic pathways via global, non-specific demethylation. This result strongly underscores the importance of

**Fig. 5 | Pharmacologic inhibition of DUSP26 sensitizes the kidney to AKI.**
**A** Serum creatinine (sCr) and blood urea nitrogen (BUN) at 48 h after cisplatin in control (Con), NSC87877 (NSC), cisplatin (Cis), or Cis+ NSC. Each dot represents one biologically independent mouse. **B** Representative H&E-stained kidney sections from the Cis and Cis+NSC groups. Scale bar, 50 μm. **C** Tubular injury score (0–4). Ten random fields were scored per mouse and averaged to yield one value per mouse. Each dot represents one biologically independent mouse ($n = 6$). **D** Representative immunofluorescence for kidney injury molecule-1 (KIM-1) in Con, Cis, and Cis+NSC kidneys. Scale bar, 50 μm. Images are representative of biologically independent mice (Con, $n = 5$; Cis, $n = 6$; Cis+NSC, $n = 6$) with similar results; two sections per mouse were examined. **E** Representative TUNEL staining in Con, Cis, and Cis+NSC kidneys. Scale bar, 50 μm. Images are representative of biologically independent mice (Con, $n = 5$; Cis, $n = 6$; Cis+NSC, $n = 6$) with similar results; two sections per mouse were examined. **F** sCr and BUN at 48 h after bilateral renal

ischemia/reperfusion injury (IRI) with vehicle or NSC pretreatment, or sham surgery. Each dot represents one biologically independent mouse. **G** Representative H&E-stained kidney sections from the IRI model. Scale bar, 50 μm. **H** Tubular injury score in the IRI model. Ten random fields were scored per mouse and averaged to yield one value per mouse. Each dot represents one biologically independent mouse ($n = 6$). **I** Immunoblot and quantification of KIM-1 and cleaved caspase-3 (c-Cas3) normalized to β-ACTIN. Each dot represents one biologically independent mouse. **J** Representative TUNEL staining in Sham, IRI, and IRI + NSC kidneys. Scale bar, 50 μm. Images are representative of biologically independent mice (Sham, $n = 5$; IRI, $n = 6$; IRI + NSC, $n = 6$) with similar results; two sections per mouse were examined. Data are mean ± SEM. Statistical significance was assessed by one-way ANOVA with Tukey's multiple-comparisons test (two-sided; adjusted $P$ values); exact $P$ values are shown in the plots. Source data are provided as a Source Data file.

developing targeted epigenetic interventions rather than relying on broad global inhibitors.

From a translational perspective, the DUSP26-p53[S312] signaling axis offers a potential therapeutic target for AKI intervention. Furthermore, advanced drug-delivery platforms such as cell-specific nanocarriers are being explored to selectively ferry therapeutics to injured renal cells[53,54], which could synergize with DUSP26-targeted strategies by enhancing precision and minimizing off-target effects. Unlike broad-spectrum kinase inhibition, which often perturbs essential homeostatic processes, modulating a phosphatase that selectively attenuates stress-induced apoptosis provides a conceptually refined approach[55]. The intrinsic challenge with phosphatase-based therapy lies in substrate promiscuity and context-dependent activity. Indeed, indiscriminate activation of DUSP26 may disrupt physiological signaling in non-injured tissues, underscoring the need for precision strategies that preserve its cytoprotective effects while minimizing off-target consequences. To this end, several complementary approaches may be envisaged. First, rational design of small molecules or stabilized peptides that mimic DUSP26's catalytic action at p53 Ser312 could confer substrate-level specificity and potentially bypass the need to modulate DUSP26 expression globally. Second, emerging protein–protein interaction enhancers may be leveraged to selectively stabilize the DUSP26–p53 complex, enhancing site-specific dephosphorylation without invoking broader phosphatome reprogramming[56]. Third, targeted epigenetic reactivation of DUSP26 via site-specific demethylating agents or CRISPR-dCas9-based epigenetic editors could restore endogenous phosphatase expression in a locus- and cell-type–specific manner. Given the conservation of the DUSP26–p53 axis across hepatic and renal epithelial tissues (Supplementary Fig. 5), these strategies may also be applicable to other acute organ injuries or failure.

Several limitations should be acknowledged. Although we demonstrate a clear interaction between DUSP26 and p53, additional substrates may contribute to its cytoprotective effects. High-resolution proteomic strategies, such as substrate-trapping mutants or proximity labeling platforms (e.g., APEX, BioID), could uncover the broader signaling landscape modulated by DUSP26[57]. Additionally, our human biopsy data are limited by cohort size and disease heterogeneity; although DUSP26 expression correlated inversely with ATN severity and positively with eGFR, these findings require further substantiation in larger, disease-specific validation studies. Moreover, while we validated our findings across murine and cell-based models, their extrapolation to human AKI requires further substantiation in patient-derived organoids or ex vivo perfused kidney systems. Lastly, our study primarily addressed acute injury responses; the role of DUSP26 in kidney repair, fibrosis, and maladaptive remodeling remains to be elucidated. This question is especially pertinent given that persistent p53 activation after AKI can drive mitochondrial dysfunction and maladaptive repair[58], suggesting that DUSP26 might influence long-term recovery and fibrosis.

In summary, our findings position DUSP26 as a key endogenous phosphatase that restrains stress-induced tubular apoptosis by modulating p53 Ser312 phosphorylation. This DUSP26-p53S312 axis represents not only a mechanistic advance in understanding AKI pathogenesis but also a potential therapeutic node amenable to precise intervention. Future strategies that restore or potentiate DUSP26 activity may form the basis for targeted therapies to ameliorate epithelial injury in AKI and potentially other acute organ syndromes.

## Methods
### Ethics statement
All human studies complied with relevant ethical regulations and were approved by the Institutional Ethics Committee of the Second Xiangya Hospital, Central South University (Approval/Protocol No. 2020YFC2005004). Written informed consent was obtained for the use and publication of de-identified clinical information; for individuals under 18 years of age, consent was obtained from a parent/legal guardian in accordance with institutional guidelines. No identifiable individual-level information is included. No compensation was provided to participants.

All animal procedures complied with relevant ethical regulations and the NIH Guide for the Care and Use of Laboratory Animals and were approved by the Institutional Animal Care and Use Committee (IACUC) of the Second Xiangya Hospital, Central South University (Approval No. 2022418). All mice were housed under specific-pathogen-free conditions in a temperature-controlled room (21–26 °C) with a 12 h dark/12 h light cycle and 40–60% humidity. Prior to tissue and blood collection, mice were euthanized by placement in a 5% isoflurane-saturated plexiglass chamber for 5 min, followed by cervical dislocation after confirming deep anesthesia by the absence of an active paw-withdrawal reflex.

### Sex as a biological variable
Human renal tissue specimens were obtained from male and female patients, and sex was abstracted from the clinical record. Sex-stratified analyses were not performed due to limited sample size and the use of archival specimens, which precluded adequately powered assessment of sex-specific effects. For all mouse experiments, only male mice were used to minimize potential confounding effects of estrogen on acute kidney injury susceptibility.

### Human kidney specimens and clinicopathological assessment
Human renal tissue specimens were obtained from patients diagnosed with acute tubular necrosis (ATN) by clinically indicated renal biopsy (ATN group, $n = 8$; male and female), together with de-identified clinicopathological information (Supplementary Table 1). Control renal tissue specimens ($n = 7$; male and female) were derived from the unaffected poles of nephrectomy samples from patients undergoing surgery for localized renal tumors. All analyses were performed on archival formalin-fixed, paraffin-embedded sections; no additional

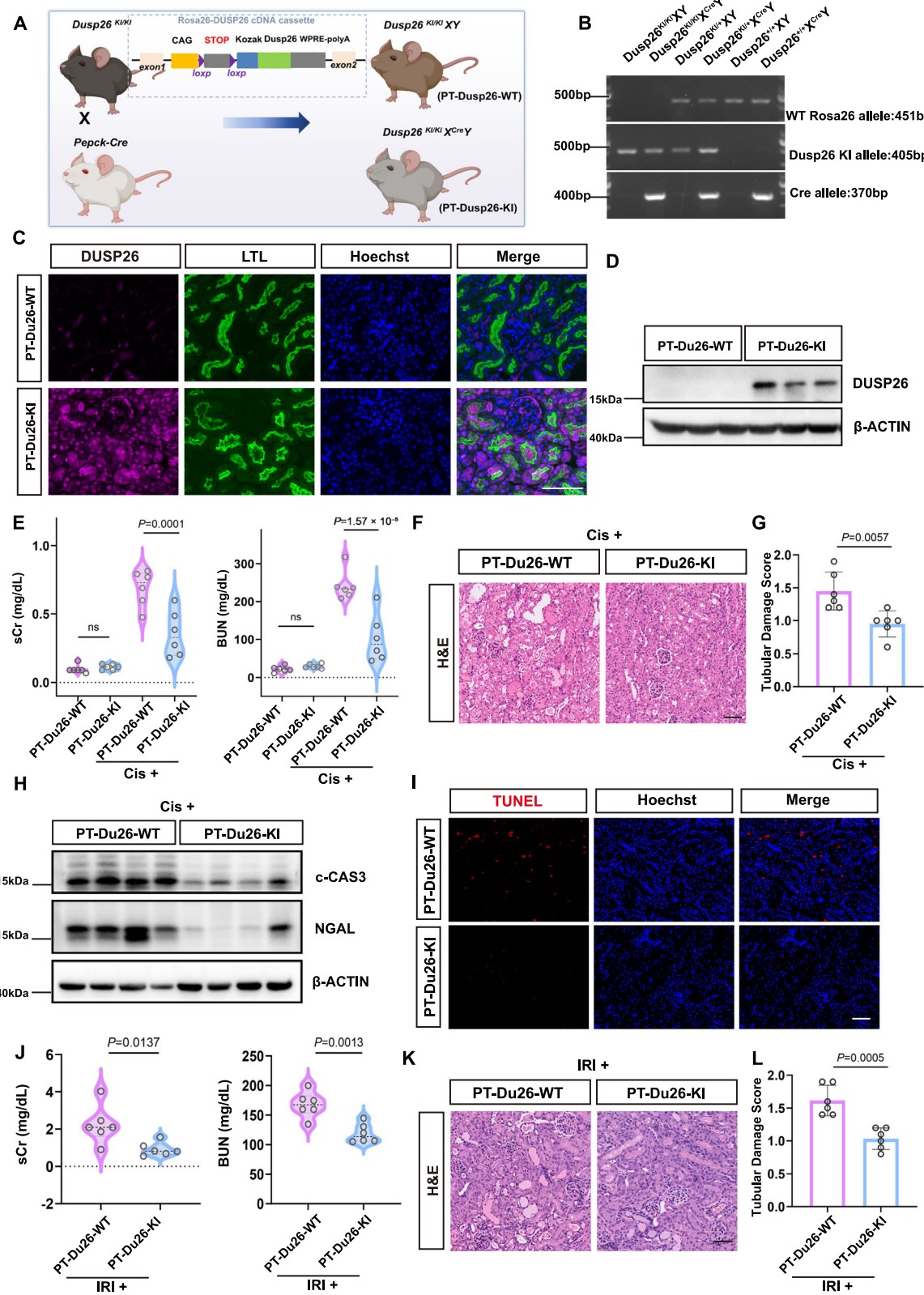

tissue was collected for research purposes beyond standard-of-care procedures, and no participants were prospectively recruited for this study. Because inclusion was based on clinical indication and tissue availability, selection/indication bias is possible (e.g., biopsy-confirmed ATN may not represent all AKI phenotypes). In addition, nephrectomy-adjacent "control" tissue may not fully represent a healthy kidney, which may limit generalizability.

To assess DUSP26 expression in proximal tubules, paraffin sections were subjected to immunofluorescence co-staining for DUSP26 and Lotus tetragonolobus lectin (LTL; a proximal tubule marker) as described below. Quantification was performed using Image-Pro Plus 6.0 by measuring DUSP26 integrated optical density (IOD) and normalizing to area (IOD/area, arbitrary units). Briefly, LTL-positive regions were identified on the LTL channel using a consistent

**Fig. 6 | Proximal tubule-specific knock-in of DUSP26 attenuates AKI. A** Breeding scheme to generate PT-Dusp26-KI (Rosa26^LSL-Dusp26; Pepck-Cre) and Cre-negative littermate controls (PT-Dusp26-WT). Created in BioRender. Fu, Y. (2026) https://BioRender.com/3k1zyxd. **B** Representative tail-DNA genotyping PCR showing WT Rosa26 (451 bp), KI (405 bp) and Cre (370 bp) bands. Genotyping PCR was repeated twice independently with similar results. **C** Representative immunofluorescence confirming proximal-tubule DUSP26 overexpression in PT-Dusp26-KI kidneys at baseline. Images are representative of biologically independent mice ($n = 6$ per genotype); two sections per mouse gave similar results. Scale bar, 50 μm. **D** Representative immunoblot confirming DUSP26 overexpression in PT-Dusp26-KI versus PT-Dusp26-WT kidneys (β-ACTIN). Blots are representative of biologically independent mice ($n = 6$ per genotype) with similar results. **E** Serum creatinine (sCr) and blood urea nitrogen (BUN) levels in PT-Dusp26-WT and PT-Dusp26-KI mice 24 h after cisplatin injection. **F** Representative H&E kidney sections after cisplatin. Scale bar, 50 μm. **G** Quantification of tubular damage scores based on H&E

staining in (**F**). **H** Immunoblot of cleaved caspase-3 (c-CAS3) and NGAL at 24 h post-cisplatin (β-ACTIN). Blots are representative of biologically independent mice ($n = 6$ per genotype) and were reproduced in two independent experiments with similar results. **I** Representative TUNEL staining at 24 h post-cisplatin. Images are representative of biologically independent mice ($n = 6$ per genotype); two sections per mouse gave similar results. Scale bar, 50 μm. **J** sCr and BUN after ischemia–reperfusion injury (IRI) in PT-Dusp26-WT and PT-Dusp26-KI mice; individual data points are shown ($n = 6$ per group). **K** Representative H&E kidney sections after IRI. Scale bar, 50 μm. **L** Quantification of tubular damage scores based on H&E staining in (**K**). For (**G, L**), bars indicate mean ± SEM. For (**E, J**), violin plots show median and interquartile range. Statistical significance was assessed by two-way ANOVA with Tukey's multiple-comparisons test (two-sided; adjusted $P$ values) for (**E**), and unpaired Student's $t$ test (two-sided) for (**G, J, L**); exact $P$ values are shown in the plots. Source data are provided as a Source Data file.

---

thresholding strategy across all samples to generate an LTL (+) mask. This mask was applied to the DUSP26 channel to quantify DUSP26 IOD/area within proximal tubules (LTL (+)). LTL-negative regions (LTL (−)) were defined as tissue areas outside the LTL (+) mask within the same field and quantified in an identical manner. For each specimen, values from five non-overlapping cortical fields were averaged to yield a single biological value per individual. Comparisons between Control and ATN groups were performed separately for LTL (−) and LTL (+) compartments using two-sided unpaired Student's t-tests. Correlations between LTL (+) tubular DUSP26 IOD/area and clinical/pathological indices (ATN score and eGFR) were assessed using two-sided Pearson correlation, with corresponding $P$ values reported.

## Mouse strains and sources
Male C57BL/6J mice were purchased from SJA Laboratory Animal Corporation (Hunan, China). Proximal tubule–specific DUSP26 knock-in mice (PT-Du26-KI; male) were generated by crossing conditional Rosa26-LSL-DUSP26 knock-in mice (Shanghai Model Organisms Center, Inc.) with Pepck-Cre transgenic mice kindly provided by Dr. Volker Haase (Vanderbilt University School of Medicine, Nashville, TN, USA). Male p53-knockout (p53-KO) mice on a C57BL/6J background (generated by Cyagen Bioscience) were used for in vivo p53 rescue experiments.

## Generation of proximal tubule-specific DUSP26 knock-in mice
Following previously established methods[52], we generated proximal tubular epithelial cell-specific DUSP26 knock-in (PT-Du26-KI) mice. Briefly, conditional DUSP26 knock-in mice, in which a loxP-STOP-loxP (LSL) cassette followed by the DUSP26 cDNA was inserted into the Rosa26 locus (designated as Dusp26^KI/KI), were obtained from Shanghai Model Organisms Center, Inc. These mice were crossed with Pepck-Cre transgenic mice, which express Cre recombinase under the control of the Pepck promoter, leading to Cre activity specifically in proximal tubular epithelial cells. Since the Pepck-Cre transgene is X-linked, male offspring with the genotype Dusp26^KI/KI; X^CreY were designated as PT-Du26-KI mice. Their Cre-negative male littermates, Dusp26^KI/KI; XY, served as wild-type controls (PT-Du26-WT). The breeding strategy is illustrated in Supplementary Fig. 1.

## AAV-mediated in vivo p53 rescue model
p53-knockout (p53-KO) mice (on a C57BL/6J background) were generated by deleting exon 3 through exon 9 using CRISPR/Cas9 technology, and were obtained from Cyagen Bioscience. Adeno-associated virus (AAV) vectors were custom-packaged by Genechem (Shanghai, China). The AAV vector backbone used was GV908, which employs the kidney tubular epithelial cell-specific Ksp-cadherin promoter (1.3k) to drive transgene expression. The following constructs were generated: AAV-p53-WT: Carrying the full-length murine *Trp53* cDNA; AAV-p53-S312A: Carrying the *Trp53* (NM_011640.4) cDNA with a site-directed

S312A mutation. The empty vector control (CON753). All vectors co-expressed EGFP via an FT2A element. Vectors were packaged as AAV9 serotype.

Male p53-KO mice were randomly allocated into three groups. Male mice were exclusively used in this study to minimize the potential confounding effects of estrogen, which is known to confer protection against AKI. For injection, the required volume of each virus was calculated to achieve the target dose of $8 \times 10^{11}$ vg/mouse. The calculated viral volume was then diluted with sterile PBS to a final injection volume of 100 μL (within the 80–120 μL range). Prior to injection, both the viral aliquots and the mice were pre-warmed. Mice received a single, slow tail vein injection of the corresponding AAV-p53-WT, AAV-p53-S312A, or AAV-EV. Twenty-one days post-injection, successful and specific AAV transduction in renal tubular epithelial cells was confirmed by EGFP fluorescence (Fig. 9C). To induce AKI, mice were administered a single intraperitoneal (i.p.) injection of cisplatin (30 mg/kg). Mice were euthanized 48 h after cisplatin administration, and blood and kidney tissues were harvested for subsequent analysis.

## Acute kidney injury (AKI) models
For bilateral renal ischemia–reperfusion injury (bIRI), 8–10-week old (22–25 g) mice were anesthetized with pentobarbital sodium (60 mg/kg, intraperitoneal), placed on a temperature-controlled surgical platform, and core temperature was maintained at 37 °C throughout the procedure. Following a midline laparotomy, both renal pedicles were exposed and occluded with microvascular clamps for 30 min. Reperfusion was initiated by clamp removal and confirmed visually by restoration of kidney color. Blood and kidneys were harvested 48 h after reperfusion unless otherwise specified. For cisplatin-induced AKI, mice received a single intraperitoneal injection of cisplatin (30 mg/kg) and were euthanized 48 h later; control mice received an equivalent volume of saline. For pharmacological intervention, the DUSP26 inhibitor NSC87877 was dissolved in PEG300/Tween-80/saline to a working concentration of 10 mg/mL. Mice received NSC87877 (50 mg/kg) or vehicle by intraperitoneal injection 1 h prior to bIRI surgery or cisplatin administration and then once daily thereafter. Serum creatinine and blood urea nitrogen (BUN) were measured using commercial kits (Roche Diagnostics, Germany) according to manufacturer instructions. Blood was collected via retro-orbital bleeding and centrifuged at $3000 \times g$ for 10 min to obtain serum.

## Hepatic ischemia–reperfusion injury (HIRI) model
HIRI was performed in anesthetized mice using intraperitoneal injection of pentobarbital sodium (60 mg/kg). Body temperature was maintained using a heating pad, and monitored via a rectal probe throughout the procedure. After a midline laparotomy and gentle exteriorization of the intestine, the portal triad branch supplying the left lateral liver lobe was occluded for 1.5 h using a microvascular clamp to induce regional hepatic ischemia, which was visually confirmed by

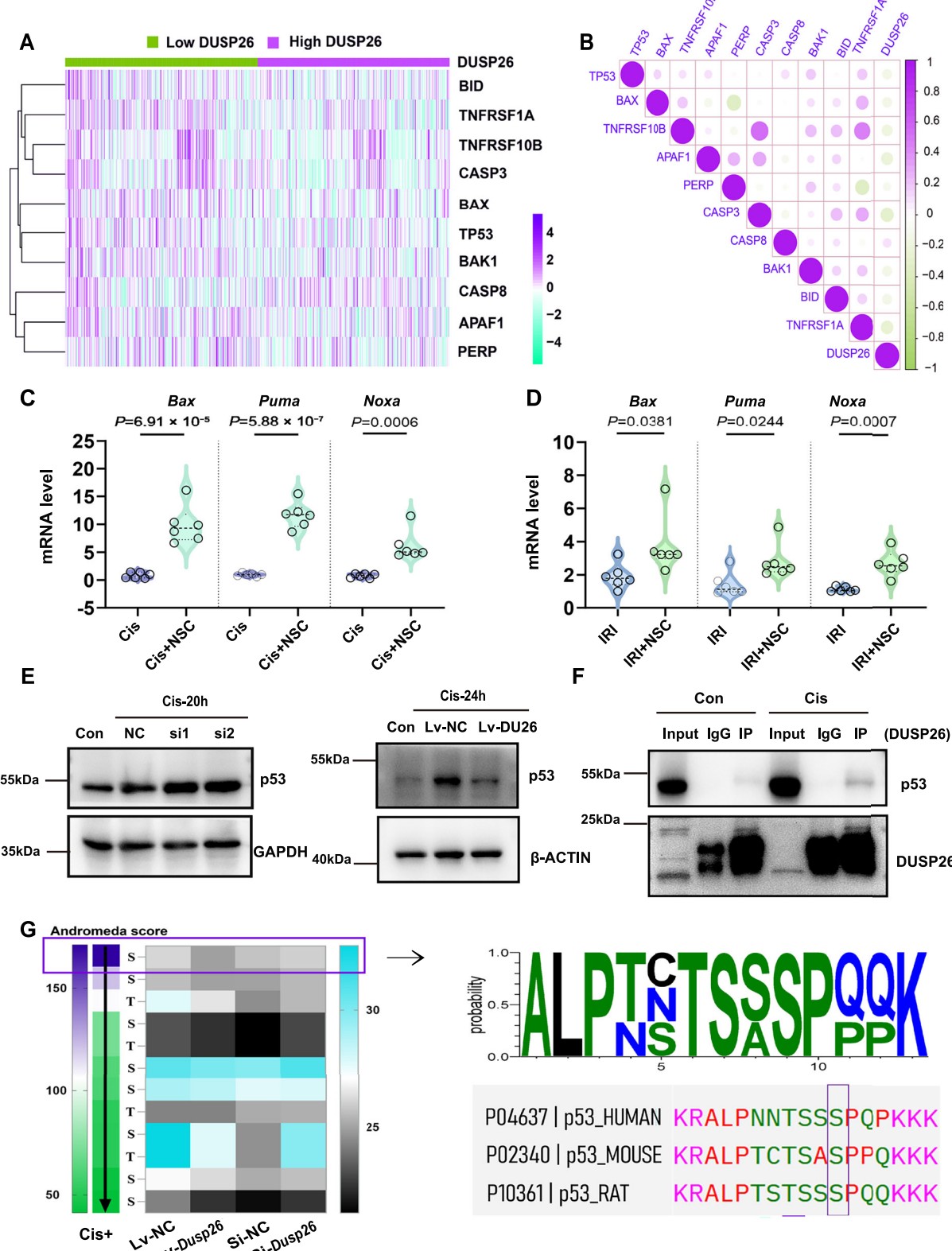

blanching of the liver. The exposed abdominal organs were kept moist using sterile saline-soaked gauze. Reperfusion was initiated by clamp removal. Mice were euthanized 24 h after reperfusion under deep anesthesia. Blood was collected for plasma ALT and AST assays, and liver tissues were harvested, rinsed in saline, and divided: one portion was snap-frozen and stored at −80 °C for molecular analyses, while the other was fixed and paraffin-embedded for histological studies.

Histological liver injury was assessed using the Suzuki criteria, based on scoring sinusoidal congestion, hepatocyte ballooning, and hepatocellular necrosis.

### Cell culture and in vitro injury models
Mouse proximal tubular epithelial cells (BUMPT; originally established by Dr. Lieberthal, Boston University) were maintained at 37 °C in a

**Fig. 7 | DUSP26 negatively regulates pro-apoptotic gene expression via interaction with p53. A** Heatmap analysis of apoptosis-related genes. Data were derived from three human ischemic AKI datasets from the Gene Expression Omnibus (GEO) database (GSE21374, GSE30718, GSE43974). Patients were stratified by median expression into high and low DUSP26 groups. **B** Co-expression analysis (Pearson correlation) between DUSP26 and indicated pro-apoptotic genes in the combined human AKI datasets. **C** Quantitative RT–PCR (qRT–PCR) of *Bax*, *Puma* and *Noxa* mRNAs in mice pretreated with vehicle or NSC87877 (NSC) followed by cisplatin. Dots, biologically independent mice (*n* = 6 per group). (**D**) qRT–PCR of *Bax, Puma* and *Noxa* mRNAs in mice pretreated with vehicle or NSC and subjected to ischemia–reperfusion injury (IRI; 30 min ischemia/48 h reperfusion). Dots, biologically independent mice (*n* = 6 per group). **E** Representative immunoblot of total p53 in BUMPT cells after cisplatin with DUSP26 knockdown (si-*Dusp26*) or overexpression (Lv-DU26) and matched controls. Representative of three biologically independent experiments with similar results (*n* = 3). **F** Endogenous co-immunoprecipitation in BUMPT cells treated with cisplatin (20 μM, 24 h); immunoprecipitation with anti-DUSP26 or IgG and immunoblotting for p53. Representative of 3 biologically independent experiments with similar results (*n* = 3). **G** Phosphoproteomics of BUMPT cells with DUSP26 overexpression or silencing and respective controls; p53 Ser312 was the top-ranked phosphorylation site and corresponds to human p53 Ser315. **C**, **D** Data are shown as violin plots. Center line indicates the median and dashed lines indicate the 25th and 75th percentiles. Statistical significance for (**C**, **D**) was assessed by unpaired Student's *t* test (two-sided); exact *P* values are shown in the plots. Source data are provided as a Source Data file.

humidified incubator with 5% $CO_2$ and cultured in Dulbecco's modified Eagle's medium (DMEM) supplemented with 10% fetal bovine serum (FBS) in standard tissue culture dishes. For cisplatin injury, BUMPT cells were treated with cisplatin (20 μM; Sigma-Aldrich) for 12, 24, or 36 h. For hypoxia/reoxygenation (H/R), cells were cultured in serum- and glucose-free DMEM under 1% $O_2$ for 8 h in a hypoxia chamber, followed by reoxygenation in complete DMEM under normoxia for 2, 4, 6, 8, or 10 h.

### siRNA knockdown and lentiviral overexpression
For transient silencing, BUMPT cells were transfected with *Dusp26*-targeting siRNAs (RiboBio, Guangzhou, China): si-m-*Dusp26*–1 (GTGGTCATAGTCTCACTGA) and si-m-*Dusp26*–2 (GGAAGCAGCT-GAGCTGTAA), using Lipofectamine RNAiMAX (Thermo Fisher Scientific) according to the manufacturer's instructions. A scrambled siRNA served as the negative control. For transfection, cells were seeded to reach ~70–80% confluence at the time of transfection, and siRNAs were used at a final concentration of 50 nM. Cells were incubated for 24 h prior to downstream treatments or analyses. For stable overexpression, BUMPT cells were transduced with lentivirus encoding mouse DUSP26 (pLenti-EF1-mCherry-P2A-Puro-CMV-DUSP26-WPRE; MOI = 40) or control lentivirus (pLenti-EF1-mCherry-P2A-Puro-CMV-MCS-WPRE). Media were replaced 12 h post-infection, and mCherry fluorescence was assessed at 72 h to confirm transduction efficiency. For selection of transduced cells, puromycin was applied at 2.0 μg/mL for 48–72 h as the primary selection condition.

### DUSP26 catalytic mutant and p53 rescue constructs in vitro
A catalytically inactive DUSP26 mutant (C152S) was generated by substituting cysteine 152 with serine. Plasmids encoding wild-type DUSP26 (pcDNA3.1(+)-DUSP26-3×FLAG; OBiO plasmid H37372) and DUSP26(C152S) (pcDNA3.1(+)-DUSP26(C152S)−3×FLAG; OBiO plasmid H37373), as well as the corresponding empty vector control, were constructed in the pcDNA3.1(+)-MCS-3×FLAG backbone (H6919). A CRISPR/Cas9-based p53-knockout (p53-KO) BUMPT cell line was generated using guide RNAs targeting mouse *Trp53* (GenBank NM_011640.4). sgRNA protospacer sequences (5′–3′) was PCA16734, ATAAGCCTGAAAATGTCTCC. The sgRNA/Cas9 expression vector used was GV465 (hU6-sgRNA–CBA promoter–3×FLAG-SpCas9–T2A–puro; GeneChem, Shanghai, China). Single-cell clones were expanded, and p53 depletion was confirmed by Western blot.

For rescue experiments, p53-KO cells were transfected with plasmids encoding wild-type p53 (pcDNA3.1(+)-Trp53-3×FLAG, OBiO, H38331; GenBank NM_011640.3), p53-S312A (pcDNA3.1(+)-Trp53(S312A)−3×FLAG, OBiO, H38332), or the matched empty vector (pcDNA3.1( + )-MCS-3×FLAG, OBiO, H6919) using Lipofectamine 3000 (Thermo Fisher Scientific) according to the manufacturer's instructions. The S312A substitution was introduced by site-directed mutagenesis, and the insert sequence was verified by Sanger sequencing performed by OBiO (sequencing primers: CMV-F, 5′-CGCAAATGGGCGGTAGGCGTG-3′; pcDNA-SEQR, 5′-TTATTAGGAAAAGACAGTGGG-3′).

### Tissue processing and hematoxylin and eosin (H&E) staining
Kidney tissues were fixed in 4% paraformaldehyde, dehydrated, paraffin-embedded, and sectioned at 4 μm. H&E staining was performed using standard procedures. Tubular injury was scored semi-quantitatively based on tubular dilation, brush border loss, tubular epithelial cell swelling/necrosis, and intraluminal debris or cast formation using the following scale: 0 (no injury), 1 (<25% of tubules injured), 2 (25–50%), 3 (50–75%), and 4 (>75%). Ten randomly selected cortical fields per animal were scored and averaged to yield a single value per animal. All scoring was performed by a pathologist blinded to the experimental groups.

### Immunofluorescence staining
Kidney sections were dewaxed, rehydrated, and subjected to antigen retrieval by immersion in citrate buffer (pH 6.0) followed by microwave heating using a stepped program (8 min at medium power, 8 min standby, and 7 min at medium–low power), then cooled to room temperature. Samples were blocked for 1 h at room temperature in blocking buffer 5% BSA in PBS with 0.1% Triton X-100, and incubated overnight at 4 °C with primary antibodies against DUSP26 (1:100; Invitrogen, PA5-22013), KIM-1 (1:200; R&D Systems, AF1817), and Lotus tetragonolobus lectin (LTL; 1:100; Vector Laboratories, RFL-1321). After washing, samples were incubated for 1 h at room temperature with species-appropriate secondary antibodies (Abcam, 1:400). Nuclei were counterstained with Hoechst 33342 (Beyotime, C1029) according to the manufacturer's instructions. Images were acquired on a Zeiss Axio Imager Z2 fluorescence microscope using ZEN (version 3.7; Carl Zeiss, Germany) acquisition software, with identical settings maintained across experimental groups within each experiment. Whole-slide images were acquired using CaseViewer (v2.4.0; 3DHISTECH, Hungary). Quantitative analysis, including fluorescence intensity and positive area percentage, was performed using Image-Pro Plus (v6.0; Media Cybernetics, USA). Specifically, the integrated optical density (IOD) was measured in five randomly selected high-power fields per section to ensure unbiased quantification.

### TUNEL assay
Apoptotic cells were detected using a TUNEL assay kit (Roche, Germany) according to the manufacturer's instructions. Sections were deparaffinized, rehydrated, permeabilized, and incubated with terminal deoxynucleotidyl transferase and fluorescein-labeled dUTP. Nuclei were counterstained with Hoechst 33342 (Beyotime, C1029). TUNEL-positive cells were quantified in 10 randomly selected fields per section.

### Immunoblotting
Total protein was extracted from tissue or cultured cells using RIPA lysis buffer supplemented with protease and phosphatase inhibitors (Sigma, P8340). Protein concentrations were determined using the BCA assay. Equal amounts of protein were separated by SDS-PAGE and transferred onto PVDF membranes. After blocking with 5% non-

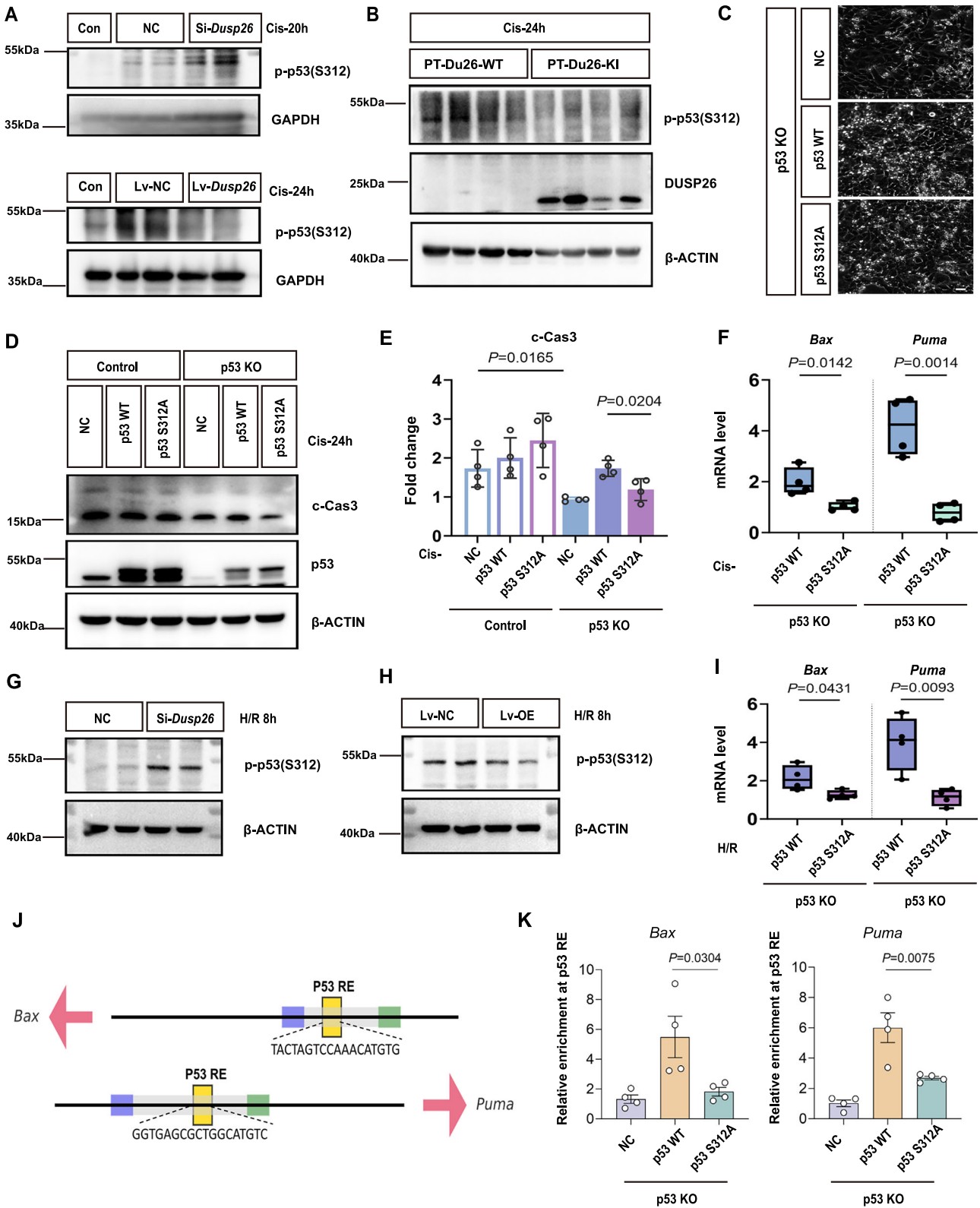

fat milk, membranes were incubated overnight at 4 °C with primary antibodies (all diluted at 1:1000) against DUSP26 (Invitrogen, PA5-22013), p53 (Cell Signaling Technology, 2524S), p-p53 (S15) (Cell Signaling Technology, 9284S), p-p53 (S20) (Cell Signaling Technology, 9287T), p-p53 (S46) (Proteintech, 28960), cleaved caspase-3 (Cell Signaling Technology, 9664S), NGAL (R&D Systems, AF1857), KIM-1 (R&D Systems, AF1817), GAPDH (Proteintech, 10494), β-actin (Proteintech, 66009), DNMT1 (Cell Signaling Technology, 5032),

DNMT3a (abcam, ab188470), DNMT3b (abcam, ab2851) and FLAG (Sigma-Aldrich, 102698217), as appropriate. After washing, membranes were incubated with HRP-conjugated secondary antibodies (1:5000; Proteintech). Chemiluminescent western blot signals were captured using Tanon Image Analysis Software (v4.2; Tanon, China). Quantification was performed using ImageJ (v1.54 g; NIH, USA). Uncropped and unprocessed scans of blots/gels are provided in the Source Data file.

**Fig. 8 | DUSP26 regulates tubular cell apoptosis in AKI by dephosphorylating p53 at serine 312. A** Representative immunoblot of phospho-p53 (Ser312) and total p53 in cells after cisplatin with DUSP26 knockdown or overexpression (n = 4 biologically independent experiments with similar results). **B** Representative immunoblot of phospho-p53 (Ser312) and total p53 in kidneys from proximal tubule-specific Dusp26 knock-in (PT-Dusp26-KI) and Cre-negative littermate control (PT-Dusp26-WT) mice after cisplatin (n = 6 biologically independent mice per group with similar results). **C** Representative bright-field images of p53 knockout (p53-KO) cells transfected with empty vector (NC), p53-WT, or p53-S312A and treated with cisplatin (n = 4 biologically independent experiments with similar results). Scale bar, 25 μm. **D** Representative immunoblot of p53 and cleaved caspase-3 (c-Cas3) in lysates from (**C**) (n = 4 biologically independent experiments with similar results). **E** Densitometry of c-Cas3 expression in (**D**). **F** Quantitative PCR (qPCR) of Bax and Puma in p53-WT or p53-S312A cells after cisplatin (n = 4 biologically independent

experiments; technical qPCR replicates were averaged within each biological replicate). Representative immunoblots of phospho-p53 (Ser312) and total p53 in DUSP26-silenced (**G**) or DUSP26-overexpressing (**H**) BUMPT cells after hypoxia/reoxygenation (H/R) (n = 4 biologically independent experiments with similar results). **I** qPCR of Bax and Puma in p53-WT or p53-S312A cells after H/R (n = 4 biologically independent experiments; technical qPCR replicates averaged within each biological replicate). **J** Schematic of p53 response elements in the Bax and Puma promoters. **K** ChIP–qPCR of p53 occupancy at Bax and Puma promoters in p53-KO cells reconstituted with p53-WT or p53-S312A, plotted as fold over IgG (n = 4 biologically independent experiments). Box-and-whisker plots (**F, I**): center line, median; box, 25th–75th percentiles; whiskers, min–max; points, biologically independent experiments. Data in (**E, K**) are mean ± SEM with individual data points overlaid. Statistical tests are indicated in the panels; all tests were two-sided and exact P values are shown. Source data are provided as a Source Data file.

### Quantitative real-time PCR (qPCR)

Total RNA was extracted using TRIzol reagent (Invitrogen, USA) and reverse-transcribed into cDNA using a PrimeScript RT reagent kit (Takara, Japan). qPCR was performed using SYBR Green Master Mix (Roche, Germany) on a LightCycler 96 Application Software (v1.1; Roche, Germany) and QuantStudio Real-Time PCR Software (v1.3; Applied Biosystems, USA). Gene expression was normalized to ACTIN, and relative expression levels were calculated using the $2^{-\Delta\Delta Ct}$ method. Primer sequences are listed in Supplementary Table 2.

### Chromatin immunoprecipitation (ChIP) and ChIP–qPCR

ChIP was performed using the High-Sensitivity ChIP Kit (ab185913; Abcam) according to the manufacturer's protocol with the following conditions. BUMPT cells (1 × 10⁶ per reaction) were cross-linked with 1% formaldehyde for 10 min at room temperature and quenched with 1.25 M glycine. Chromatin was sonicated (15 s on/30 s off, 20 cycles, 170–190 W) to yield DNA fragments of 100–700 bp (peak ~300 bp), verified by agarose gel electrophoresis. For each reaction, 2 μg chromatin was incubated in antibody-coated strip wells (100 μL total volume) containing antibody buffer, blocker, and enrichment enhancer. Antibodies were added on a per-well mass basis following the kit guidance (final amount 0.8 μg antibody per well for antibodies of interest; 0.8 μg per well for RNA polymerase II as the positive control; and 0.8 μg per well for non-immune IgG as the negative control).

For p53 ChIP, antibodies against p53 (Cell Signaling Technology, 2524) were used; normal mouse IgG served as negative control, and RNA polymerase II antibody provided in the kit served as positive control. After immunoprecipitation, wells were washed and DNA was released. Crosslinks were reversed with RNase A (10 mg/mL, 42 °C, 30 min) and Proteinase K (10 mg/mL, 60 °C, 45 min), followed by heating at 95 °C for 15 min. DNA was purified and eluted in 20 μL. ChIP DNA was quantified by qPCR using SYBR Green Master Mix (Applied Biosystems) on a QuantStudio 6 Flex system (Thermo Fisher Scientific). Primers targeting p53 response element–containing regions in Bax and Puma promoters are listed in Supplementary Table 2.

To assess DNMT occupancy at the Dusp26 promoter, BUMPT cells were treated with cisplatin (20 μM) for 24 h and subjected to ChIP–qPCR using anti-DNMT1 (CST, 5032), anti-DNMT3A (CST, 9768), anti-DNMT3B (CST, 44145), or species-matched IgG. The same primer set spanning the CpG-rich region of the Dusp26 promoter was used across DNMT ChIP assays (Supplementary Table 2). Fold enrichment over IgG was calculated as $2^{(Ct(IgG) - Ct(IP))}$.

### Methylation-specific PCR (MSP)

MSP primers for methylated (M) and unmethylated (U) alleles were designed using MethPrimer. Genomic DNA was isolated from cells after the indicated treatments and subjected to bisulfite conversion using the DNA Bisulfite Conversion Kit (DP215; TIANGEN, Beijing, China) according to the manufacturer's instructions, which converts

unmethylated cytosines to uracils while preserving methylated cytosines. Bisulfite-converted DNA was eluted in 20 μL elution buffer. MSP was performed using the MSP Kit (EM101; TIANGEN) in 20 μL reactions containing bisulfite-converted DNA (<500 ng), 1 μL each primer (10 μM), 1.6 μL dNTPs (2.5 mM), 1 U MSP DNA Polymerase, 2 μL 10× MSP PCR Buffer, and nuclease-free water to volume. Separate PCR reactions were set up for M-primers and U-primers using the same bisulfite-converted DNA template. Primer sequences were shown in Supplementary Table 2. PCR products (10 μL) were resolved by agarose gel electrophoresis and assessed for the presence of methylated or unmethylated amplicons.

### Targeted bisulfite sequencing (TBS)

Genomic DNA was extracted from mouse kidneys collected after ischemia–reperfusion injury (Control, n = 3; AKI, n = 3). For each sample, 100 ng genomic DNA was bisulfite converted using the EZ DNA Methylation-Gold Kit (Zymo Research) according to the manufacturer's instructions. Three bisulfite-PCR amplicons spanning the Dusp26 promoter (mm39; chr8: 31,579,529–31,579,791 (Dusp26_1, 263 bp, −25 bp to TSS), 31,579,670–31,579,930 (Dusp26_2, 261 bp, +116 bp), and 31,579,843–31,580,126 (Dusp26_3, 284 bp, +289 bp)) were amplified using ZymoTaq PreMix (Zymo Research) with the following primer pairs (Supplementary Table 2). Amplicons were purified and used for library construction with the VAHTS Universal Plus DNA Library Prep Kit (Vazyme), followed by paired-end sequencing on an Illumina NovaSeq 6000 using NovaSeq Control Software (Illumina; version 1.7.5). Raw reads were quality-filtered and adapter-trimmed using fastp v0.20.0, aligned to the mm39 reference genome, and CpG methylation calls were extracted using Bismark v0.22.3. CpG sites with ≥10× coverage were retained for downstream analysis. Differential methylation between groups was assessed using DMLtest (Wald test) implemented in the DSS (v2.43.2), and sites with p < 0.05 were considered significant.

### Generation and purification of phospho-specific anti–p-p53 (Ser312) antibody

A phospho-specific antibody recognizing p53 phosphorylated at Ser312 was generated by Hangzhou HuaAn Biotechnology Co., Ltd. using the synthetic phosphopeptide TSA (pSer)PPQKKKPLDC. The peptide was conjugated to KLH via Sulfo-SMCC crosslinking and used to immunize three New Zealand White rabbits (RB8154, RB8155 and RB8156) with a primary injection followed by boosters on days 14, 21, and 35 (Freund's complete adjuvant for priming and incomplete Freund's adjuvant for boosters). Serum collected after the final boost was evaluated by indirect ELISA against phosphorylated and non-phosphorylated peptides. Phospho-specific IgG was purified using a dual-affinity strategy: enrichment on a phosphopeptide column followed by depletion on a non-phosphorylated peptide column. Purified IgG was dialyzed into PBS (pH 7.4), sterile-filtered (0.22 μm), and

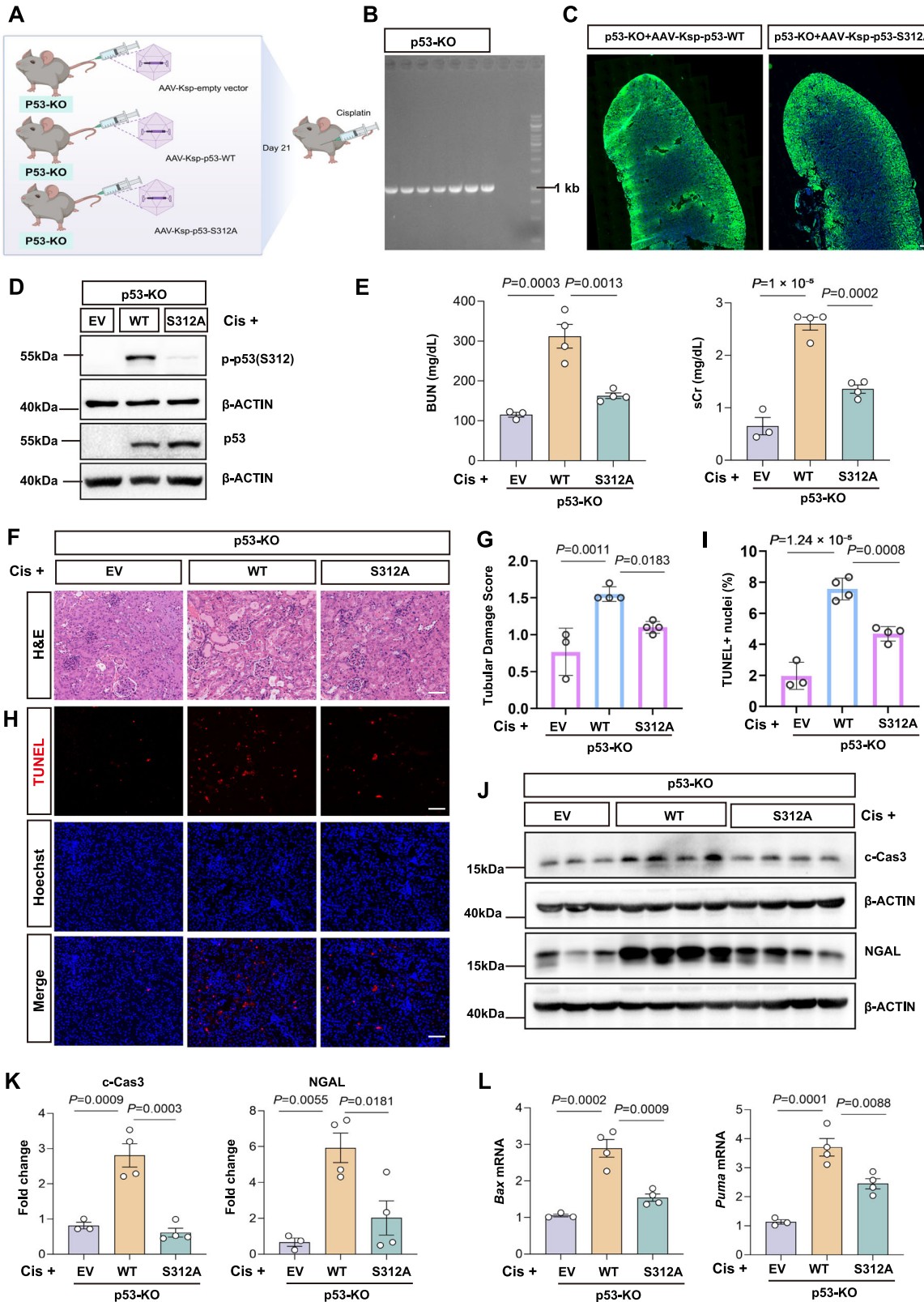

quantified by absorbance at 280 nm (DeNovix DS-11). The final concentration of the RB8154 preparation was 0.54 mg/mL.

## Mass spectrometry (proteomics) and data processing

To identify DUSP26-regulated phosphorylation events under cisplatin stress, BUMPT cells with DUSP26 knockdown or overexpression (and matched controls) were treated with cisplatin (20 µM) for 24 h.

For discovery analysis, cells from three biological replicates per group were pooled into one analytical sample, yielding four samples. Proteins were reduced with DTT and alkylated with iodoacetamide, digested with trypsin (Promega; enzyme-to-protein ratio 1:50) at 37 °C for 20 h, and peptides were desalted, lyophilized, and reconstituted in 0.1% formic acid. Peptides were analyzed on an Easy-nLC 1000 system coupled to a Q Exactive mass spectrometer (Thermo

**Fig. 9 | Phosphorylation of p53 at Ser312 is required for its pro-apoptotic function and AKI in vivo. A** Schematic of the genetic rescue model. p53 knockout (p53-KO) mice received tail-vein AAV-Ksp expressing empty vector (EV), p53-WT, or p53-S312A; cisplatin was administered at day 21 post-injection. Created in BioRender. Fu, Y. (2026) https://BioRender.com/5zdp2vt. **B** Representative PCR genotyping of tail DNA confirming p53-KO status ($n = 11$ mice); genotyping was repeated twice with similar results. **C** Representative whole-kidney fluorescence showing efficient AAV transduction at day 21 (EV, $n = 3$; p53-WT, $n = 4$; p53-S312A, $n = 4$ biologically independent mice); two sections per mouse. Scale bar, 500 μm. (**D**) Representative immunoblot of total p53 and phospho-p53(Ser312) in kidney lysates 48 h after cisplatin (EV, $n = 3$; p53-WT, $n = 4$; p53-S312A, $n = 4$ biologically independent mice); reproduced in two independent blots with similar results. β-ACTIN, loading control. (**E**) Blood urea nitrogen (BUN) and serum creatinine (sCr) 48 h after cisplatin. Each dot represents one biologically independent mouse (EV, $n = 3$; p53-

WT, $n = 4$; p53-S312A, $n = 4$). **F** Representative H&E-stained kidney sections. Scale bar, 50 μm. **G** Tubular damage score for (**F**). Each dot represents one mouse ($n$ as in **E**). **H** Representative TUNEL staining. Scale bar, 50 μm. **I** Percentage of TUNEL-positive nuclei. Each dot represents one mouse ($n$ as in **E**). (**J**) Representative immunoblot of cleaved caspase-3 (c-Cas3) and neutrophil gelatinase–associated lipocalin (NGAL) 48 h after cisplatin ($n$ as in **E**); reproduced in two independent blots with similar results. β-ACTIN, loading control. **K** Quantification of c-Cas3 and NGAL in (**J**), normalized to β-ACTIN and expressed relative to EV. Each dot represents one mouse ($n$ as in **E**). **L** Quantitative PCR (qPCR) of *Bax* and *Puma* mRNA, normalized to an internal control and expressed relative to EV. Each dot represents one mouse ($n$ as in **E**). Data are mean ± SEM with individual data points overlaid. Statistical significance was assessed by one-way ANOVA with Tukey's multiple-comparisons test (two-sided; adjusted $P$ values); exact $P$ values are shown in the plots. Source data are provided as a Source Data file.

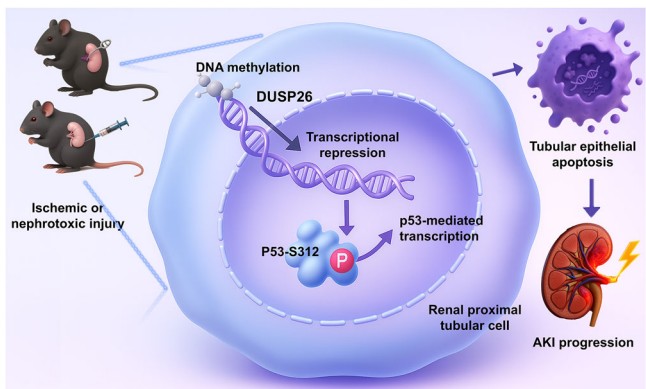

**Fig. 10 | Schematic model of DUSP26-mediated regulation of p53 signaling in AKI.** Following ischemic or nephrotoxic injury, DNA methylation occurs at the gene promoter of DUSP26, leading to transcriptional repression and reduced DUSP26 expression in kidney proximal tubular cells. The downregulation of DUSP26 impairs its ability to dephosphorylate p53 at serine 312, resulting in increased p53 phosphorylation and activation. This, in turn, enhances p53 transcriptional activity, promoting the expression of pro-apoptotic genes such as *Bax* and *Puma*. The upregulation of these genes accelerates tubular cell apoptosis and aggravates kidney injury. Restoration of DUSP26 expression, either genetically or pharmacologically, reduces p53$^{S312}$ phosphorylation, suppresses apoptotic signaling, and mitigates both toxic and ischemic AKI.

Fisher Scientific) operated in positive ion mode. MS1 scans were acquired over m/z 300–1800 at a resolution of 70,000 (at *m/z* 200) with an AGC target of $1 \times 10^6$ and a maximum injection time of 50 ms. The top 20 precursors were selected for HCD fragmentation with an isolation window of 2 m/z; MS2 spectra were acquired at a resolution of 17,500 with NCE 27 eV, dynamic exclusion of 30 s, and an underfill ratio of 0.1%. Raw files were processed using MaxQuant (v1.6.14) against the UniProt Mus musculus database (uniprot_Mus_musculus_87808_20240110). Carbamidomethylation (C) was set as a fixed modification; oxidation (M), phosphorylation (STY), GlyGly (K), acetylation (K), and protein N-terminal acetylation were set as variable modifications. Up to two missed cleavages were allowed. Peptide mass tolerance was 20 ppm and fragment mass tolerance was 0.1 Da. Protein- and peptide-level FDR were controlled at 1%. iBAQ was enabled. Phosphorylation sites with localization probability >0.75 were considered Class I sites.

### Bioinformatic and machine learning analysis (GEO)

We analyzed three GEO datasets (GSE21374, GSE30718, and GSE43974) derived from patients with early-stage AKI after kidney transplantation without evidence of rejection. In this context, the donor kidney

inevitably undergoes ischemia-reperfusion injury (IRI), making these datasets a valuable model for studying human IRI-induced AKI. Batch effects across datasets were corrected using the sva (v3.52.0) and limma (v3.60.0) packages in R (v4.4.0). Differentially expressed genes (DEGs) between AKI and control samples were identified with adjusted $P$ values < 0.05 and |log$_2$ fold change| > 1. To prioritize candidate genes, we applied two machine learning approaches: least absolute shrinkage and selection operator (LASSO) regression and support vector machine-recursive feature elimination (SVM-RFE). LASSO regression was performed using the glmnet (v4.1-8) package to identify variables with minimal binomial deviance. SVM-RFE was conducted using the e1071 (v1.7-14) and caret (v6.0-94) packages with 10-fold cross-validation. The diagnostic performance was assessed by receiver operating characteristic (ROC) curve analysis using the pROC (v1.18.5) package. Heatmaps and volcano plots were generated using the pheatmap (v1.0.12) and ggplot2 (v3.5.1) packages.

### Statistical analysis

Statistical analyses were performed using GraphPad Prism v10. Data are presented as mean ± SEM unless otherwise specified. Two-group comparisons used an unpaired two-tailed Student's $t$ test. One-way ANOVA with Tukey's post hoc test was used for single-factor designs with ≥3 groups. Two-way ANOVA with Sidak's or Tukey's post hoc test was used for two-factor designs as specified in figure legends. No outliers were excluded. Sample sizes were determined by experimental feasibility and prior experience in the field, and exact $n$ values are indicated in the figure legends and Source Data.

### Reporting summary

Further information on research design is available in the Nature Portfolio Reporting Summary linked to this article.

### Data availability

The transcriptomic datasets used in this study are available in the Gene Expression Omnibus (GEO) database under accession codes GSE21374, GSE30718 and GSE43974. The mass spectrometry proteomics data generated in this study have been deposited in the ProteomeXchange Consortium via the PRIDE partner repository under dataset identifier PXD070789. The bisulfite sequencing data generated in this study have been deposited in the NCBI BioProject database under accession code PRJNA1365351. All datasets generated or analyzed in this study will be available without access restrictions upon publication. Source data are provided with this paper.

### Code availability

No custom code was used in this study. All data analyses were performed using standard packages in R (version 4.4.0) and GraphPad Prism (version 10.0).

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

## Acknowledgements

This work was supported by the Postdoctoral Fellowship Program and China Postdoctoral Science Foundation (grant no. GZB20250487 to Y.F.), the Scientific Research Project of the Hunan Provincial Health Commission (grant no. 20254762 to Y.F.), the National Natural Science Foundation of China (grant nos. 82222013 and 82470731 to A.C.), the Key Program of the Natural Science Foundation of Hunan Province (grant no. 2025JJ30048 to A.C.), the National Key Clinical Specialty Scientific Research Project of the Second Xiangya Hospital, Central South University (grant no. 02201322 to A.C.), the U.S. Department of Veterans Affairs (grant nos. 1TK6BX005236 and I01BX000319 to Z.D.), and the U.S. National Institutes of Health (grant nos. 5R01DK058831 and 5R01DK087843 to Z.D.).

## Author contributions

Y.F. and Z.D. conceived the project and designed the experiments. Y.F. performed most of the experiments, analyzed the data, and wrote the manuscript. Y.X. and Y.H. contributed to the generation and analysis of the genetically modified mouse models. Y.F. and Y.X. performed bioinformatics analysis and supported statistical interpretation. J.C. and S.D. assisted with histological evaluation and clinical sample acquisition. A.C. and Z.D. supervised the entire project and revised the manuscript. All authors read and approved the final version of the manuscript.

## Competing interests

The authors declare no competing interests.
