## [Transparent Peer Review file · Nature Communications]

DUSP26 protects against acute kidney injury by dephosphorylating p53 at serine 312

Corresponding Author: Professor Zheng Dong

Version 0:

Reviewer comments:

Reviewer #1

(Remarks to the Author)

While the importance of phosphorylation events in AKI has been extensively studied, little is known about the role of dephosphorylation. DUSP6 is an atypical dual-specificity phosphatase. Although DUSP6 has been implicated in multiple diseases, the role of DUSP26 in the kidney remains unclear.

Using integrative bioinformatic analyses of publicly available AKI datasets, the authors identified DUSP26 as a potential important gene in AKI pathogenesis. Subsequently, they found that DUSP26 was highly expressed in proximal tubular cells in normal kidneys, but its expression was significantly reduced due to hypermethylation of the *Dusp26* promoter during AKI in both patients and mice.

Through pharmacological inhibition and proximal tubule-specific overexpression of DUSP26, the authors demonstrated that DUSP26 exerts a strong protective effect against AKI induced by cisplatin or I/R. Mechanistically, DUSP26 inactivated p53 by dephosphorylating it at S312, thereby suppressing apoptosis in renal tubular cells.

This study reveals a previously unrecognized and important role of DUSP26 in attenuating kidney injury, positioning it as a promising therapeutic target for AKI. The work is very well-designed and comprehensive, spanning from patient dataset analysis, phenotypic characterization of genetic mouse models, and mechanistic elucidation.

I have a few minor comments:

1. TUNEL positive tubular cells in Figure 5 may need to be quantified to assess apoptosis levels. Were there any alterations in cleaved caspase 3 in the kidneys following NSC treatment?
2. The phenotypes of the PT-specific DUSP26 overexpressing mice are significant. It would be helpful to confirm DUSP26 expression levels by Western blotting and immunofluorescence.
3. The findings on the role of promoter methylation in *Dusp26* silencing in renal tubular cells are exciting (Figure 3). In Figure 3E, treatment with 5-Aza, a DNA methyltransferase inhibitor, enhanced DUSP26 protein expression in DUMPT cells in the presence of cisplatin. It would be interesting to see whether p53 phosphorylation at S312 and cleaved casp3 were decreased following 5-Aza treatment.
4. The authors may consider moving the routine methods (e.g., cell culture and animal models of AKI) to Supplemental Materials.

Reviewer #2

(Remarks to the Author)

*Does your article contain western blots?

Yes - For manuscripts that include western blots, authors are required to provide the full length uncropped original western blots used in their manuscript, as part of their original submission. Full length western blots should be uploaded as a single 'Supplemental Material' file, should be clearly cited in the main text, and will be published if your article is accepted for publication.

In this manuscript, the authors demonstrate that DUSP26 directly binds and dephosphorylates p53 at Ser312, thereby

attenuating p53-mediated apoptosis and reducing tubular epithelial cell death. Through an integrated approach involving clinical kidney biopsies, murine models of AKI, epigenetic analysis, and mechanistic cell-based assays, this study suggests the DUSP26-p53 axis as a regulatory node in renal injury, and addresses a clinically relevant issue. However, several important concerns must be addressed before further consideration.

Major Concerns

1. Figure 2B, statistical comparisons are not provided? It is unclear whether the differences reach statistical significance. Figures 2D and 2H, the DUSP26 staining in the control group are overexposed, which affects interpretation. Additionally, the Hoechst nuclear staining is poorly visible, making it difficult to assess cellular localization. The overall immunofluorescence quality should be improved.
2. Figure 3, although the authors employed methylation-specific PCR (MSP) to show increased promoter methylation of Dusp26, this method is semi-quantitative and lacks resolution at the single-nucleotide level. You are suggested to perform bisulfite sequencing or MeDIP-qPCR to quantify methylation at specific CpG sites accurately. In addition, the expression levels of DNA methyltransferases (e.g., DNMT1, DNMT3a, DNMT3b) in AKI models should be examined to support the epigenetic regulation mechanism.
3. Figure 4A, the Western blot for DUSP26 is unclear and difficult to interpret. A higher-quality blot is needed.
4. Functional validation of the p53 S312A mutant remains insufficient. The authors transfected WT and S312A p53 constructs into p53-knockout BUMPT cells and examined downstream gene expression and apoptosis. However, the effect of p53 WT and S312A overexpression on cleaved caspase-3 (c-Cas3) in Figure 8E is mild, and it is unclear how statistical significance was determined. Please clarify the statistical method and number of replicates used. Important functional aspects, such as the stability, nuclear translocation, and coactivator interactions of the S312A mutant, are not addressed. You should determine whether the S312A mutation affects p53 binding to Bax or Puma promoters, for example through ChIP assays.
5. Figure 5E, the quality and reliability of the TUNEL staining are questionable. Representative images do not clearly show nuclear fragmentation, and the staining lacks specificity.
6. Figure 6D. In the PT-Dusp26-WT+Cisplatin group, renal injury appears relatively mild, raising concerns about the consistency of the model.
7. Figure 7I. The expression pattern of β -actin closely resembles that of KIM-1, suggesting possible issues with loading control or blot normalization.
8. Figure 8. The band intensity and pattern of total p53 and phospho-p53 (p-p53 S312) vary substantially between different panels, which affect data interpretation. Please ensure consistency and normalization across replicates.

Minor Concerns

1. Cisplatin exposure time is inconsistent throughout the study. In some experiments, the treatment duration is 24 hours, while 20 hours in others. Please explain the rationale for choosing different time points and ensure consistency across experiments or clarify if time-course variation is intentional.
2. Some figure legends (e.g., Figures 4 and 6) lack sufficient details to interpret results.
3. Please clarify your raw phosphoproteomic data will be publicly available.

Reviewer #3

(Remarks to the Author)

Thank you for inviting me to evaluate the work conducted by Fu et al., which presents a thorough investigation finding DUSP26 as a new phosphatase that safeguards proximal tubule cells from acute kidney injury by the dephosphorylation of p53 at serine 312. In this study, Fu et al. employed an integrative transcriptomic analysis, human biopsy staining, in vitro cell models, and various mouse injury models, demonstrating that DUSP26 expression is downregulated in AKI and that the loss of DUSP26 exacerbates tubular apoptosis, whereas DUSP26 overexpression confers protection against both CIS-AKI and IR-AKI. Mechanistically, they illustrate that DUSP26 interacts with p53 and specifically dephosphorylates Ser312, attenuating p53's pro-apoptotic transcriptional activity. The p53 S312A mutant notably replicates the protective effect of DUSP26 overexpression, while NSC87877 exacerbates harm in a liver ischemia model, indicating a more extensive organ-protective function.

The study examines a critical issue: how phosphatases mitigate stress-induced p53 in acute kidney injury, within a domain where p53 is extensively recognized as a principal facilitator of tubular cell apoptosis. The study is methodologically rigorous and comprehensive, encompassing human data, genetic models, CRISPR cell lines, biochemical assays (using a proprietary phospho-p53S312 antibody and phosphoproteomics), and two separate AKI models, hence providing robust support for the conclusions. The discovery that DUSP26 influences a non-canonical p53 site (Ser312) is unprecedented and may have significant ramifications for renal disease and cancer biology. Furthermore, the translational aspect of human tissue and liver injury is a strength, indicating that this pathway may be amenable to therapeutic targeting.

The work possesses significant scientific worth and importance to nephrology and cell death research. Nevertheless, essential clarifications and supplementary evidence are required to comprehensively substantiate the assertions (see to points below).

Major issues

1. The ethic number should be provided on line 96.
2. To further clarify the relationship between DUSP26 and kidney function, Figure 1 should include the correlation between DUSP26 and creatine, nitrogen, and eGFR.
3. Figure 3E illustrates the 20-hour treatment of BUMPT cells with the DUSP26 phosphatase inhibitor NSC87877 (5 μ M) and cisplatin (20 μ M). Figure 3i illustrates the TUNEL staining of DUSP26-overexpressing cells after 24 hours of cisplatin

treatment. The time point for tunel staining is distinct; how can this be elucidated?

4. The PT-Dusp26-KI model is crucial; however, the manuscript fails to demonstrate the baseline DUSP26 levels of these animals. The authors should quantify the expression of DUSP26 in KI proximal tubules compared to WT proximal tubules, for example, through immunostaining or Western blotting of DUSP26 in untreated kidneys.
5. A recent study conducted by Xiang et al. (2024) in the field of toxicology and applied pharmacology demonstrated that DUSP26 protects against renal IR injury by restricting TAK1-JNK/p38 activation. The present study identifies a unique mechanism (p53 S312 dephosphorylation) and does not address MAPKs. The authors should elaborate on the connections between these discoveries. For instance, is the DUSP26–p53 axis independent of DUSP26's impact on JNK/p38? The discussion should acknowledge that DUSP26 likely has multiple substrates and cite Xiang et al., as well as clarify how the p53 effect fits into this context.
6. The mechanistic focus is on Ser312, which is non-canonical in AKI contexts. The authors demonstrate that p53-driven apoptosis is diminished by preventing Ser312 phosphorylation (S312A). To ensure comprehensiveness, it is necessary to determine whether DUSP26 directly influences the stability of p53 or other p53 phosphorylation sites. The proteomics identified Ser312 as the most significant hit (Fig. 7G); however, it is uncertain whether canonical sites (e.g. Ser15, Ser20) were also examined. Additionally, it would be beneficial to establish a connection between the Ser312 modification and existing literature. For instance, prior mouse studies demonstrated that the phosphorylation of Ser312 compromises the function of the tumor suppressor p53, which is consistent with our discovery that its dephosphorylation enhances cell survival. The importance of Ser312 would be emphasized by including a concise citation to that effect.
7. The mechanistic insight and compelling observation that DUSP26 downregulation is mediated by promoter hypermethylation (Fig. 3) are noteworthy. Nevertheless, the authors' previous study, which was published in *Kidney International* (2017), indicated that the treatment with the DNA methyltransferase inhibitor 5-Aza actually exacerbated tubular cell mortality in cisplatin-induced AKI. This evident discrepancy is worthy of discussion. The authors should emphasize in the Discussion that 5-Aza may promote cell death because of its extensive epigenetic effects or potential off-target consequences, despite the fact that it restores DUSP26 expression in injured tubular cells.
8. The representative immunofluorescence image in Fig. 1H, which depicts DUSP26 expression in the control group, is somewhat less clear and well-defined than the corresponding AKI image. The authors may wish to select a higher-resolution image that more clearly illustrates tubular DUSP26 expression in control tissue in order to enhance visual consistency and clarity.

Additionally, there are a few minor issues that require attention.

1. In Fig. 6A (schematic of knock-in strategy), it would help to label the DUSP26 cDNA cassette in the Rosa26 locus for clarity.
2. All figure legends should specify n and statistical tests (most do). For example, Fig. 1H–J should note how many human samples were quantified (n=8,7 are in Supp Table). Fig. 8C–E (cell images and blots) should state the number of replicates.
3. In Figs. 7–9, ensure that asterisks (*) are defined (the legends mostly do). For clarity, abbreviations (e.g. AKI, IRI, HIRI) should be spelled out once in the legend.
4. Introduction: "Dual specificity phosphatase 26 (DUSP26) is an atypical MAPK phosphatase." Perhaps briefly define "atypical" (lacks ERK-binding motif) or cite a ref.
5. Methods: Indicate how ATN histology was scored. The "Suzuki criteria" for liver should be defined at first mention (though that's in legend).
6. Discussion: In the paragraph starting "Integrating human AKI transcriptomic datasets...", it may read better to split it into two shorter sentences for readability.
7. Check consistency: the paper uses "Dusp26" vs "DUSP26" (gene vs protein). Ensure italicization for the gene if referring to mouse Dusp26 gene vs protein.

Reviewer #4

(Remarks to the Author)

The manuscript proposes that DUSP26 protects against AKI by selectively dephosphorylating p53 at Ser312 (human Ser315) and that DUSP26 is downregulated via promoter hypermethylation. The conceptual space is narrow: the p53–AKI link is already extensively established, and DUSP26 has been reported as a p53 regulator in other cell types (e.g., neuroblastoma; PMID:20562916), including reports of Ser20/Ser37 dephosphorylation. Against this background, the evidence presented for a Ser312-focused mechanism and an epigenetic cause for DUSP26 loss is not yet sufficient.

1. The authors should more clearly distinguish a Ser312-centered pathway from the established N-terminal mechanisms (Ser20/Ser37) within the same renal/AKI models and conditions. As presented, it remains difficult to exclude that the observed effects are mediated primarily through the known N-terminal sites rather than Ser312.
2. There is no in-vivo evidence that Ser312 phosphorylation is required. Although renal p-p53(Ser312) is reduced in PT-Dusp26-KI mice after injury, this is correlative rather than a test of necessity. Genetic evidence—for example, a non-phosphorylatable p53-S312A knock-in would be needed to show requirement.
3. The epigenetic evidence remains preliminary. MSP is qualitative and does not resolve methylation at individual CpG sites. Please provide quantitative, locus-resolved methylation across the DUSP26 promoter, preferably using bisulfite sequencing, and relate these measurements to DUSP26 mRNA and protein in matched samples. If a causal role is proposed, include a functional rescue using demethylation by pharmacological agents or locus-targeted editing, and, where feasible, add chromatin evidence by demonstrating DNMT occupancy and relevant histone marks with ChIP.
4. The pharmacologic inhibitor NSC87877 is known to have off-target effects. Genetic approaches to modulate DUSP26 activity would help confirm specificity and reinforce the mechanistic conclusions.
5. Long-term outcomes after AKI, such as fibrosis and maladaptive repair, are not assessed. Given p53's known role in these

processes, examining later time points or discussing the potential implications would enhance the translational relevance. Although DUSP26 appears to suppress apoptosis and exert protective effects in the acute phase, persistent survival of abnormal cells may increase the risk of tumorigenesis or, conversely, inhibit proper tubular regeneration, potentially promoting the transition from AKI to CKD.

6. The human biopsy data are limited by small sample size and heterogeneous underlying diseases, which complicate interpretation of associations such as the inverse correlation between DUSP26 and ATN severity.

7. The liver ischemia–reperfusion injury extension is interesting, but it diverts focus from the kidney-centric narrative. Why was the liver selected for investigation? What is the expression profile of DUSP26 across tissues? Consider streamlining or moving some of these data to the Supplementary Information unless they are essential to the main conclusions.

Minor Comments:

1. Regarding the generation of proximal tubule-specific DUSP26 knock-in mice, the manuscript refers to “DUSP26 flox/flox” mice in the Method section.

2. In Figure 2B and 2C, the y-axis lacks clarification regarding the reference value used for normalization. Please specify what was set as “1” in the quantification.

Reviewer #5

(Remarks to the Author)

Version 1:

Reviewer comments:

Reviewer #1

(Remarks to the Author)

The authors have added substantial amounts of results to strengthen the manuscript, and my comments have been adequately addressed. The manuscript is now acceptable for publication in Nature Communications.

Reviewer #2

(Remarks to the Author)

The authors have addressed my concerns.

Reviewer #3

(Remarks to the Author)

The author has thoroughly addressed my previous concerns, and I have carefully reviewed the questions raised by other reviewers along with the author's responses. The entire revised manuscript and figure revisions have been scrutinized, and the paper now meets publication standards. I personally recommend acceptance.

Reviewer #4

(Remarks to the Author)

The authors have responded to my primary concerns regarding the DUSP26-p53 S312 axis with extensive new experimentation. The question of specificity for S312 dephosphorylation is resolved by new in vivo data from the PT-DUSP26-KI mice, which demonstrate a remarkably preferential effect on S312 over other N-terminal phosphorylation sites (Supplementary Fig. 7). Similarly, the question of whether S312 phosphorylation is required for AKI pathogenesis has been decisively addressed by the genetic rescue model utilizing AAV-driven p53-S312A re-expression in p53-knockout mice (Fig. 9), providing conclusive in vivo evidence of necessity. Furthermore, the conclusion that DUSP26's protective effect is dependent on its phosphatase activity is adequately supported by multiple genetic approaches, including siRNA knockdown and the phosphatase-dead C152S mutant (Fig. 4), appropriately reinforcing the pharmacological inhibitor findings. Acknowledging the long-term role of DUSP26 as a future research direction within the Discussion's limitations section is also a reasonable clarification of the study's scope.

However, a lapse in mechanistic rigor remains regarding the epigenetic regulation of DUSP26 expression. While quantitative Bisulfite Sequencing confirms the existence of hypermethylation and DNMT1 recruitment is shown by ChIP-qPCR, DNMT1 is primarily a maintenance methyltransferase. Given that DUSP26 promoter hypermethylation implies de novo methylation, the principal executors of this process are DNMT3A and DNMT3B. The mechanistic chain remains incomplete because there are no direct Chromatin Immunoprecipitation (ChIP) data demonstrating the recruitment of DNMT3A or DNMT3B to the DUSP26 promoter. Proving the direct involvement of these de novo methyltransferases is highly desirable to achieve full scientific rigor for the proposed epigenetic mechanism of gene silencing.

Reviewer #5

(Remarks to the Author)

Version 2:

Reviewer comments:

Reviewer #4

(Remarks to the Author)

The authors have fully addressed the suggestion, and I have no further comments.

Reviewer #5

(Remarks to the Author)

POINT-BY-POINT RESPONSE TO REVIEWERS' COMMENTS

Reviewer #1:

While the importance of phosphorylation events in AKI has been extensively studied, little is known about the role of dephosphorylation. DUSP6 is an atypical dual-specificity phosphatase. Although DUSP6 has been implicated in multiple diseases, the role of DUSP26 in the kidney remains unclear.

Using integrative bioinformatic analyses of publicly available AKI datasets, the authors identified DUSP26 as a potential important gene in AKI pathogenesis. Subsequently, they found that DUSP26 was highly expressed in proximal tubular cells in normal kidneys, but its expression was significantly reduced due to hypermethylation of the Dusp26 promoter during AKI in both patients and mice.

Through pharmacological inhibition and proximal tubule-specific overexpression of DUSP26, the authors demonstrated that DUSP26 exerts a strong protective effect against AKI induced by cisplatin or I/R. Mechanistically, DUSP26 inactivated p53 by dephosphorylating it at S312, thereby suppressing apoptosis in renal tubular cells.

This study reveals a previously unrecognized and important role of DUSP26 in attenuating kidney injury, positioning it as a promising therapeutic target for AKI. The work is very well-designed and comprehensive, spanning from patient dataset analysis, phenotypic characterization of genetic mouse models, and mechanistic elucidation.

We sincerely thank the reviewer for the thorough evaluation of our manuscript. We are particularly grateful for the reviewer's appreciation of the novelty and significance of this study. We are very encouraged that the reviewer found our study to be "very well-designed and comprehensive" and that it reveals a "previously unrecognized and important role of DUSP26 in attenuating kidney injury." We have carefully addressed the comments, which we believe have further improved the manuscript.

I have a few minor comments:

1. TUNEL positive tubular cells in Figure 5 may need to be quantified to assess

apoptosis levels. Were there any alterations in cleaved caspase 3 in the kidneys following NSC treatment?

We agree that quantifying the TUNEL-positive cells is crucial for rigorously assessing the level of apoptosis. As suggested, we have now quantified the TUNEL-positive tubular cells in the relevant experiments. The statistical analysis for the TUNEL data has been added to the revised manuscript in Supplementary Figure 4. Furthermore, we have performed Western blot analysis to examine c-Cas3 levels in the kidneys following NSC treatment and included the blot in the revised Figure 5I.

Supplementary Figure 4. Quantification of TUNEL-positive cells in DUSP26-inhibited AKI models.

(A) Quantification of the percentage of TUNEL-positive cells in kidney tissues from mice treated with cisplatin (Cis) or cisplatin plus the DUSP26 inhibitor NSC87877 (Cis+NSC).

(B) Quantification of the percentage of TUNEL-positive cells in kidney tissues from mice subjected to ischemia-reperfusion injury (IRI) or IRI plus NSC87877 (IRI+NSC).

Data are presented as mean \pm SEM (n = 6 per group). Statistical significance was determined using an unpaired, two-tailed Student's t-test. ***P < 0.001, ****P < 0.0001.

Figure 5. Pharmacologic inhibition of DUSP26 sensitizes the kidney to AKI.

(A) Mice were pretreated with the DUSP26 inhibitor NSC87877 (NSC, 50 mg/kg), followed by cisplatin (Cis) administration. Serum creatinine (sCr) and blood urea nitrogen (BUN) levels were assessed 48 h later. “Con” denotes vehicle-treated control mice.

(B) Representative H&E-stained kidney sections from control and NSC-treated cisplatin groups; scale bar, 50 μ m.

(C) Tubular injury was scored based on the extent of epithelial brush border loss, tubular cell swelling, necrosis, and intraluminal debris or cast formation as follows: 0 = no injury; 1 = <25% of tubules injured; 2 = 25 - 50%; 3 = 50 - 75%; 4 = >75%. Scores were averaged from 10 randomly selected fields per animal.

(D) Representative immunofluorescence images of Kidney Injury Molecule-1 (KIM-1) (magenta) expression in kidney tissues from the cisplatin model. Nuclei are stained with Hoechst (blue). Scale bar = 50 μ m.

(E) Representative TUNEL staining for apoptosis (red) in kidney tissues from the cisplatin model. Nuclei are stained with Hoechst (blue). Scale bar = 50 μ m.

(F) Mice were pretreated with NSC and subjected to either bilateral renal ischemia (30 min) followed by reperfusion for 48 h or sham surgery. Serum creatinine and BUN levels were measured.

(G) Representative H&E-staining in kidney sections from the ischemic AKI model. Scale bar = 50 μ m.

(H) Tubular injury score in the ischemic AKI model.

(I) Top: Representative Western blot analysis of KIM-1 and cleaved caspase-3 (c-Cas3) in renal cortex and outer medulla from the IRI model. Bottom: Densitometric quantification of KIM-1 and c-Cas3, normalized to β -ACTIN.

(J) Representative TUNEL staining in kidneys from the ischemic AKI model.

Data in all quantitative graphs are presented as mean \pm SEM (n = 6 per group). Statistical significance was determined using a one-way ANOVA with Tukey's post-hoc test (for A and F) or an unpaired, two-tailed Student's t-test (for C, H, and both graphs in I). *P < 0.05, **P < 0.01, ***P < 0.001, **** P<0.0001.

2. The phenotypes of the PT-specific DUSP26 overexpressing mice are significant. It would be helpful to confirm DUSP26 expression levels by Western blotting and immunofluorescence.

We appreciate the reviewer's comment to further validate the PT-specific DUSP26 overexpressing model. As suggested, we have now performed experiments to confirm the DUSP26 expression levels in our proximal tubule (PT)-specific DUSP26 overexpressing (PT-DUSP26-KI) mice and their wild-type (WT) littermates. Specifically, we have added co-immunofluorescence staining data for DUSP26 and the proximal tubule marker LTL, which is now included in the revised manuscript as Figure 6C. Furthermore, we have confirmed the elevated DUSP26 protein levels via Western blot analysis, and this result is now presented in Figure 6D.

Figure 6. Proximal tubule-specific knock-in of DUSP26 attenuates AKI.

(A) Conditional *Dusp26* overexpression mice (*Rosa26*loxP-STOP-loxP-*Dusp26*) were crossed with *Pepck-Cre* mice to generate *Rosa26*loxP-STOP-loxP-*Dusp26*; *Pepck-Cre* mice (designated as PT-*Dusp26*-KI), with Cre-negative littermates as controls (designated as PT-*Dusp26*-WT).

(B) Representative PCR genotyping of tail DNA showing bands for the Wild-type (451bp), KI allele (405bp), and Cre (370bp) alleles.

(C) Representative immunofluorescence staining confirming DUSP26 (magenta) overexpression in PT-*Dusp26*-KI kidneys at baseline. The proximal tubule marker LTL (Lotus Tetragonolobus Lectin) is shown in green, and nuclei are stained with Hoechst (blue). Scale bar = 50 μ m.

(D) Representative Western blot confirming DUSP26 protein overexpression in PT-*Dusp26*-KI kidney

lysates in comparison to WT.

(E) Serum creatinine (sCr) and blood urea nitrogen (BUN) levels in PT-Dusp26-WT and PT-Dusp26-KI mice 24 h after cisplatin injection.

(F) Representative H&E-staining in kidney sections from PT-Du26-KI mice and PT-Du26-WT mice after cisplatin treatment; scale bar, 50 μ m.

(G) Quantification of tubular damage scores based on H&E staining in (F).

(H) Immunoblot analysis of cleaved caspase-3 (c-Cas3) and NGAL in kidney tissues at 24 h after cisplatin administration.

(I) Representative TUNEL (Terminal deoxynucleotidyl transferase dUTP nick end labeling) staining for apoptosis (red) in kidney sections after cisplatin. Nuclei are stained with Hoechst (blue). Scale bar = 50 μ m.

(J) Serum creatinine and BUN levels in PT-Du26-KI and PT-Du26-WT mice following ischemia-reperfusion injury (IRI).

(K) Representative H&E staining of kidney tissues after IRI. Scale bar = 50 μ m.

(L) Quantification of tubular damage scores based on H&E staining in (K).

Data in quantitative graphs (E, G, J, L) are shown with all individual data points (n = 6 per group). Bars in (G, L) represent mean \pm SEM; data in (E, J) are shown as violin plots with median and interquartile range. Statistical significance for (E) was determined using a two-way ANOVA with Tukey's post-hoc test. Statistical significance for (G, J, and L) was determined using an unpaired, two-tailed Student's t-test. *P < 0.05, **P < 0.01, ***P < 0.001, ****P < 0.0001.

3. The findings on the role of promoter methylation in Dusp26 silencing in renal tubular cells are exciting (Figure 3). In Figure 3E, treatment with 5-Aza, a DNA methyltransferase inhibitor, enhanced DUSP26 protein expression in DUMPT cells in the presence of cisplatin. It would be interesting to see whether p53 phosphorylation at S312 and cleaved casp3 were decreased following 5-Aza treatment.

We thank the reviewer for this insightful suggestion. To address this point, we performed the requested experiment to detect p-p53(S312) and c-Cas3 levels in DUMPT cells following 5-Aza treatment in the presence of cisplatin. These new data have now been included in Supplementary Figure 8. The results show that 5-Aza treatment partially reduced the phosphorylation of p53 at S312, which is consistent with its role in upregulating DUSP26 expression (as seen in Figure 3E).

However, we observed that the level of cleaved caspase 3 remained elevated. This finding, while seemingly counterintuitive, is consistent with our previous work published in *Kidney International* (Guo C, Pei L, Xiao X, et al. DNA methylation protects against cisplatin-induced kidney injury by regulating specific genes, including interferon regulatory factor 8. *Kidney Int*, 2017, 92: 1194-1205). In that

study, we demonstrated that 5-Aza, as a global DNA methyltransferase inhibitor, can exacerbate cisplatin-induced kidney injury by activating other apoptosis-related genes (eg. IRF8) through demethylation. This explains why 5-Aza increased cleaved caspase 3 or apoptosis during cisplatin treatment in our current study. This result underscores the importance of developing targeted epigenetic interventions rather than relying on global methylation inhibitors.

Supplementary Figure 8. 5-Aza treatment paradoxically increases apoptosis despite reducing p-p53(S312) phosphorylation in cisplatin-injured cells.

(A) Representative Western blot analysis of phosphorylated p53 at Ser312 (p-p53(S312)), total p53, and cleaved caspase-3 (c-Cas3). BUMPT cells were treated with cisplatin (Cis) in the presence of vehicle (Ve) or the DNMT inhibitor 5-Aza-2'-deoxycytidine (Aza) for 24 hours. β-ACTIN served as a loading control.

(B) Quantification of Western blot band intensities from (A) for p-p53(S312), total p53, and c-Cas3. Data are normalized to β-ACTIN and expressed as fold change relative to the Cis+Ve group. Data in bar charts are shown with all individual data points (n = 4 per group). Bars represent mean ± SEM. Statistical significance was determined using an unpaired, two-tailed Student's t-test. *P < 0.05, ns (not significant).

4. The authors may consider moving the routine methods (e.g., cell culture and animal models of AKI) to Supplemental Materials.

As suggested, we have moved the detailed descriptions of routine methods, including those for cell culture and the animal models of AKI, to the Supplemental Materials section in the revised manuscript.

Reviewer #2:

In this manuscript, the authors demonstrate that DUSP26 directly binds and dephosphorylates p53 at Ser312, thereby attenuating p53-mediated apoptosis and

reducing tubular epithelial cell death. Through an integrated approach involving clinical kidney biopsies, murine models of AKI, epigenetic analysis, and mechanistic cell-based assays, this study suggests the DUSP26-p53 axis as a regulatory node in renal injury, and addresses a clinically relevant issue. However, several important concerns must be addressed before further consideration.

We thank the reviewer for recognizing the clinical relevance of our study and the integrated approach, which combines clinical biopsies, murine models, and mechanistic assays. We also thank the reviewer for the constructive criticism and suggestions. We have performed additional experiments to address them thoroughly as follows.

Major Concerns

1. Figure 2B, statistical comparisons are not provided? It is unclear whether the differences reach statistical significance. Figures 2D and 2H, the DUSP26 staining in the control group are overexposed, which affects interpretation. Additionally, the Hoechst nuclear staining is poorly visible, making it difficult to assess cellular localization. The overall immunofluorescence quality should be improved.

We thank the reviewer for this valuable feedback on Figure 2.

Figure 2B (and 2F): We have now added explicit statistical comparison lines (brackets) to the revised Figures to clearly indicate the comparisons between the control and experimental groups.

Figure 2D and 2H: We have processed the original images to optimize the display settings. Specifically, we adjusted the DUSP26 channel to reduce saturation in the controls and enhanced the Hoechst channel for better visibility. The revised Figure 2D and 2H now clearly show the cellular localization without overexposure.

Figure 2. DUSP26 is downregulated in cisplatin- and ischemia-induced AKI in mice.

(A) Representative Western blot (WB) showing DUSP26 protein expression in kidney cortex and outer medulla lysates from mice at 48 and 72 h after cisplatin (Cis) injection. “Con” represents control (saline-injected) mice.

(B) Densitometric quantification of DUSP26 protein from (A), normalized to β-ACTIN.

(C) Quantitative RT-PCR (qRT-PCR) analysis of *Dusp26* mRNA expression in kidney cortex and outer medulla at 48 and 72 h after cisplatin treatment.

(D) Representative immunofluorescence (IF) staining of kidney sections 48 h after cisplatin administration. Staining shows DUSP26 (magenta), proximal tubule marker LTL (Lotus Tetragonolobus Lectin, green), and nuclei (Hoechst, blue). Scale bar = 50 μm.

(E) Representative WB analysis of DUSP26 expression in the renal cortex and outer medulla from male C57BL/6 mice subjected to bilateral renal ischemia (bIRI) (30 min ischemia), followed by 48 or 72 h of reperfusion. “Sham” denotes sham-operated control mice.

(F) Densitometric quantification of DUSP26 protein expression from (E), normalized to β-ACTIN.

(G) qRT-PCR analysis of *Dusp26* mRNA expression in the renal cortex and outer medulla at 48 and 72 h after bIRI.

(H) Representative IF images showing co-localization of DUSP26 (magenta) and LTL (green) in kidneys from Sham and bIRI mice at 48 h post-reperfusion. Nuclei are stained with Hoechst (blue); Scale bar, 50 μ m.

Data in bar charts (B, C, F, G) are presented as mean \pm SEM (n = 6 per group). In these panels, data were normalized to the respective control group (“Con or “Sham”), which was set to a value of 1. Statistical significance was determined using a one-way ANOVA with Tukey’s post-hoc test. *P < 0.05, **P < 0.01, ***P < 0.001.

2. Figure 3, although the authors employed methylation-specific PCR (MSP) to show increased promoter methylation of Dusp26, this method is semi-quantitative and lacks resolution at the single-nucleotide level. You are suggested to perform bisulfite sequencing or MeDIP-qPCR to quantify methylation at specific CpG sites accurately. In addition, the expression levels of DNA methyltransferases (e.g., DNMT1, DNMT3a, DNMT3b) in AKI models should be examined to support the epigenetic regulation mechanism.

We thank the reviewer for this insightful comment and excellent suggestion to strengthen our epigenetic analysis. We agree that more quantitative and mechanistic data are crucial. We have addressed both points as follows:

1) Bisulfite Sequencing: As suggested, in addition to our original MSP results (Figure 3A - C), we have now performed bisulfite sequencing analysis (BS-seq) of the Dusp26 promoter region, which provides quantitative, single-CpG-level resolution. As shown in the new Figures 3G-I, we analyzed multiple CpG sites across three amplicons. The results show significantly higher methylation frequencies at specific CpG sites in AKI samples compared with controls (Figure 3I), which is further corroborated by heatmap visualization (Figure 3H). This provides robust, quantitative evidence of Dusp26 promoter hypermethylation.

2) DNMTs Mechanism: To support the epigenetic regulation mechanism, we examined the DNA methyltransferases. First, as requested, we quantified the expression of DNMTs. Our new data demonstrate that DNMT1, DNMT3a, and DNMT3b are all significantly upregulated in our cisplatin-induced AKI model (new Supplementary Figure 6). Second, to directly link this upregulation to the Dusp26 promoter, we also performed a ChIP-qPCR assay. As shown in the new Figure 3F, we observed a significant increase in the enrichment of DNMT1 at the Dusp26 promoter

in the cisplatin group.

Together, these new results provide robust, quantitative, and mechanistic epigenetic evidence (BS-seq, DNMT upregulation, and direct DNMT1 promoter binding) supporting our conclusion that *Dusp26* is silenced by promoter hypermethylation in AKI.

Figure 3. Promoter methylation mediates *DUSP26* downregulation in proximal tubular epithelial cells following injury.

(A) Visualization of CpG island distribution within the *Dusp26* promoter region using the MethPrimer database.

(B) Genomic DNA was subjected to bisulfite conversion, and methylation-specific (M) and unmethylated (U) primers targeting the CpG-rich region of the *Dusp26* promoter were designed for PCR amplification.

(C) Methylation-specific PCR (MSP) showing promoter methylation status of *DUSP26* in control, cisplatin, and cisplatin+5-Aza groups, as well as in control, H/R, and H/R+5-Aza groups.

(D) qRT-PCR time course of Dusp26 mRNA expression in BUMPT cells treated with cisplatin (20 μ M) in the presence of vehicle (Ve) or 5-Aza (1 μ M) for the indicated times (n=4).

(E) Representative Western blot showing DUSP26 protein expression in BUMPT cells treated with 5-Aza following cisplatin (24 h) or H/R (8 h) induction (n=4).

(F) Chromatin Immunoprecipitation (ChIP)-qPCR analysis showing the relative enrichment of DNMT1 at the Dusp26 promoter in BUMPT cells (Control vs. Cisplatin 24h).

(G) Bisulfite sequencing (BS-seq) analysis of three different amplicons (Dusp26_1, Dusp26_2, Dusp26_3) in the Dusp26 promoter from control (Con) and IRI-AKI mouse kidney tissue (n=3). Bars show the methylation percentage at individual CpG sites.

(H) Heatmap visualizing the methylation frequency of individual CpG sites within the Dusp26_1 amplicon from (G).

(I) Bar graphs showing the quantification of methylation percentage at specific, representative CpG sites from the BS-seq analysis in (G) and (H).

Data in bar charts (F, I) are presented as mean \pm SEM. Statistical significance was determined using a two-way ANOVA with Sidak's post-hoc test (for D) or an unpaired, two-tailed Student's t-test (for F and I), *P < 0.05, **P < 0.01.

Supplementary Figure 6. Cisplatin treatment induces the expression of DNA methyltransferases (DNMTs) in BUMPT cells.

(A) Representative Western blot analysis of DNMT1, DNMT3A, and DNMT3B expression in BUMPT cells treated with cisplatin (Cis-cells) for the indicated time points (12, 24, and 36 hours). Control (Con) cells were untreated. β -ACTIN was used as a loading control.

(B) Quantification of Western blot band intensities from (A) for DNMT1, DNMT3A, and DNMT3B. Data are normalized to β -ACTIN and expressed as fold change relative to the control (Con) group.

Data in bar charts are shown with all individual data points (n = 4 per group). Bars represent mean \pm SEM. Statistical significance was determined using one-way ANOVA with Tukey's post-hoc test. *P < 0.05, **P < 0.01.

3. Figure 4A, the Western blot for DUSP26 is unclear and difficult to interpret. A higher-quality blot is needed.

As requested, we have now replaced the original image with a new, higher-quality representative Western blot in the revised Figure 4A. This new blot clearly illustrates the knockdown efficiency of DUSP26 siRNA (si-DUSP26).

Figure 4. DUSP26 functions as a protective factor against acute injury in proximal tubular epithelial cells.

(A) Representative immunoblot of DUSP26 and cleaved caspase-3 (c-Cas3) in BUMPT cells. Cells were transfected with negative control (NC) siRNA or DUSP26-targeting siRNA (si-Dusp26) and then treated with cisplatin (20 μ M) for 20 h.

(B) Densitometric quantification of c-Cas3 from (A), normalized to GAPDH. (n = 4 per group).

(C) Immunoblot of c-Cas3 in BUMPT cells (transfected with NC or si-Dusp26) subjected to 8 h of hypoxia/reoxygenation (H/R).

(D) Densitometric quantification of c-Cas3 from (C), normalized to β -ACTIN. (n = 4 per group).

(E) Representative TUNEL (red) and Hoechst (blue) staining. BUMPT cells were pre-treated with the DUSP26 inhibitor NSC87877 (5 μ M) or vehicle (Ve, DMSO) before cisplatin (20 μ M) treatment for 20 h. "Con" represents untreated control cells. Scale bar = 50 μ m.

(F) Quantification of the percentage of TUNEL-positive cells from (E).

(G) Immunoblot of DUSP26 and c-Cas3 in BUMPT cells stably infected with control lentivirus

(Lv-NC) or DUSP26-overexpressing lentivirus (Lv-OE) (MOI=40). Cells were treated with cisplatin (20 μ M) for 24 h.

(H) Densitometric quantification of c-Cas3 from (G), normalized to β -ACTIN. (n = 4 per group).

(I) Representative TUNEL (red) and Hoechst (blue) staining of Lv-NC and Lv-OE BUMPT cells treated with cisplatin (20 μ M) for 24 h. Scale bar = 50 μ m.

(J) Quantification of the percentage of TUNEL-positive cells from (I).

(K) Immunoblot of c-Cas3 in stable Lv-NC and Lv-OE BUMPT cells subjected to 8 h H/R.

(L) Densitometric quantification of c-Cas3 from (K), normalized to β -ACTIN. (n = 4 per group).

(M) The full-length DUSP26 cDNA (GenBank: AY902194) encodes a 211-amino acid protein containing a 131-amino acid dual-specificity phosphatase (DSP) catalytic domain. This domain includes the conserved DSP motif "VHCAVGVSRS" and the MKP motif "AYLM".

(N) Immunoblot of FLAG (confirming expression) and c-Cas3. BUMPT cells were transfected with empty vector (Ve), wild-type DUSP26 (WT), or the phosphatase-inactive C152S mutant (C152). "Con" represents untransfected cells. Cells were treated with cisplatin (20 μ M) for 20 h.

(O) Densitometric quantification of c-Cas3 from (N), normalized to β -ACTIN. (n = 4 per group).

Data in all graphs are presented as mean \pm SEM. Statistical significance was determined using an unpaired, two-tailed Student's t-test (for B, D, H, L) or one-way ANOVA with Tukey's post-hoc test (for F, J, O). *P < 0.05, P < 0.01.

4. Functional validation of the p53 S312A mutant remains insufficient. The authors transfected WT and S312A p53 constructs into p53-knockout BUMPT cells and examined downstream gene expression and apoptosis. However, the effect of p53 WT and S312A overexpression on cleaved caspase-3 (c-Cas3) in Figure 8E is mild, and it is unclear how statistical significance was determined. Please clarify the statistical method and number of replicates used. Important functional aspects, such as the stability, nuclear translocation, and coactivator interactions of the S312A mutant, are not addressed. You should determine whether the S312A mutation affects p53 binding to Bax or Puma promoters, for example through ChIP assays.

We thank the reviewer for this comment regarding the functional validation of the p53 S312A mutant. We have addressed each concern as follows:

1) Clarification of Statistics (Figure 8E): We apologize for the lack of clarity. As noted in the original figure legend, these experiments were performed with n = 4 biological replicates. The statistical analysis used was a one-way ANOVA followed by Tukey's post-hoc test. We have now included the information in the revised figure legend.

2) We agree with the reviewer's observation that the change in c-Cas3 (Figure 8E) appears modest. We believe this is due to two factors: (a) These experiments are conducted in a p53-knockout (KO) background, where the baseline level of apoptosis

induced by cisplatin is already significantly reduced. (b) The activation of c-Cas3 is regulated by multiple signaling pathways, not solely by p53. Therefore, the S312A mutation, which affects only one specific regulatory site on p53, is expected to result in a partial, rather than a complete, suppression of the apoptotic phenotype.

3) The reviewer raises excellent points about other functional aspects, such as stability and translocation. We agree that the most critical functional test, as suggested by the reviewer, is to determine if the S312A mutation affects p53's DNA-binding activity at target promoters. To address this, we have now performed the suggested ChIP-qPCR assays in p53-KO cells reconstituted with either p53 WT or the S312A mutant. We targeted the p53 response elements (REs) within the Bax and Puma promoters (schematic shown in new Figure 8J). The new data, presented in Figure 8K, clearly show that while wild-type p53 exhibited robust enrichment at both promoters, this enrichment was markedly reduced in the S312A mutants. This result provides direct mechanistic evidence that phosphorylation at Ser312 (the site targeted by DUSP26) is critical for enhancing p53 recruitment to its pro-apoptotic target genes, thus validating the functional impact of the DUSP26-p53(S312) axis. These data, particularly the ChIP-qPCR analysis, provide the necessary functional validation suggested by the reviewer.

Figure 8. DUSP26 regulates tubular cell apoptosis in AKI by dephosphorylating p53 at serine 312.

(A) Cisplatin-induced p53 phosphorylation at Ser312 in DUSP26 knockdown or overexpression BUMPT cells. n = 4 per group.

(B) p53 Ser312 phosphorylation during cisplatin AKI in PT-Du26-KI and PT-Du26-WT mice. n = 6 per group.

(C) p53-WT or S312A mutant plasmids were transfected into p53 knockout (p53-KO) BUMPT cells, followed by cisplatin treatment. Bright-field images of the cells were then captured.

(D) p53-WT or S312A mutant plasmids were transfected into p53-KO BUMPT cells, which were then treated with cisplatin to collect lysate for immunoblot analysis of p53 and cleaved caspase-3 (c-Cas3). n = 4 per group.

(E) Densitometry of c-Cas3 expression in (D).

(F) qPCR analysis of Bax and Puma in p53-WT or S312A cells following cisplatin treatment.

(G) Immunoblot of Ser312 phosphorylated p53 in DUSP26 knockdown BUMPT cells after 8 h of hypoxia/reoxygenation (H/R). n = 4 per group.

(H) Immunoblot of Ser312 phosphorylated p53 in DUSP26-overexpressing BUMPT cells after 8 h of H/R. n = 4 per group.

(I) qPCR analysis of Bax and Puma in p53-WT or S312A cells after H/R treatment.

(J) Schematic representation of p53 response elements (REs) located within the Bax and Puma promoters. The canonical p53-binding motifs are indicated (yellow boxes), with sequences shown below each element.

(K) ChIP-qPCR analysis showing p53 enrichment at Bax and Puma promoters in p53 knockout (p53 KO) cells reconstituted with wild-type (p53 WT) or phosphorylation-deficient (p53 S312A) constructs. p53 occupancy was quantified relative to IgG control and normalized to input DNA (n = 4).

Please note that Western blot panels (e.g., A, B, D, G, H) represent distinct experiments with different models, treatments, or time points. All quantitative comparisons are made within a single panel relative to its respective internal controls. Data are presented as mean \pm SEM. *P < 0.05, **P < 0.01 by one-way ANOVA with Tukey's post hoc test.

5. Figure 5E, the quality and reliability of the TUNEL staining are questionable. Representative images do not clearly show nuclear fragmentation, and the staining lacks specificity.

For Figure 5E TUNEL staining, we have replaced with new, higher-magnification representative images in the revised manuscript. The new image more clearly demonstrates the co-localization of the apoptotic signal (TUNEL, red) with the nuclear stain (Hoechst, blue), confirming the assay's nuclear specificity. We agree with the reviewer's observation regarding the lack of overt nuclear fragmentation. In this regard, it is noteworthy that the TUNEL assay detects DNA strand breaks, a key early apoptotic event that does not always coincide with the later morphological stage of nuclear fragmentation.

Figure 5. Pharmacologic inhibition of DUSP26 sensitizes the kidney to AKI.

(A) Mice were pretreated with the DUSP26 inhibitor NSC87877 (NSC, 50 mg/kg), followed by cisplatin (Cis) administration. Serum creatinine (sCr) and blood urea nitrogen (BUN) levels were assessed 48 h later. “Con” denotes vehicle-treated control mice.

(B) Representative H&E-stained kidney sections from control and NSC-treated cisplatin groups; scale bar, 50 μ m.

(C) Tubular injury was scored based on the extent of epithelial brush border loss, tubular cell swelling, necrosis, and intraluminal debris or cast formation as follows: 0 = no injury; 1 = <25% of tubules injured; 2 = 25–50%; 3 = 50–75%; 4 = >75%. Scores were averaged from 10 randomly selected fields per animal.

(D) Representative immunofluorescence images of Kidney Injury Molecule-1 (KIM-1) (magenta) expression in kidney tissues from the cisplatin model. Nuclei are stained with Hoechst (blue). Scale bar

= 50 μ m.

(E) Representative TUNEL staining for apoptosis (red) in kidney tissues from the cisplatin model. Nuclei are stained with Hoechst (blue). Scale bar = 50 μ m.

(F) Mice were pretreated with NSC and subjected to either bilateral renal ischemia (30 min) followed by reperfusion for 48 h or sham surgery. Serum creatinine and BUN levels were measured.

(G) Representative H&E-staining in kidney sections from the ischemic AKI model. Scale bar = 50 μ m

(H) Tubular injury score in the ischemic AKI model.

(I) Top: Representative Western blot analysis of KIM-1 and cleaved caspase-3 (c-Cas3) in renal cortex and outer medulla from the IRI model. Bottom: Densitometric quantification of KIM-1 and c-Cas3, normalized to β -ACTIN.

(J) Representative TUNEL staining in kidneys from the ischemic AKI model.

Data in all quantitative graphs are presented as mean \pm SEM (n = 6 per group). Statistical significance was determined using a one-way ANOVA with Tukey's post-hoc test (for A and F) or an unpaired, two-tailed Student's t-test (for C, H, and both graphs in I). *P < 0.05, **P < 0.01, ***P < 0.001, ****P < 0.0001.

6. Figure 6D. In the PT-Dusp26-WT+Cisplatin group, renal injury appears relatively mild, raising concerns about the consistency of the model.

We thank the reviewer for this important observation. We agree that in the specific representative H&E image for the PT-Dusp26-WT + cisplatin group (shown in the new Figure 6F), the renal injury might have appeared relatively mild.

We would like to emphasize that the cisplatin-induced injury model itself was consistent and reproducible across all experiments. This is supported by our quantitative data for renal function (sCr/BUN, Figure 6E) and, most directly, by the quantitative tubular damage scores (Figure 6G), which were assessed from multiple animals and fields.

The perceived variation in that single representative image was due to mild, expected inter-individual differences. To better reflect the typical phenotype observed in this cohort, we have replaced the representative H&E image in Figure 6F with another section from the same group that more clearly illustrates the injury pattern. We hope this revision resolves the reviewer's concern about model consistency.

Figure 6. Proximal tubule-specific knock-in of DUSP26 attenuates AKI.

(A) Conditional *Dusp26* overexpression mice (*Rosa26loxP-STOP-loxP-Dusp26*) were crossed with *Pepck-Cre* mice to generate *Rosa26loxP-STOP-loxP-Dusp26; Pepck-Cre* mice (designated as PT-Dusp26-KI), with Cre-negative littermates as controls (designated as PT-Dusp26-WT).

(B) Representative PCR genotyping of tail DNA showing bands for the Wild-type (451bp), KI allele (405bp), and Cre (370bp) alleles.

(C) Representative immunofluorescence staining confirming DUSP26 (magenta) overexpression in PT-Dusp26-KI kidneys at baseline. The proximal tubule marker LTL (*Lotus Tetragonolobus Lectin*) is shown in green, and nuclei are stained with Hoechst (blue). Scale bar = 50 μ m.

(D) Representative Western blot confirming DUSP26 protein overexpression in PT-Dusp26-KI kidney lysates at baseline, normalized to β -ACTIN.

(E) Serum creatinine (sCr) and blood urea nitrogen (BUN) levels in PT-Dusp26-WT and

PT-Dusp26-KI mice 24 h after cisplatin injection.

(F) Representative H&E-staining in kidney sections from PT-Du26-KI mice and PT-Du26-WT mice after cisplatin treatment; scale bar, 50 μ m.

(G) Quantification of tubular damage scores based on H&E staining in (F).

(H) Immunoblot analysis of cleaved caspase-3 (c-Cas3) and NGAL in kidney tissues at 24 h after cisplatin administration.

(I) Representative TUNEL (Terminal deoxynucleotidyl transferase dUTP nick end labeling) staining for apoptosis (red) in kidney sections after cisplatin. Nuclei are stained with Hoechst (blue). Scale bar = 50 μ m.

(J) Serum creatinine and BUN levels in PT-Du26-KI and PT-Du26-WT mice following ischemia-reperfusion injury (IRI).

(K) Representative H&E staining of kidney tissues after IRI. Scale bar = 50 μ m.

(L) Quantification of tubular damage scores based on H&E staining in (K).

Data in quantitative graphs (E, G, J, L) are shown with all individual data points (n = 6 per group). Bars in (G, L) represent mean \pm SEM; data in (E, J) are shown as violin plots with median and interquartile range. Statistical significance for (E) was determined using a two-way ANOVA with Tukey's post-hoc test. Statistical significance for (G, J, and L) was determined using an unpaired, two-tailed Student's t-test. *P < 0.05, **P < 0.01, ***P < 0.001, **** P < 0.0001.

7. Figure 7I. The expression pattern of β -actin closely resembles that of KIM-1, suggesting possible issues with loading control or blot normalization.

We agree that the band pattern in the previous image was not ideal and could lead to misinterpretation. To address this, we have re-run the samples from this experiment and replaced the image in the revised Figure 5I with a new representative Western blot. For quantification, the densitometry value for KIM-1 in each lane was normalized to the densitometry value of its corresponding β -actin loading control from the same lane.

Figure 5. Pharmacologic inhibition of DUSP26 sensitizes the kidney to AKI.

(A) Mice were pretreated with the DUSP26 inhibitor NSC87877 (NSC, 50 mg/kg), followed by cisplatin (Cis) administration. Serum creatinine (sCr) and blood urea nitrogen (BUN) levels were assessed 48 h later. “Con” denotes vehicle-treated control mice.

(B) Representative H&E-stained kidney sections from control and NSC-treated cisplatin groups; scale bar, 50 μ m.

(C) Tubular injury was scored based on the extent of epithelial brush border loss, tubular cell swelling, necrosis, and intraluminal debris or cast formation as follows: 0 = no injury; 1 = <25% of tubules injured; 2 = 25–50%; 3 = 50–75%; 4 = >75%. Scores were averaged from 10 randomly selected fields per animal.

(D) Representative immunofluorescence images of Kidney Injury Molecule-1 (KIM-1) (magenta) expression in kidney tissues from the cisplatin model. Nuclei are stained with Hoechst (blue). Scale bar = 50 μ m.

(E) Representative TUNEL staining for apoptosis (red) in kidney tissues from the cisplatin model. Nuclei are stained with Hoechst (blue). Scale bar = 50 μ m.

(F) Mice were pretreated with NSC and subjected to either bilateral renal ischemia (30 min) followed by reperfusion for 48 h or sham surgery. Serum creatinine and BUN levels were measured.

(G) Representative H&E-staining in kidney sections from the ischemic AKI model. Scale bar = 50 μ m

(H) Tubular injury score in the ischemic AKI model.

(I) Top: Representative Western blot analysis of KIM-1 and cleaved caspase-3 (c-Cas3) in renal cortex and outer medulla from the IRI model. Bottom: Densitometric quantification of KIM-1 and c-Cas3, normalized to β -ACTIN.

(J) Representative TUNEL staining in kidneys from the ischemic AKI model.

Data in all quantitative graphs are presented as mean \pm SEM (n = 6 per group). Statistical significance was determined using a one-way ANOVA with Tukey's post-hoc test (for A and F) or an unpaired, two-tailed Student's t-test (for C, H, and both graphs in I). *P < 0.05, **P < 0.01, ***P < 0.001, ****P < 0.0001.

8. Figure 8. The band intensity and pattern of total p53 and phospho-p53 (p-p53 S312) vary substantially between different panels, which affect data interpretation. Please ensure consistency and normalization across replicates.

We thank the reviewer for this observation and agree that the absolute band intensities for p53 and p-p53(S312) appear to vary across the different panels in Figure 8.

This variation is not surprising, as these panels represent separate and distinct experiments conducted under different models, conditions, and time points. They are not intended for direct inter-panel comparison. Specifically: Fig 8A (top): si-Dusp26 in cells (Cisplatin 20h); Fig 8A (bottom): Lv-Dusp26 in cells (Cisplatin 24h); Fig 8B: PT-DUSP26-KI mouse tissue; Fig 8G: si-Dusp26 in cells (H/R 8h); Fig 8H: Lv-Dusp26 in cells (H/R 8h).

To ensure the validity of these intra-panel comparisons, all samples within a single experiment were processed in parallel, and all Western blots for that experiment were run on the same membrane. For quantification, the p-p53(S312) signal in each lane was normalized to its respective loading control (β -ACTIN/GAPDH) from the same lane. Therefore, the variations between different panels do not affect the interpretation or reliability of our conclusions, which are drawn only from the relative changes within each panel. To avoid any future misunderstanding, we have added a note to the revised figure legend to explicitly state this.

Figure 8. DUSP26 regulates tubular cell apoptosis in AKI by dephosphorylating p53 at serine 312.

(A) Cisplatin-induced p53 phosphorylation at Ser312 in DUSP26 knockdown or overexpression BUMPT cells. n = 4 per group.

(B) p53 Ser312 phosphorylation during cisplatin AKI in PT-Du26-KI and PT-Du26-WT mice. n = 6 per group.

(C) p53-WT or S312A mutant plasmids were transfected into p53 knockout (p53-KO) BUMPT cells, followed by cisplatin treatment. Bright-field images of the cells were then captured.

(D) p53-WT or S312A mutant plasmids were transfected into p53-KO BUMPT cells, which were then treated with cisplatin to collect lysate for immunoblot analysis of p53 and cleaved caspase-3 (c-Cas3). n = 4 per group.

(E) Densitometry of c-Cas3 expression in (D).

(F) qPCR analysis of Bax and Puma in p53-WT or S312A cells following cisplatin treatment.

(G) Immunoblot of Ser312 phosphorylated p53 in DUSP26 knockdown BUMPT cells after 8 h of hypoxia/reoxygenation (H/R). n = 4 per group.

(H) Immunoblot of Ser312 phosphorylated p53 in DUSP26-overexpressing BUMPT cells after 8 h of

H/R. n = 4 per group.

(I) qPCR analysis of Bax and Puma in p53-WT or S312A cells after H/R treatment.

(J) Schematic representation of p53 response elements (REs) located within the Bax and Puma promoters. The canonical p53-binding motifs are indicated (yellow boxes), with sequences shown below each element.

(K) CHIP-qPCR analysis showing p53 enrichment at Bax and Puma promoters in p53 knockout (p53 KO) cells reconstituted with wild-type (p53 WT) or phosphorylation-deficient (p53 S312A) constructs. p53 occupancy was quantified relative to IgG control and normalized to input DNA (n = 4).

Please note that Western blot panels (e.g., A, B, D, G, H) represent distinct experiments with different models, treatments, or time points. All quantitative comparisons are made within a single panel relative to its respective internal controls. Data are presented as mean \pm SEM. *P < 0.05, **P < 0.01 by one-way ANOVA with Tukey's post hoc test.

Minor Concerns

1. Cisplatin exposure time is inconsistent throughout the study. In some experiments, the treatment duration is 24 hours, while 20 hours in others. Please explain the rationale for choosing different time points and ensure consistency across experiments or clarify if time-course variation is intentional.

We thank the reviewer for this sharp observation and the request for clarification. The reviewer is correct that different time points were used, and this was an intentional experimental design choice to optimally capture the distinct biological questions being asked (i.e., loss-of-function vs. gain-of-function).

For loss-of-function (e.g., si-Dusp26 knockdown or NSC inhibitor treatment) study, we used a 20-hour cisplatin treatment. This relatively earlier time point represents a phase where the injury is significant but not yet maximal. Our rationale was that this provides a more sensitive window to observe an exacerbation of injury. If we waited until 24h, the severe, saturated damage might mask the additional injury caused by DUSP26 inhibition.

For gain-of-function (e.g., DUSP26 overexpression), we used a 24-hour cisplatin treatment. At this later time point, the cisplatin-induced damage is robust and highly evident. This provides a clear and necessary baseline against which the protective effects of DUSP26 overexpression can be best observed and quantified.

2. Some figure legends (e.g., Figures 4 and 6) lack sufficient details to interpret results.

We thank the reviewer for this important feedback. We have now thoroughly revised the legends for Figure 4, Figure 6, and all other figures in the manuscript to ensure all essential information is included directly within the legend.

Specifically, the revised legends now clearly state: 1) Definitions for all abbreviations used (e.g., BUMPT, H/R, c-Cas3, NC, OE, Ve). 2) Specific experimental parameters for each panel (e.g., drug concentrations and precise treatment/induction times). 3) The exact biological replicate number (n) for each individual panel's quantification. 4) A clear description of the statistical tests used for each comparison (e.g., Student's t-test or one-way ANOVA with post-hoc test) and how data are presented. We believe these changes make the results much clearer and easier to interpret.

3. Please clarify your raw phosphoproteomic data will be publicly available

We thank the reviewer for this point. We confirm that all raw phosphoproteomic data, including the raw files and result files, have been successfully deposited to the ProteomeXchange Consortium via the PRIDE partner repository with the identifier# PXD070789.

The data are currently private pending publication. To facilitate review, the following temporary credentials can be used to access the dataset directly:

Username: reviewer_pxd070789@ebi.ac.uk Password: kqm6gfuklzZT

We have also added the official data availability statement to the "Data Availability" section of the manuscript.

*Does your article contain western blots?

Yes - For manuscripts that include western blots, authors are required to provide the full length uncropped original western blots used in their manuscript, as part of their original submission. Full length western blots should be uploaded as a single 'Supplemental Material' file, should be clearly cited in the main text, and will be published if your article is accepted for publication.

We have complied with the journal's policy. A supplementary file containing all full-length, uncropped original Western blots for every blot presented in the manuscript (main and supplementary figures) has been uploaded.

This file is titled “supporting data”, and each blot within this file is clearly labeled to correspond to its respective figure in the manuscript.

Reviewer #3 (Remarks to the Author):

Thank you for inviting me to evaluate the work conducted by Fu et al., which presents a thorough investigation finding DUSP26 as a new phosphatase that safeguards proximal tubule cells from acute kidney injury by the dephosphorylation of p53 at serine 312. In this study, Fu et al. employed an integrative transcriptomic analysis, human biopsy staining, in vitro cell models, and various mouse injury models, demonstrating that DUSP26 expression is downregulated in AKI and that the loss of DUSP26 exacerbates tubular apoptosis, whereas DUSP26 overexpression confers protection against both CIS-AKI and IR-AKI. Mechanistically, they illustrate that DUSP26 interacts with p53 and specifically dephosphorylates Ser312, attenuating p53’s pro-apoptotic transcriptional activity. The p53 S312A mutant notably replicates the protective effect of DUSP26 overexpression, while NSC87877 exacerbates harm in a liver ischemia model, indicating a more extensive organ-protective function.

The study examines a critical issue: how phosphatases mitigate stress-induced p53 in acute kidney injury, within a domain where p53 is extensively recognized as a principal facilitator of tubular cell apoptosis. The study is methodologically rigorous and comprehensive, encompassing human data, genetic models, CRISPR cell lines, biochemical assays (using a proprietary phospho-p53S312 antibody and phosphoproteomics), and two separate AKI models, hence providing robust support for the conclusions. The discovery that DUSP26 influences a non-canonical p53 site (Ser312) is unprecedented and may have significant ramifications for renal disease and cancer biology. Furthermore, the translational aspect of human tissue and liver injury is a strength, indicating that this pathway may be amenable to therapeutic targeting.

The work possesses significant scientific worth and importance to nephrology and cell death research. Nevertheless, essential clarifications and supplementary evidence are required to comprehensively substantiate the assertions (see to points

below).

We are extremely grateful to the reviewer for the detailed, insightful, and highly positive evaluation of our work. We are very encouraged by the positive assessment of our study as “methodologically rigorous and comprehensive,” noting that our discovery of the S312 site is “unprecedented” and possesses “significant scientific worth and importance.”

We have carefully addressed all the “essential clarifications and supplementary evidence” the reviewer requested. We believe the new data, supplementary figures, and revisions have substantially strengthened the manuscript. Below are the point-by-point responses to each of the reviewer’s points.

Major issues

1. The ethic number should be provided on line 96.

We thank the reviewer for pointing out this important omission. As requested, we have now added the ethics approval number and relevant details to the Methods section of the revised manuscript (Line 98).

2. To further clarify the relationship between DUSP26 and kidney function, Figure 1 should include the correlation between DUSP26 and creatine, nitrogen, and eGFR.

We thank the reviewer for this excellent suggestion to strengthen the clinical relevance of our findings. As requested, we have analyzed the correlation between DUSP26 expression (LTL (+) IOD) and the available clinical kidney function parameters for this patient cohort. We found a strong, statistically significant positive correlation between DUSP26 expression and the estimated Glomerular Filtration Rate (eGFR) ($r=0.7864$, $P=0.0005$). This new analysis has been added to the revised manuscript in Figure 1J to support the conclusion.

Figure 1. Down-regulation of DUSP26 in human AKI.

(A) Principal Component Analysis (PCA) plots of three Gene Expression Omnibus (GEO) datasets (GSE21374, GSE30718, GSE43974) before and after batch correction.

(B) Volcano plot of differentially expressed genes (DEGs) between AKI and control samples. Pink dots indicate upregulated genes; green dots indicate downregulated genes.

(C) Least Absolute Shrinkage and Selection Operator (LASSO) regression analysis. The y-axis shows binomial deviance, and the x-axis shows $\log(\lambda)$. The dashed line indicates the optimal λ value.

(D) Support Vector Machine-Recursive Feature Elimination (SVM-RFE) analysis, showing the 10-fold cross-validation (CV) error relative to the number of features.

(E) Venn diagram showing the 7 overlapping AKI-related genes identified by both LASSO and SVM-RFE.

(F) Line plot showing the normalized expression patterns (z-score) of the 7 overlapping genes in control and AKI samples.

(G) Receiver Operating Characteristic (ROC) curve analysis for the diagnostic performance of the overlapping genes, showing Area Under the Curve (AUC) values (top table). The graph (bottom)

details the ROC curve for DUSP26 (AUC = 0.70).

(H) Representative immunofluorescence staining of human kidney biopsy specimens from healthy controls (n=7) and AKI patients (n=8). Staining shows DUSP26 (magenta), proximal tubule marker LTL (green), and nuclei (Hoechst, blue). Scale bar = 50 μ m.

(I) Quantification of DUSP26 expression (Integrated Optical Density, IOD / area) in LTL-negative (LTL (-)) and LTL-positive (LTL (+)) tubules from (H).

(J) Left: Correlation between LTL (+) DUSP26 IOD and the Acute Tubular Necrosis (ATN) score. Right: Correlation between LTL (+) DUSP26 IOD and estimated Glomerular Filtration Rate (eGFR) (ml/min/1.73m²).

Data in (I) are presented as mean \pm SEM; statistical significance was determined using a two-way ANOVA with Sidak's post-hoc test. Data in (J) were analyzed using Pearson's correlation coefficient (r). **** P<0.0001.

3. Figure 3E illustrates the 20-hour treatment of BUMPT cells with the DUSP26 phosphatase inhibitor NSC87877 (5 μ M) and cisplatin (20 μ M). Figure 3i illustrates the TUNEL staining of DUSP26-overexpressing cells after 24 hours of cisplatin treatment. The time point for tunel staining is distinct; how can this be elucidated?

The reviewer is correct that different time points were used for these two experiments, and this was an intentional experimental design to optimally capture the distinct biological questions being asked (i.e., loss-of-function vs. gain-of-function).

For Figure 3E: This experiment uses the DUSP26 inhibitor NSC87877 to test if inhibiting DUSP26 exacerbates injury. For this, we used a 20-hour cisplatin treatment. This earlier time point represents a phase where the injury is significant but not yet maximal. Our rationale was that this provides a more sensitive window to observe an exacerbation of apoptosis. If we had waited until 24h, the severe, saturated damage might mask the additional harm caused by DUSP26 inhibition.

For Figure 3I: This experiment tests if DUSP26 overexpression protects against injury. For this, we used a 24-hour cisplatin treatment. At this later time point, the cisplatin-induced damage and apoptosis are robust. This provides a good baseline against which the protective effects of DUSP26 overexpression can be best observed and quantified.

4. The PT-Dusp26-KI model is crucial; however, the manuscript fails to demonstrate the baseline DUSP26 levels of these animals. The authors should quantify the expression of DUSP26 in KI proximal tubules compared to WT proximal tubules, for example, through immunostaining or Western blotting of DUSP26 in untreated

kidneys.

We agree that further validating the baseline overexpression in our KI model is essential, and we have now performed the requested experiments.

To confirm DUSP26 expression levels under baseline conditions, we have added two new panels to Figure 6:

In Figure 6C, we have added new co-immunofluorescence staining data for DUSP26 (magenta) and the proximal tubule marker LTL (green). This visually confirms the specific and elevated expression of DUSP26 in the proximal tubules of PT-DUSP26-KI mice compared to WT littermates.

In Figure 6D, we have added a new Western blot analysis of kidney lysates from untreated WT and KI mice. This result quantitatively confirms the significantly elevated DUSP26 protein levels in the KI group.

Figure 6. Proximal tubule-specific knock-in of DUSP26 attenuates AKI.

(A) Conditional *Dusp26* overexpression mice (*Rosa26*loxP-STOP-loxP-*Dusp26*) were crossed with *Pepck-Cre* mice to generate *Rosa26*loxP-STOP-loxP-*Dusp26*; *Pepck-Cre* mice (designated as PT-*Dusp26*-KI), with Cre-negative littermates as controls (designated as PT-*Dusp26*-WT).

(B) Representative PCR genotyping of tail DNA showing bands for the Wild-type (451bp), KI allele (405bp), and Cre (370bp) alleles.

(C) Representative immunofluorescence staining confirming DUSP26 (magenta) overexpression in PT-*Dusp26*-KI kidneys at baseline. The proximal tubule marker LTL (Lotus Tetragonolobus Lectin) is shown in green, and nuclei are stained with Hoechst (blue). Scale bar = 50 μ m.

(D) Representative Western blot confirming DUSP26 protein overexpression in PT-*Dusp26*-KI kidney

lysates at baseline, normalized to β -ACTIN.

(E) Serum creatinine (sCr) and blood urea nitrogen (BUN) levels in PT-Dusp26-WT and PT-Dusp26-KI mice 24 h after cisplatin injection.

(F) Representative H&E-staining in kidney sections from PT-Du26-KI mice and PT-Du26-WT mice after cisplatin treatment; scale bar, 50 μ m.

(G) Quantification of tubular damage scores based on H&E staining in (F).

(H) Immunoblot analysis of cleaved caspase-3 (c-Cas3) and NGAL in kidney tissues at 24 h after cisplatin administration.

(I) Representative TUNEL (Terminal deoxynucleotidyl transferase dUTP nick end labeling) staining for apoptosis (red) in kidney sections after cisplatin. Nuclei are stained with Hoechst (blue). Scale bar = 50 μ m.

(J) Serum creatinine and BUN levels in PT-Du26-KI and PT-Du26-WT mice following ischemia-reperfusion injury (IRI).

(K) Representative H&E staining of kidney tissues after IRI. Scale bar = 50 μ m.

(L) Quantification of tubular damage scores based on H&E staining in (K).

Data in quantitative graphs (E, G, J, L) are shown with all individual data points (n = 6 per group). Bars in (G, L) represent mean \pm SEM; data in (E, J) are shown as violin plots with median and interquartile range. Statistical significance for (E) was determined using a two-way ANOVA with Tukey's post-hoc test. Statistical significance for (G, J, and L) was determined using an unpaired, two-tailed Student's t-test. *P < 0.05, **P < 0.01, ***P < 0.001, ****P < 0.0001.

5. A recent study conducted by Xiang et al. (2024) in the field of toxicology and applied pharmacology demonstrated that DUSP26 protects against renal IR injury by restricting TAK1-JNK/p38 activation. The present study identifies a unique mechanism (p53 S312 dephosphorylation) and does not address MAPKs. The authors should elaborate on the connections between these discoveries. For instance, is the DUSP26–p53 axis independent of DUSP26's impact on JNK/p38? The discussion should acknowledge that DUSP26 likely has multiple substrates and cite Xiang et al., as well as clarify how the p53 effect fits into this context.

We thank the reviewer for bringing the recent work by Xiang et al. (2024) to our attention. We have now cited the Xiang et al. (2024) study within the paragraph that discusses the DUSP family's classical role in modulating MAPK signaling. The text now reads: "...Indeed, this classical MAPK-centric mechanism was recently suggested for DUSP26 as well ^[49]." Immediately following this, we explicitly contrast this classical mechanism with our own findings, positioning our p53 discovery as a novel, non-canonical pathway. The text now states: "...This finding, identified via our unbiased phosphoproteomic screen and validated with the p53-S312A mutant,

highlights a non-canonical, MAPK-independent axis for DUSP26 in AKI". (Line 521-530).

6. The mechanistic focus is on Ser312, which is non-canonical in AKI contexts. The authors demonstrate that p53-driven apoptosis is diminished by preventing Ser312 phosphorylation (S312A). To ensure comprehensiveness, it is necessary to determine whether DUSP26 directly influences the stability of p53 or other p53 phosphorylation sites. The proteomics identified Ser312 as the most significant hit (Fig. 7G); however, it is uncertain whether canonical sites (e.g. Ser15, Ser20) were also examined. Additionally, it would be beneficial to establish a connection between the Ser312 modification and existing literature. For instance, prior mouse studies demonstrated that the phosphorylation of Ser312 compromises the function of the tumor suppressor p53, which is consistent with our discovery that its dephosphorylation enhances cell survival. The importance of Ser312 would be emphasized by including a concise citation to that effect.

We thank the reviewer for this excellent suggestion to comprehensively examine the specificity of DUSP26 for p53-Ser312 relative to other canonical p53 phosphorylation sites and overall p53 stability.

To address this directly, we performed a new Western blot analysis using kidney lysates from our PT-Dusp26-KI and PT-Du26-WT mice following cisplatin-induced AKI (Cis-24h). These new data have been added to the revised manuscript as Supplementary Figure 7.

We probed for multiple key p53 phosphorylation sites simultaneously. The results clearly demonstrate that DUSP26 overexpression dramatically and preferentially reduced phosphorylation at Ser312 (p-p53(S312)). In contrast, phosphorylation at Ser20 (p-p53(S20)) and Ser46 (p-p53(S46)) was unchanged. While we did observe a slight reduction at Ser15 (p-p53(S15)), the effect was marginal compared to the profound and specific dephosphorylation seen at S312. This strongly supports our phosphoproteomic data (Fig. 7G) and confirms that S312 is the primary dephosphorylation site of DUSP26.

The same experiment also examined total p53 levels. We observed a modest but

significant reduction in total p53 protein in the DUSP26-KI group. This finding suggests that DUSP26-mediated dephosphorylation at S312 may also contribute to reduced p53 stability, which is consistent with the overall protective, anti-apoptotic function of DUSP26.

Finally, we have also added discussion to connect this S312 modification to the existing literature. As the reviewer noted, the function of this site is complex. Our Discussion now addresses this by citing studies suggesting the role of S312 is “debated” and “context-dependent”, referencing key literature where S312 phosphorylation is shown to be dispensable [41] versus where it is necessary for p53’s full function [42, 43]. (Line 491-497). We believe this comprehensive discussion accurately situates our novel findings (that DUSP26-mediated dephosphorylation at S312 is protective in AKI) within the known complexity of p53 regulation.

Supplementary Figure 7. DUSP26 overexpression in vivo specifically inhibits p53-Ser312 phosphorylation, but not N-terminal sites, during cisplatin-induced AKI.

(A) Representative Western blot analysis of phosphorylated p53 at Ser312 (p-p53(S312)), Ser20 (p-p53(S20)), Ser15 (p-p53(S15)), Ser46 (p-p53(S46)), and total p53. Kidney lysates were collected from PT-Dusp26-WT and PT-Dusp26-KI mice 48 hours after cisplatin administration. β -ACTIN served as a loading control.

(B) Quantification of Western blot band intensities from (A) for p-p53(S312), p-p53(S20), p-p53(S15), p-p53(S46), and total p53. All levels were normalized to β -ACTIN and expressed as fold change relative to the PT-Dusp26-WT group.

Data in bar charts are shown with all individual data points (n = 6 per group). Bars represent mean \pm SEM. Statistical significance was determined using an unpaired, two-tailed Student’s t-test. *P < 0.05, ***P < 0.001, ns (not significant).

7. The mechanistic insight and compelling observation that DUSP26 downregulation is mediated by promoter hypermethylation (Fig. 3) are noteworthy. Nevertheless, the authors' previous study, which was published in *Kidney International* (2017), indicated that the treatment with the DNA methyltransferase inhibitor 5-Aza actually exacerbated tubular cell mortality in cisplatin-induced AKI. This evident discrepancy is worthy of discussion. The authors should emphasize in the Discussion that 5-Aza may promote cell death because of its extensive epigenetic effects or potential off-target consequences, despite the fact that it restores DUSP26 expression in injured tubular cells.

We thank the reviewer for this insightful comment. This apparent discrepancy was also raised by Reviewer #1, which prompted us to perform a new experiment (now included as Supplementary Figure 8). In this experiment, we confirmed that while 5-Aza treatment successfully restored DUSP26 expression (Fig. 3E) and partially reduced p-p53(S312), it paradoxically failed to reduce (and even increased) cleaved caspase 3 levels.

To further clarify this, we have now added a new paragraph (Line 536-546), clarifying that 5-Aza's global, non-specific demethylating activity likely activates other pro-apoptotic pathways, which masks the benefit of DUSP26 restoration and exacerbates the overall injury.

Supplementary Figure 8. 5-Aza treatment paradoxically increases apoptosis despite reducing p-p53(S312) phosphorylation in cisplatin-injured cells.

(A) Representative Western blot analysis of phosphorylated p53 at Ser312 (p-p53(S312)), total p53, and cleaved caspase-3 (c-Cas3). BUMPT cells were treated with cisplatin (Cis) in the presence of vehicle (Ve) or the DNMT inhibitor 5-Aza-2'-deoxycytidine (Aza) for 24 hours. β -ACTIN served as a loading control.

(B) Quantification of Western blot band intensities from (A) for p-p53(S312), total p53, and c-Cas3. Data are normalized to β -ACTIN and expressed as fold change relative to the Cis+Ve group. Data in

bar charts are shown with all individual data points (n = 4 per group). Bars represent mean \pm SEM. Statistical significance was determined using an unpaired, two-tailed Student's t-test. *P < 0.05, ns (not significant).

8. The representative immunofluorescence image in Fig. 1H, which depicts DUSP26 expression in the control group, is somewhat less clear and well-defined than the corresponding AKI image. The authors may wish to select a higher-resolution image that more clearly illustrates tubular DUSP26 expression in control tissue in order to enhance visual consistency and clarity.

As suggested, we have now replaced the representative immunofluorescence image in Figure 1H. We have selected a new, higher-resolution image that we believe more clearly and accurately illustrates the baseline DUSP26 expression in the proximal tubules of control tissue. This change enhances the visual consistency between the control and AKI panels.

Figure 1. Down-regulation of DUSP26 in human AKI.

(A) Principal Component Analysis (PCA) plots of three Gene Expression Omnibus (GEO) datasets (GSE21374, GSE30718, GSE43974) before and after batch correction.

(B) Volcano plot of differentially expressed genes (DEGs) between AKI and control samples. Pink dots indicate upregulated genes; green dots indicate downregulated genes.

(C) Least Absolute Shrinkage and Selection Operator (LASSO) regression analysis. The y-axis shows binomial deviance, and the x-axis shows $\log(\lambda)$. The dashed line indicates the optimal λ value.

(D) Support Vector Machine-Recursive Feature Elimination (SVM-RFE) analysis, showing the 10-fold cross-validation (CV) error relative to the number of features.

(E) Venn diagram showing the 7 overlapping AKI-related genes identified by both LASSO and SVM-RFE.

(F) Line plot showing the normalized expression patterns (z-score) of the 7 overlapping genes in control and AKI samples.

(G) Receiver Operating Characteristic (ROC) curve analysis for the diagnostic performance of the overlapping genes, showing Area Under the Curve (AUC) values (top table). The graph (bottom)

details the ROC curve for DUSP26 (AUC = 0.70).

(H) Representative immunofluorescence staining of human kidney biopsy specimens from healthy controls (n=7) and AKI patients (n=8). Staining shows DUSP26 (magenta), proximal tubule marker LTL (green), and nuclei (Hoechst, blue). Scale bar = 50 μ m.

(I) Quantification of DUSP26 expression (Integrated Optical Density, IOD / area) in LTL-negative (LTL(-)) and LTL-positive (LTL(+)) tubules from (H).

(J) Left: Correlation between LTL(+) DUSP26 IOD and the Acute Tubular Necrosis (ATN) score.. Right: Correlation between LTL (+) DUSP26 IOD and estimated Glomerular Filtration Rate (eGFR) (ml/min/1.73m²).

Data in (I) are presented as mean \pm SEM; statistical significance was determined using a two-way ANOVA with Sidak's post-hoc test. Data in (J) were analyzed using Pearson's correlation coefficient (r). **** P<0.0001.

Additionally, there are a few minor issues that require attention.

1. In Fig. 6A (schematic of knock-in strategy), it would help to label the DUSP26 cDNA cassette in the Rosa26 locus for clarity.

As suggested, we have now explicitly labeled the “Rosa26-DUSP26 cDNA cassette” in the revised schematic in Figure 6A to improve clarity.

2. All figure legends should specify n and statistical tests (most do). For example, Fig. 1H–J should note how many human samples were quantified (n=8,7 are in Supp Table). Fig. 8C–E (cell images and blots) should state the number of replicates.

As requested, we have now conducted a thorough review and revised all figure legends throughout the manuscript to ensure that the biological replicate number (n) and the specific statistical test used are clearly stated for every panel.

3. In Figs. 7–9, ensure that asterisks (*) are defined (the legends mostly do). For clarity, abbreviations (e.g. AKI, IRI, HIRI) should be spelled out once in the legend.

As suggested, we have revised the legends for Figure 7, Figure 8, and the newly designated Supplementary Figure 5 (formerly Figure 9) to ensure all statistical

asterisks (*) are clearly defined and all key abbreviations (e.g., AKI, IRI, HIRI) are spelled out upon their first use.

4. Introduction: “Dual specificity phosphatase 26 (DUSP26) is an atypical MAPK phosphatase.” Perhaps briefly define “atypical” (lacks ERK-binding motif) or cite a ref.

We have revised the Introduction to clarify that DUSP26 is considered “atypical” because it lacks the conserved ERK-binding motif. This clarification has been added to the original sentence, which was already supported by a citation (line 66).

5. Methods: Indicate how ATN histology was scored. The “Suzuki criteria” for liver should be defined at first mention (though that’s in legend).

We thank the reviewer for these suggestions. As requested, we have revised the Methods section to address both points:

In the “Histology and staining” subsection, we have now included a detailed definition of the semi-quantitative scoring system used for acute tubular necrosis (ATN).

In the “Hepatic ischemia-reperfusion injury (HIRI) model” subsection, we have now added a definition of the Suzuki criteria (Supplementary Methods 11).

6. Discussion: In the paragraph starting “Integrating human AKI transcriptomic datasets...”, it may read better to split it into two shorter sentences for readability.

As requested, we have now split this sentence into two shorter sentences in the revised Discussion (line 486-490).

7. Check consistency: the paper uses “Dusp26” vs “DUSP26” (gene vs protein). Ensure italicization for the gene if referring to mouse *Dusp26* gene vs protein.

We have now thoroughly reviewed and corrected the entire manuscript to ensure we adhere to the standard nomenclature conventions. As requested, we have standardized the following:

The mouse gene is now referred to as *Dusp26* (title case, italicized).

The human gene is referred to as *DUSP26* (all caps, italicized).

All protein (both human and mouse) is referred to as DUSP26 (all caps, non-italicized).

Reviewer #4 (Remarks to the Author):

The manuscript proposes that DUSP26 protects against AKI by selectively dephosphorylating p53 at Ser312 (human Ser315) and that DUSP26 is downregulated via promoter hypermethylation. The conceptual space is narrow: the p53–AKI link is already extensively established, and DUSP26 has been reported as a p53 regulator in other cell types (e.g., neuroblastoma; PMID:20562916), including reports of Ser20/Ser37 dephosphorylation. Against this background, the evidence presented for a Ser312-focused mechanism and an epigenetic cause for DUSP26 loss is not yet sufficient.

We thank the reviewer for this critical evaluation. In this study, we have identified DUSP26 as a key phosphatase of p53 in AKI. We have further pinpointed Ser312 in p53 as the main target site of DUSP26 in this disease. Moreover, we have demonstrated DUSP26 is subjected to epigenetic regulation via promoter hypermethylation. These findings, as recognized by other 3 reviewers, are novel in the research field. In the revision, we have added substantial new experimental data, including new phosphorylation site-specific Western blots and quantitative bisulfite sequencing, to further strengthen the work.

1. The authors should more clearly distinguish a Ser312-centered pathway from the established N-terminal mechanisms (Ser20/Ser37) within the same renal/AKI models and conditions. As presented, it remains difficult to exclude that the observed effects are mediated primarily through the known N-terminal sites rather than Ser312.

To directly address this concern, we have analyzed p53 phosphorylation at Ser312 as well as the key N-terminal sites (Ser15, Ser20, and Ser46) in PT-DUSP26-WT and PT-DUSP26-KI mice kidneys after cisplatin treatment. The results (shown in new Supplementary Figure 7B) demonstrate the clear regulatory specificity of DUSP26 towards Ser312: DUSP26 overexpression (PT-Du26-KI) profoundly inhibited the phosphorylation of p53 at Ser312, whereas no significant effects on p-p53(S20) and p-p53(S46). We did observe a modest decrease in

p-p53(S15) and total p53 levels, but this effect is minor compared to the overwhelming inhibition observed at S312. This in vivo specificity is consistent with our initial unbiased global phosphoproteomic screening result (Figure 7G). Thus, both unbiased omics analysis and our new in vivo data (showing profound S312 inhibition by DUSP26 knock-in with no effect on S20 or S46 and marginal effect on S15) suggest that the observed protective effect of DUSP26 is mediated preferentially by dephosphorylating p-p53 at S312.

Supplementary Figure 7. DUSP26 overexpression in vivo specifically inhibits p53-Ser312 phosphorylation, but not N-terminal sites, during cisplatin-induced AKI.

(A) Representative Western blot analysis of phosphorylated p53 at Ser312 (p-p53(S312)), Ser20 (p-p53(S20)), Ser15 (p-p53(S15)), Ser46 (p-p53(S46)), and total p53. Kidney lysates were collected from PT-Dusp26-WT and PT-Dusp26-KI mice 48 hours after cisplatin administration. β -ACTIN served as a loading control.

(B) Quantification of Western blot band intensities from (A) for p-p53(S312), p-p53(S20), p-p53(S15), p-p53(S46), and total p53. All levels were normalized to β -ACTIN and expressed as fold change relative to the PT-Dusp26-WT group.

Data in bar charts are shown with all individual data points (n = 6 per group). Bars represent mean \pm SEM. Statistical significance was determined using an unpaired, two-tailed Student's t-test. *P < 0.05, ***P < 0.001, ns (not significant).

2. There is no in-vivo evidence that Ser312 phosphorylation is required. Although renal p-p53(Ser312) is reduced in PT-Dusp26-KI mice after injury, this is correlative rather than a test of necessity. Genetic evidence—for example, a non-phosphorylatable p53-S312A knock-in would be needed to show requirement.

We thank the reviewer for this insightful and constructive comment. While a

p53-S312A knock-in mouse model would be ideal, generating this model is a time-consuming process that extends far beyond the revision period. After discussing with Dr. Thomas Chan (the handling editor of this manuscript at Nature Communications), we have his approval to conduct an alternative genetic rescue experiment, in which wild-type (WT) p53 and its non-phosphorylatable S312A mutant were separately expressed in p53 knockout (KO) mice kidneys using a kidney tubule-specific AAV delivery system.

Specifically, we used a kidney proximal tubule-specific Ksp-cadherin promoter-driven AAV to re-express either WT p53, S312A p53, or an empty vector in kidney tubules via tail vein injection (Figure 9A). Immunofluorescence confirmed efficient and specific expression in the kidney (Figure 9C). The mice were then subjected to cisplatin treatment. As shown in the Western blot in Figure 9D, a strong p-p53(S312) signal was detected in kidney tissues transduced with WT p53. In stark contrast, this signal was absent in mice kidneys with the S312A mutant, despite equivalent expression levels of total p53. Importantly, WT p53 mice showed severe renal dysfunction (significantly elevated BUN and sCr) following cisplatin treatment, whereas the mice with S312A mutant were resistant to cisplatin injury, showing much lower BUN and sCr levels (Figure 9E). Histological analysis with H&E (Figure 9F, G) and TUNEL staining (Figure 9H, I) revealed extensive tubular injury and apoptosis in the WT p53 mice kidneys. Conversely, the S312A p53 group exhibited markedly reduced tubular damage and significantly fewer apoptotic cells. These results are consistent with the expression of kidney injury (NGAL) and apoptotic (cleaved caspase-3) markers (Figure 9J, K). The p53 target genes Bax and Puma were dramatically upregulated in the WT p53 group but were significantly blunted in the S312A p53 group (Figure L). In summary, these new in vivo data from our genetic rescue model provide direct evidence that the phosphorylation of p53 at Ser312 is required for its full pro-apoptotic function and the subsequent development of acute kidney injury.

Figure 9. Phosphorylation of p53 at Ser312 is required for its pro-apoptotic function and AKI in vivo.

(A) Schematic illustration of the genetic rescue model. p53-knockout (KO) mice were injected via tail vein with an AAV (adeno-associated virus) carrying a kidney proximal tubule-specific Ksp-cadherin promoter driving either an empty vector (EV), wild-type p53 (p53-WT), or a non-phosphorylatable p53-S312A mutant construct.

(B) Representative PCR genotyping of tail DNA confirming p53-KO status.

(C) Representative immunofluorescence analysis of whole kidney sections showing efficient AAV transduction (green fluorescence, indicative of AAV-mediated gene expression) in renal tubular epithelial cells of p53-KO mice at Day 21 post-injection. Nuclei are stained with Hoechst (blue). (Original magnification, 2x).

(D) Representative Western blot analysis of total p53 and phosphorylated p53 at Ser312 (p-p53(S312)) in kidney lysates from p53-KO mice rescued with EV, p53-WT, or p53-S312A, 48 hours after cisplatin administration. β -ACTIN served as a loading control.

(E) Serum blood urea nitrogen (BUN) and creatinine (sCr) levels in p53-KO mice rescued with EV, p53-WT, or p53-S312A, 48 hours after cisplatin (Cis) injection.

(F) Representative H&E-stained kidney sections from p53-KO mice rescued with EV, p53-WT, or p53-S312A, 48 hours after cisplatin treatment. Scale bar = 50 μ m.

(G) Quantification of tubular damage scores based on H&E staining in (F).

(H) Representative TUNEL (Terminal deoxynucleotidyl transferase dUTP nick end labeling) staining for apoptosis (red) in kidney sections from p53-KO mice rescued with EV, p53-WT, or p53-S312A, 48 hours after cisplatin treatment. Nuclei are stained with Hoechst (blue). Scale bar = 50 μ m.

(I) Quantification of TUNEL-positive nuclei percentage based on staining in (H).

(J) Representative Western blot analysis of cleaved caspase-3 (c-Cas3) and NGAL in kidney lysates from p53-KO mice rescued with EV, p53-WT, or p53-S312A, 48 hours after cisplatin administration. β -ACTIN served as a loading control.

(K) Quantification of Western blot band intensities for c-Cas3 and NGAL from (J), normalized to β -ACTIN and expressed as fold change relative to the EV group.

(L) Quantitative PCR (qPCR) analysis of mRNA levels for the p53 target genes Bax and Puma in kidney tissues from p53-KO mice rescued with EV, p53-WT, or p53-S312A, 48 hours after cisplatin administration. mRNA levels are normalized to an internal control and expressed as fold change relative to the EV group.

Data in bar charts (E, G, I, K, L) are shown with all individual data points (n =3-4 per group). Bars represent mean \pm SEM. Statistical significance was determined using one-way ANOVA with Tukey's post-hoc test. *P < 0.05, **P < 0.01, ***P < 0.001.

3. The epigenetic evidence remains preliminary. MSP is qualitative and does not resolve methylation at individual CpG sites. Please provide quantitative, locus-resolved methylation across the DUSP26 promoter, preferably using bisulfite sequencing, and relate these measurements to DUSP26 mRNA and protein in matched samples. If a causal role is proposed, include a functional rescue using demethylation by pharmacological agents or locus-targeted editing, and, where feasible, add chromatin evidence by demonstrating DNMT occupancy and relevant histone marks with ChIP.

We thank the reviewer for this constructive suggestion and accordingly, we have performed all three suggested experiments.

1) Quantitative Bisulfite Sequencing: we performed quantitative, locus-resolved bisulfite sequencing of the *Dusp26* promoter in kidney tissue. The new data (now Figure 3G, H, I) confirm our MSP findings, showing significant hypermethylation at multiple specific CpG sites in AKI kidneys compared to controls.

2) Functional Rescue: We performed the suggested functional rescue using the DNMT inhibitor 5-Aza. As shown in Figure 3D and 3E, 5-Aza treatment successfully restored both *Dusp26* mRNA and protein expression in tubular cells during cisplatin and H/R treatment.

3) ChIP assay: We performed the suggested ChIP-qPCR assay showing a significant increase in the enrichment of DNMT1 at the *Dusp26* promoter in cisplatin-treated cells, directly demonstrating DNMT occupancy.

Together, these experiments have provided robust, converging lines of evidence that *Dusp26* silencing in AKI is a causal, DNMT-mediated event.

Figure 3. Promoter methylation mediates DUSP26 downregulation in proximal tubular epithelial cells following injury.

(A) Visualization of CpG island distribution within the *Dusp26* promoter region using the MethPrimer database.

(B) Genomic DNA was subjected to bisulfite conversion, and methylation-specific (M) and

unmethylated (U) primers targeting the CpG-rich region of the *Dusp26* promoter were designed for PCR amplification.

(C) Methylation-specific PCR (MSP) showing promoter methylation status of *DUSP26* in control, cisplatin, and cisplatin+5-Aza groups, as well as in control, H/R, and H/R+5-Aza groups.

(D) qRT-PCR time course of *Dusp26* mRNA expression in BUMPT cells treated with cisplatin (20 μ M) in the presence of vehicle (Ve) or 5-Aza (1 μ M) for the indicated times (n=4).

(E) Representative Western blot showing *DUSP26* protein expression in BUMPT cells treated with 5-Aza following cisplatin (24 h) or H/R (8 h) induction (n=4).

(F) Chromatin Immunoprecipitation (ChIP)-qPCR analysis showing the relative enrichment of DNMT1 at the *Dusp26* promoter in BUMPT cells (Control vs. Cisplatin 24h).

(G) Bisulfite sequencing (BS-seq) analysis of three different amplicons (*Dusp26_1*, *Dusp26_2*, *Dusp26_3*) in the *Dusp26* promoter from control (Con) and IRI-AKI mouse kidney tissue (n=3). Bars show the methylation percentage at individual CpG sites.

(H) Heatmap visualizing the methylation frequency of individual CpG sites within the *Dusp26_1* amplicon from (G).

(I) Bar graphs showing the quantification of methylation percentage at specific, representative CpG sites from the BS-seq analysis in (G) and (H).

Data in bar charts (F, I) are presented as mean \pm SEM. Statistical significance was determined using a two-way ANOVA with Sidak's post-hoc test (for D) or an unpaired, two-tailed Student's t-test (for F and I), *P < 0.05, **P < 0.01.

4. The pharmacologic inhibitor NSC87877 is known to have off-target effects. Genetic approaches to modulate *DUSP26* activity would help confirm specificity and reinforce the mechanistic conclusions.

We agree that NSC87877 may have off-target effects. To complement the test of NSC87877, we utilized siRNA-mediated knockdown of *Dusp26* in tubular cells (Figure 4A-D). This genetic approach phenocopied the NSC87877 inhibitor effect, resulting in an exacerbation of cisplatin-induced apoptosis as shown by cleaved caspase-3. Moreover, we employed genetic overexpression models both in vivo (the PT-*Dusp26*-KI mice, Figure 6) and in vitro (lentiviral *DUSP26* overexpression, Figure 4G-L), which demonstrated robust protective effects, confirming the role of *DUSP26*.

To prove the mechanism relies on the phosphatase activity (which NSC87877 targets), we tested the C152S phosphatase-dead mutant of *DUSP26* (Figure 4N, O). This genetic mutant failed to provide protection, confirming the effect is dependent on *DUSP26*'s catalytic function.

In summary, our experiment using three genetic approaches (knockdown,

overexpression, and the catalytic mutant) support the observation of the NSC87877 inhibitor, confirming the role of the phosphatase activity of DUSP26.

5. Long-term outcomes after AKI, such as fibrosis and maladaptive repair, are not assessed. Given p53's known role in these processes, examining later time points or discussing the potential implications would enhance the translational relevance. Although DUSP26 appears to suppress apoptosis and exert protective effects in the acute phase, persistent survival of abnormal cells may increase the risk of tumorigenesis or, conversely, inhibit proper tubular regeneration, potentially promoting the transition from AKI to CKD.

We thank the reviewer for this insightful comment. Indeed, p53's role in kidney repair is complex and suppressing acute injury (as DUSP26 does in this study) may have different or unintended long-term consequences. Performing a comprehensive *in vivo* study to track these long-term outcomes (e.g., fibrosis at later time points) would be a substantial undertaking and is beyond the scope of the present study, which is focused on the mechanisms of acute injury. Nonetheless, we have followed the reviewer's suggestion to significantly expand the "Limitations" section of our Discussion (line 581-586). Briefly, we state that this study is focused on the acute injury phase, and the role of DUSP26 in long-term outcomes like maladaptive repair and fibrosis remains an important, open question, especially given the known complicated role of p53 after kidney injury.

The human biopsy data are limited by small sample size and heterogeneous underlying diseases, which complicate interpretation of associations such as the inverse correlation between DUSP26 and ATN severity.

This is another important note. To minimize potential bias, all included patients exhibited histopathological evidence of acute tubular necrosis (ATN), and the degree of ATN was quantified by two renal pathologists under blinded conditions to ensure objectivity. Despite these constraints, DUSP26 expression showed a statistically significant inverse correlation with ATN severity and, importantly, a positive correlation with estimated glomerular filtration rate (eGFR), indicating that higher

DUSP26 expression is associated with better renal function. These clinicopathological trends are consistent with our findings in animal models and in-vitro systems, together supporting the mechanistic link between DUSP26 activation and tubular protection. We have revised the Discussion to further acknowledge the limitations of the human cohort while emphasizing the translational relevance and the need for larger, disease-specific validation studies (line 575-578).

6. The liver ischemia–reperfusion injury extension is interesting, but it diverts focus from the kidney-centric narrative. Why was the liver selected for investigation? What is the expression profile of DUSP26 across tissues? Consider streamlining or moving some of these data to the Supplementary Information unless they are essential to the main conclusions.

We agree that the data on hepatic ischemia-reperfusion injury (HIRI), while interesting, may divert focus from the primary, kidney-centric narrative of our study. The initial rationale was to explore if the protective mechanism of DUSP26 was specific to the kidney or if it represented a more general stress-response mechanism, but we agree that this is not essential to the main conclusions. Therefore, we have moved this entire section (including the data on ALT/AST, H&E, TUNEL, and Western blots for the HIRI model) from the main text to the Supplementary Figure 5. Furthermore, to retain the finding without disrupting the manuscript's flow, we have added a brief mention of these supplementary findings in the Discussion section (line 509-517), suggesting that the role of DUSP26 in regulating p53-mediated apoptosis might extend to other organs, warranting future investigation.

Minor Comments:

1. Regarding the generation of proximal tubule-specific DUSP26 knock-in mice, the manuscript refers to “DUSP26 flox/flox” mice in the Method section.

We sincerely thank the reviewer for noticing this typo error. Our model is a conditional knock-in (KI), where the DUSP26 cDNA (preceded by a loxP-STOP-loxP [LSL] cassette) was inserted into the Rosa26 locus, as shown in Figure 6A. We have now corrected it to *Dusp26*^{KI/KI} throughout the revised manuscript, including the

Methods section description, the corresponding labels in Figure 6B (genotyping gel) and Supplementary Figure 1 (breeding schematic)

2. In Figure 2B and 2C, the y-axis lacks clarification regarding the reference value used for normalization. Please specify what was set as "1" in the quantification.

We have revised the legend for Figure 2 to explicitly state that all quantitative “Fold Change” data in panels B, C, F, and G were normalized to their respective control group (i.e., the “Con” or “Sham” group), which was set to a value of 1.

Reviewer #5 (Remarks to the Author)

Response: We thank the reviewer for participating in the peer-review process and for informing us of their co-review role as part of the Nature Communications training initiative.

POINT-BY-POINT RESPONSE TO REVIEWERS' COMMENTS

Reviewer #1:

The authors have added substantial amounts of results to strengthen the manuscript, and my comments have been adequately addressed. The manuscript is now acceptable for publication in Nature Communications.

We sincerely thank Reviewer #1 for the positive evaluation and for recommending acceptance. We are grateful that the additional experiments strengthened the manuscript and adequately addressed the reviewer's concerns.

Reviewer #2:

The authors have addressed my concerns.

We thank Reviewer #2 for the constructive feedback and for confirming that the concerns have been addressed.

Reviewer #3:

The author has thoroughly addressed my previous concerns, and I have carefully reviewed the questions raised by other reviewers along with the author's responses. The entire revised manuscript and figure revisions have been scrutinized, and the paper now meets publication standards. I personally recommend acceptance.

We are grateful to Reviewer #3 for the thorough reassessment of the revised manuscript and for recommending acceptance. We appreciate the reviewer's careful scrutiny of both the revised text and figures.

Reviewer #4:

The authors have responded to my primary concerns regarding the DUSP26-p53 S312 axis with extensive new experimentation. The question of specificity for S312 dephosphorylation is resolved by new in vivo data from the PT-DUSP26-KI mice, which demonstrate a remarkably preferential effect on S312 over other N-terminal phosphorylation sites (Supplementary Fig. 7). Similarly, the question of whether S312 phosphorylation is required for AKI pathogenesis has been decisively addressed by the genetic rescue model utilizing AAV-driven p53-S312A re-expression in p53-knockout mice (Fig. 9), providing conclusive in vivo evidence of necessity. Furthermore, the conclusion that DUSP26's protective effect is dependent on its

phosphatase activity is adequately supported by multiple genetic approaches, including siRNA knockdown and the phosphatase-dead C152S mutant (Fig. 4), appropriately reinforcing the pharmacological inhibitor findings. Acknowledging the long-term role of DUSP26 as a future research direction within the Discussion's limitations section is also a reasonable clarification of the study's scope.

However, a lapse in mechanistic rigor remains regarding the epigenetic regulation of DUSP26 expression. While quantitative Bisulfite Sequencing confirms the existence of hypermethylation and DNMT1 recruitment is shown by ChIP-qPCR, DNMT1 is primarily a maintenance methyltransferase. Given that DUSP26 promoter hypermethylation implies de novo methylation, the principal executors of this process are DNMT3A and DNMT3B. The mechanistic chain remains incomplete because there are no direct Chromatin Immunoprecipitation (ChIP) data demonstrating the recruitment of DNMT3A or DNMT3B to the DUSP26 promoter. Proving the direct involvement of these de novo methyltransferases is highly desirable to achieve full scientific rigor for the proposed epigenetic mechanism of gene silencing.

We appreciate Reviewer #4 for clearly identifying this mechanistic gap. To directly test the involvement of de novo DNA methyltransferases, we performed additional ChIP-qPCR assays in BUMPT cells under basal conditions and after cisplatin injury (20 μ M, 24 h), using ChIP-grade antibodies against DNMT3A and DNMT3B, with IgG as negative control and the same *Dusp26* promoter primer pair used for DNMT1 ChIP.

We observed a significant increase in DNMT3A occupancy at the CpG-rich region of the *Dusp26* promoter in cisplatin-treated cells compared with controls, supporting a de novo methylation component associated with *Dusp26* repression. In contrast, DNMT3B did not show significant enrichment at the *Dusp26* promoter under the same conditions.

We believe these new data directly address the reviewer’s concern by providing the missing mechanistic evidence for recruitment of a de novo DNA methyltransferase (DNMT3A) to the *Dusp26* promoter during injury-associated silencing.

Reviewer #5:

We thank Reviewer #5 and the co-reviewer for their time and careful evaluation of our work, and we appreciate Nature Communications’ initiative to recognize and train early career researchers in peer review.

A point-by-point response to the reviewers' comments

Manuscript ID: NCOMMS-25-49844B

Title: “DUSP26 protects against acute kidney injury by dephosphorylating p53 at serine 312”

We thank the reviewers for their time and constructive evaluation of our manuscript. Below we provide a point-by-point response to the comments included in the final revision request.

Reviewer #4 (Remarks to the Author):

The authors have fully addressed the suggestion, and I have no further comments.

We sincerely thank the reviewer for confirming that the previous concerns have been fully addressed. No further changes were requested by the reviewer, and we have therefore not made additional revisions specifically in response to Reviewer #4 in this final round.

Reviewer #5 (Remarks to the Author):

We thank Reviewer #5 for their contribution to the peer-review process and for the co-review disclosure statement. This remark does not request changes to the manuscript, and no revisions were required in response.